# Temperature and strain controls on ice deformation mechanisms: insights from the microstructures of samples deformed to progressively higher strains at -10, -20 and -30 °C

Sheng Fan[1], Travis Hager[2], David J. Prior[1], Andrew J. Cross[2,3], David L. Goldsby[2], Chao Qi[4], Marianne Negrini[1], John Wheeler[5]

[1] Department of Geology, University of Otago, Dunedin, New Zealand
[2] Department of Earth and Environmental Science, University of Pennsylvania, Philadelphia, PA, USA
[3] Department of Geology and Geophysics, Woods Hole Oceanographic Institution, Woods Hole, MA, USA
[4] Key laboratory of Earth and Planetary Physics, Institute of Geology and Geophysics, Chinese Academy of Sciences, Beijing, China
[5] Department of Earth and Ocean Sciences, University of Liverpool, Liverpool, UK

*Correspondence to:* Sheng Fan (sheng.fan@postgrad.otago.ac.nz).

**Abstract** In order to better understand ice deformation mechanisms, we document the microstructural evolution of ice with increasing strain. We include data from experiments at relatively low temperatures (-20 and -30 °C), where the microstructural evolution with axial strain has never before been documented. Polycrystalline pure water ice was deformed under a constant displacement rate (strain rate ~$1.0 \times 10^{-5}\ s^{-1}$) to progressively higher strains (~3, 5, 8, 12 and 20%) at temperatures of -10, -20 and -30 °C. Microstructural data were generated from cryogenic electron backscattered diffraction (cryo-EBSD) analyses. All deformed samples contain sub-grain (low-angle misorientations) structures with misorientation axes that lie dominantly in the basal plane suggesting the activity of dislocation creep (glide primarily on the basal plane), recovery and subgrain rotation. Grain boundaries are lobate in all experiments suggesting the operation of strain-induced grain boundary migration (GBM). Deformed ice samples are characterised by interlocking big and small grains and are, on average, finer grained than undeformed samples. Misorientation analyses between nearby grains in 2-D EBSD maps are consistent with some 2-D grains being different limbs of the same irregular grain in the 3-D volume. The proportion of repeated (i.e. interconnected) grains is greater in the higher-temperature experiments suggesting that grains have more irregular shapes, probably because GBM is more widespread at higher temperatures. The number of grains per unit area (accounting for multiple occurrences of the same 3-D grain) is higher in deformed samples than undeformed samples, and it increases with strain, suggesting that nucleation is involved in recrystallisation. "Core-and-mantle" structures (rings of small grains surrounding big grains) occur in -20 and -30 °C experiments, suggesting that subgrain rotation recrystallization is active. At temperatures warmer than -20 °C, *c*-axes develop a crystallographic preferred orientation (CPO) characterized by a cone (i.e., small circle) around the compression axis. We suggest the *c*-axis cone forms via the selective growth of grains in easy slip orientations (i.e., ~45° to shortening direction) by GBM. The opening-angle of the *c*-axis cone decreases with strain, suggesting strain-induced GBM is balanced by grain rotation. Furthermore, the opening-angle of the *c*-axis cone decreases with temperature. At -30 °C, the *c*-axis CPO changes

from a narrow cone to a cluster, parallel to compression, with increasing strain. This closure of the *c*-axis cone is interpreted as the result of a more active grain rotation together with a less effective GBM. We suggest that lattice rotation, facilitated by intracrystalline dislocation glide on the basal plane is the dominant mechanism controlling grain rotation. Low-angle neighbour-pair misorientations, relating to subgrain boundaries, are more extensive and extend to higher misorientation angles at lower temperatures and higher strains supporting a relative increase in importance of dislocation activity. As the temperature decreases, the overall CPO intensity decreases, primarily because the CPO of small grains is weaker. High-angle grain boundaries between small grains have misorientation axes that have distributed crystallographic orientations. This implies that, in contrast to subgrain boundaries, grain boundary misorientation is not controlled by crystallography. Nucleation during recrystallisation cannot be explained by subgrain rotation recrystallisation alone. Grain boundary sliding of finer grains or a different nucleation mechanism that generates grains with random orientations could explain the weaker CPO of the fine-grained fraction and the lack of crystallographic control on high-angle grain boundaries.

## 1 Introduction

Glaciers and ice sheets play key roles in shaping planetary surfaces, and form important feedbacks with climate, both on Earth (Pollard, 2010; Hudleston, 2015; Kopp et al., 2017) and elsewhere in the solar system (Hartmann, 1980; Whalley and Azizi, 2003). Understanding the controls on the flow rate of terrestrial glaciers and ice sheets is crucial, as this will be a major control on future sea level change (Bindschadler et al., 2013; Dutton et al., 2015; Bamber et al., 2019). Ice core studies and field investigations suggest the temperature of ice in Antarctica and Greenland ranges between ice melting temperature and ~-30 °C (Kamb, 2008; Montagnat et al., 2014). Glacial flow is driven by gravity and facilitated by both basal sliding along the ice-bedrock interface (and/or with the shearing of subglacial till deposits) and the internal creep of ice masses. The contribution of creep deformation to the total flow rate is controlled primarily by differential stress and temperature within the ice body (Rignot et al., 2011; Hudleston, 2015). Creep experiments show a change in the mechanical behaviour as initially isotropic polycrystalline ice is deformed (Budd and Jacka, 1989; Faria et al., 2014; Hudleston, 2015). Mechanical weakening occurs during the transition from secondary creep (minimum strain rate) to tertiary creep (quasi-constant strain rate) in constant load experiments (e.g., Budd and Jacka, 1989; Montagnat et al., 2015; Hudleston, 2015; Wilson et al 2014) and from peak stress to steady-state stress in constant displacement rate experiments (e.g., Weertman, 1983; Durham et al., 1983, 2010; Vaughan et al., 2017; Qi et al., 2017). This mechanical weakening is often referred to as strain rate "enhancement" in the glaciological and ice sheet literature (Budd and Jacka, 1989; Alley, 1992; Placidi et al., 2010; Treverrow et al 2012; Budd et al., 2013). Enhancement correlates with the development of a crystallographic preferred orientation (CPO) (Jacka and Maccagnan, 1984; Vaughan et al., 2017) and also with other microstructural changes, particularly those associated with dynamic recrystallization (Duval, 1979; Duval et al., 2010; Faria et al., 2014; Montagnat et al., 2015), including grain size reduction (Craw et al., 2018; Qi et al., 2019). Understanding the deformation and recrystallization mechanisms responsible for ice microstructure and CPO development is therefore essential for quantifying how different mechanisms contribute to ice creep enhancement in nature.

The relative roles of intracrystalline plasticity, recrystallization and grain size sensitive mechanisms, especially at low temperatures, are not well known.

In this contribution we present microstructural analyses of samples deformed to successively higher strains through the transition from peak stress to flow stress at -10, -20 and -30 °C. These conditions were chosen so that the experiments included evolution of CPO towards a cone (small circle, centred on the compression direction) that occurs at high temperature and towards a cluster (maximum parallel to the compression direction) at low temperature. Our results include microstructural data from samples deformed to progressively higher strains at -20 and -30 °C. Such data have not been presented before, and they are important, as understanding how and why different CPOs develop as a function of temperature should give a better insight into the mechanisms that control CPO development and mechanical behaviour. Furthermore, understanding CPOs in nature requires extrapolation of laboratory results to the much lower strain rates that occur in nature. To do this effectively we need to know how CPOs evolve across as wide a range of temperatures and strain rates as is possible. In this paper our objectives are to study the influences of temperature and strain on microstructure and CPO development and to discuss implications for mechanical behaviour.

## 2 Methods

### 2.1 Sample fabrication

Dense, polycrystalline ice samples were prepared by the flood-freeze (standard ice) method (Cole, 1979; Durham et al., 1983; Stern et al., 1997) to produce samples with controlled grain size, random CPO and minimised porosity. We crushed ice cubes made from frozen Milli-Q water (ultra-pure water), into ice powders. These ice powders were then sieved at -30 °C in a chest freezer, to limit the particle sizes to between 180 and 250 μm. Particles were then packed into the bottom of lightly greased stainless-steel cylindrical moulds (inner diameter 25.4 mm) to achieve a porosity of ~40%. A perforated brass spacer was placed on top of the packed ice power and the mould was sealed with a double O-ring plug. Air was removed from pore spaces with a vacuum pump after the moulds were equilibrated at 0 °C in a water-ice bath for 40 minutes. Degassed Milli-Q water (at 0 °C) was then flooded into the pore spaces. The perforated spacer prevented ice particles from floating in the water. After flooding, the moulds were transferred to a -30 °C chest freezer and placed vertically into cylindrical holes in a polystyrene block, with the base of moulds touching a copper plate at the bottom of the freezer. This ensures that a freezing front migrates slowly upwards. After 24 hours the ice samples were gently pushed out of the moulds using an Arbor press. Both ends of the cylindrical samples were cut and polished to be flat, parallel with one another, and perpendicular to the sample's long axis, and the length of the sample was recorded (Table 1). Each sample was encapsulated in a thin-walled indium jacket (~0.38 mm wall thickness), the bottom of which had already been welded to (melted against) a stainless-steel end-cap. The top of the indium tube was then welded to a steel semi-internal force gauge, with a thermally insulating zirconia spacer placed between the force gauge and sample. The sample was kept cold in a -60 °C ethanol bath (Qi et al., 2017) during welding.

## 2.2 Experimental set up and process

We conducted axial compression experiments at the Ice Physics Laboratory, University of Pennsylvania. Experiments were conducted at a nitrogen gas confining pressure of ~$20 \pm 0.5$ MPa at temperatures of -10, -20, and -30°C (±0.5°C), in a cryogenic, gas-medium apparatus (Durham et al., 1983; Heard et al., 1990). Samples were left to thermally equilibrate with the apparatus for more than 60 minutes before deformation started. Deformation experiments were performed at a constant axial displacement rate, giving an initial constant strain rate of ~$1.0 \times 10^{-5}$ s-1. The experiments were terminated once final axial true strains of ~3%, 5%, 8%, 12% and 20% were achieved. After deformation, the ice samples were immediately extracted from the apparatus, photographed and measured. To minimize thermal cracking, samples were progressively cooled to ~ -30, -100 and -196 °C in about 15 minutes, and thereafter stored in a liquid nitrogen dewar. Typical time between the end of the experiments and the start of cooling was between 10 and 30 minutes. Minor static recovery of the ice microstructures may happen on this timescale (Hidas et al., 2017), but significant change in CPO or grain size is unlikely.

## 2.3 Mechanical data processing

During each experimental run, time, displacement and load were recorded once every five to seven seconds. The axial stress was calculated from the load supported by the sample divided by the instantaneous cross-sectional area of the sample, which was calculated by assuming constant sample volume during the deformation. The sample length $L(t)$ at time t is calculated from the displacement and the initial sample length ($L_0$). From this we calculate the axial stretch ($\lambda$: Eq. (1)) and the true axial strain ($\varepsilon$: Eq. (2)) (Hobbs et al., 1976).

$$\lambda = \frac{L(t)}{L_0} \tag{1}$$

$$\varepsilon = -\ln(\lambda) \tag{2}$$

## 2.4 Cryo-EBSD data

The relatively recent development of cryo-EBSD (electron backscatter diffraction) technique (Iliescu et al., 2004, Obbard et al., 2006; Piazolo et al., 2008) enables measurement of full crystallographic orientations. EBSD maps provide quantitative microstructural data, with significant detail in ice samples with sizes up to about 70 mm by 40 mm (Prior et al., 2015). We prepared the ice samples and acquired the cryo-EBSD data following the procedures described by Prior and others (2015). Samples were cut in half along the cylindrical long-axis using a band saw in a -20 °C cold room and a ~5 mm slice was cut from half of the sample. One side of the slice, at a temperature of ~-30 to -50 °C, was placed against a copper ingot (70 mm by 35 mm) at ~5 °C. As soon as a bond formed between the ice sample and the ingot, the samples were placed in a polystyrene sample transfer box (~-100 °C). We acquired a polished sample surface for cryo-EBSD by hand lapping on grit paper. The samples were polished at ~-40 °C using grit sizes of 80, 240, 600, 1200 and 2400. The sample-ingot assemblies were then

transferred to the polystyrene sample transfer box and cooled to close to liquid nitrogen temperature, before they were transferred into the SEM for the collection of cryo-EBSD data.

EBSD data were acquired using a Zeiss Sigma VP FEGSEM combined with a NordlysF EBSD camera from Oxford Instruments. We used pressure cycling in the SEM chamber remove frost and create a damage-free sample surface (Prior et al., 2015). EBSD data were acquired at a stage temperature of ~-95 °C, with 5-7 Pa nitrogen gas pressure, 30kV accelerating voltage and a beam current of ~60 nA. For each ice sample, we collected a reconnaissance map with a step size of 30 µm from the whole section and a map with a step size of 5 µm, from a selected sub-area, for detailed microanalysis (mapped areas listed in Table 2). We acquired and montaged the raw EBSD data using Oxford Instruments' Aztec software. Details on the raw EBSD data have been summarized in Table 2. The angular resolution (error of crystallographic orientation measurement for each pixel) of the EBSD data is ~0.5°.

## 2.5 Processing of the cryo-EBSD data

Grain size, grain shape, grain boundary morphology, and CPO provide useful information for inferring ice deformation processes. We quantified these microstructural parameters (among others) from raw EBSD data using the MTEX toolbox (Bachmann et al., 2011; Mainprice et al., 2015) in MATLAB.

### 2.5.1 Grain size and subgrain size

Ice grains were reconstructed from the raw EBSD pixel maps with 5 µm step size using the MTEX algorithm of Bachmann and others (2011) with a grain boundary threshold of 10°. Grains with area equivalent diameters lower than 20 µm were removed from the data. No pixel interpolation was applied to the EBSD pixel map, preserving any non-indexed space. Deformed ice is often characterised by a development of subgrain boundaries where the misorientations between neighbouring pixels are lower than the misorientation angle threshold of grain boundaries (e.g. Montagnat et al., 2015; Weikusat et al., 2017). An ice grain can be separated into several subgrains by one or more subgrain boundaries. We calculated subgrain size using boundary misorientation thresholds of $\geq 2°$. This method counts all areas enclosed by boundaries $\geq 2°$ within area enclosed by $\geq 10°$ boundaries; no internal boundaries $\geq 2°$ will be counted as either a subgrain or a grain (see Figure 5 in Trimby et al., 1998). Grain size and subgrain size were calculated as the diameter of a circle with the area equal to the measured area of each grain or subgrain. Note that grain size or subgrain sizes represent the sizes of 2-D cross sections through 3-D grains.

The 2-D measurements of a grain will always underestimate the 3-D size. It is also possible that grains, with irregular 3-D geometries, could be appear as two or more separate grains in the same 2-D slice (Monz et al., 2020). To assess these stereological issues, we have analysed the maps with some 1-D lines; the comparison of 1-D and 2-D giving some insights (Cross et al., 2017a) into the effects of taking a 2-D slice through a 3-D volume (section S3 of supplementary material). We have also assessed how many grains are in the same orientation on a 2-D slice. Grains (in 2-D) that are in the same orientation (they have a misorientation below a defined threshold) and in reasonable proximity (which depends on grain size) are

candidates for being 2-D slices through the same grain that has an irregular geometry in 3-D. These analyses are presented in section S3 of the supplementary material.

### 2.5.2 Crystallographic preferred orientation

EBSD maps with 5 µm and 30 µm step size have been used to generate the crystallographic preferred orientation (CPO) data with one point per pixel. The CPO data were contoured with a half-width of 7.5° based on the maximum of multiples of a uniform distribution (MUD) of the points, to more clearly show the CPO patterns. CPO intensity was quantified using the M-index of Skemer and others (2005). M-indices and eigenvectors (orientation and magnitude) are consistent between CPOs generated from the EBSD maps with 30 µm and 5 µm step sizes.

Ice CPOs formed during uniaxial compression at high temperatures are often characterised by $c$-axes aligning in an open cone (i.e., a small circle) (Fig. 1(a)) around the compression axis (Kamb, 1972; Jacka and Maccagnan, 1984; Wilson et al., 2014; Jacka and Li, 2000; Qi et al., 2017). The opening-angle, $\theta$, of $c$-axes cone is considered important in indicating the relative activity of grain rotation and grain boundary migration (GBM), which are competing processes in deforming ice (Piazolo et al., 2013; Qi et al., 2017). In order to quantify cone opening-angles, we counted the number of $c$-axes that lie at a given angle (co-latitude) from the compression axis—this method is adapted from Jacka and Maccagnan (1984) and Piazolo and others (2013). In practice we counted the $c$-axes between two co-latitudes separated by a 4° interval (selected by trial, see section S1 of the supplementary material) and calculated the MUD for this co-latitude range to plot on a graph of MUD as a function of co-latitude (Fig. 1(b-c)).

### 2.5.3 Misorientation

Deformation processes may leave signatures in misorientation data (Fliervoet et al., 1999; Wheeler et al., 2001, 2003; Montagnat et al., 2015; Qi et al., 2017). Misorientation describes the rotation axis and angle required to map one lattice orientation onto another (Wheeler, et al., 2001). Because of crystal symmetry, there is more than one rotation that can be used to describe a misorientation. We chose the minimum rotation angle and corresponding rotation axis to describe misorientation (i.e., the disorientation, in the material science nomenclature—Grimmer, 1979; Morawiec 1995; Wheeler et al., 2001). Here we refer to the minimum rotation angle and corresponding rotation axis as the misorientation angle and misorientation axis, separately. Misorientation angle distributions are illustrated as histograms; misorientation axes distributions are illustrated as inverse pole figures. In this study, we applied three groups of pixel-by-pixel misorientation analyses, using the EBSD data with 5 µm step size:

> (1) Neighbour-pair misorientations: using neighbouring pixels.

> (2) Random-pair misorientations: using randomly selected pixels.

> (3) Grain boundary misorientations: using pixels along the grain boundaries of neighbouring grains.

# 3 Results

## 3.1 Starting material

Undeformed samples exhibit a foam-like microstructure with straight or slightly curved grain boundaries and polygonal grain shapes (Fig. 2(a)). The grain size distribution is slightly skewed. The frequencies of grains increase slightly from the minimum cut off grain size (20 µm) to a peak at around 300 µm, and then decrease with further increasing grain size (Fig. 2(b)). Mean and median grain sizes are 297 µm and 291 µm, respectively (Table 3). The mean and median subgrain sizes at 291 and 280 µm, respectively, are very close to mean and median grain sizes (Table 3), indicating that there are very few subgrain boundaries. CPO is close to random (Fig. 2(c)), with an M-index of ~0.004 (Table 2). Neighbour-pair and random-pair misorientation angle distributions both resemble the distribution calculated for randomly oriented hexagonal crystals (Fig. 2(d), Morawiec, 1995; Wheeler et al., 2001).

## 3.2 Mechanical data

Stress-strain curves are plotted in Fig. 3. Imposed initial strain rate and temperature are shown in Table 1 together with peak and final stresses and corresponding strain rates. The strain rate increases slightly with strain (Table 1), as is required kinematically for a shortening sample at constant displacement rate. For all the deformation runs, stress initially increases as a function of strain, before reaching a peak stress at axial strains of $0.01 \leq \varepsilon \leq 0.04$. Beyond the peak stress, stress deceases with increasing strain, with the rate of stress drop decreasing with increasing strain. The rate of stress reduction is at a minimum, for each temperature, at strains larger than ~0.1. Peak and final stresses are larger at colder temperatures. Ratios of peak stress to stresses at higher strain (e.g. final stress of ~20% strain) are approximately the same at all temperatures so that all curves, when normalised to the peak stress look similar.

## 3.3 Microstructure

EBSD data are used to generate the illustrative grain orientation maps, grain sub-structure maps, grain size distributions, subgrain size distributions and misorientation angle distributions shown in Figs. 4-6. The grain size and subgrain size distributions are presented as histograms with 4 µm bins. We only show selected areas of EBSD maps so that the reader can resolve microstructural features. Quantitative microstructural analyses are based on much larger areas than those presented in the figures (Table 2).

### 3.3.1 Sub-structure

All samples deformed at -10 °C and -20 °C show large, lobate grains interlocking with finer, less lobate grains (Fig. 4(a-b), 5(a-b)). Grain boundary lobateness increases at higher strains. The scale of lobateness—that is, the amplitude of grain boundary irregularities—is smaller at -20 °C than -10 °C. At -30 °C lobate grain boundaries are less common at low strains but are a common attribute of larger grains at 20% strain (Fig. 6(a-b)). Samples deformed at -20 °C and -30 °C to strains higher than

~12% show a "core-and-mantle" structure (Gifkins, 1976; White, 1976; Ponge and Gottstein, 1998), characterised by a "net" or "necklace" of finer grains encircling larger grains.

Distinct sub-grain boundaries can be observed in all the samples (Fig. 4 (c), 5 (c) and 6 (c)). Many of the subgrain boundaries appear to be straight, some with slight curvature. A small number have strong curvature. Interconnected subgrain boundaries can be observed in some of the grains. Subgrain boundaries subdivide grains into subgrains.

The colouring of the IPF maps changes with increasing strain, corresponding to the increasing strength of the CPO. At -10 °C, grains with near-pink-and-orange colours dominate the IPF maps at strains higher than ~8% (Fig. 4(a-b)). At -20 and -30 °C, grains with red, pink and orange colours dominate the IPF maps at ~20% strain (Fig. 5(a-b) and 6 (a-b)).

### 3.3.2 Grain size

For samples deformed to ~3% strain, the grain size distributions are strongly skewed, with a clear main peak at finer grain sizes and a tail of coarser sizes with a possible broad, poorly defined secondary peak corresponding to the mean grain size of the starting material (Fig. 4(d), 5(d) and 6(d)). As strain increases, the grain size distributions generally narrow and shift towards finer grain sizes (Fig. 3(d), 4(d) and 5(d)). The secondary peak, corresponding to the mean grain size of the starting material, becomes harder to see with increasing strain and is absent by 12% strain at all temperatures.

For each sample, we calculated the mean grain diameter ($\bar{D}$) and square mean root diameter ($D_{SMR} = (\overline{\sqrt{D}})^2$) and estimated the peak grain diameter ($D_{peak}$) by visual inspection of the distributions. $D_{SMR}$ minimizes the bias from very large grains in the calculation of an average. To better describe the statistics of the skewed or bimodal grain size distributions, we also calculated median grain size ($D_{median}$), lower quartile ($D_{q,25\%}$) and higher quartile ($D_{q,75\%}$). Data are presented in Table 3. $\bar{D}$, $D_{SMR}$, $D_{median}$ and $D_{peak}$ have the relation of $\bar{D} > D_{SMR} > D_{median} > D_{peak}$, and converge as the strain increases (Fig. 4 (d), 5 (d), 6 (d) and Table 3).

We wish to compare the microstructures associated with different grain size populations. Ideally, we would like to distinguish the microstructural and CPO characteristics of recrystallised grains (i.e., grains formed during the experiment) and remnant grains (i.e., remnants of grains present in the starting material). While the mean diameter, $\bar{D}$, is commonly used to represent a characteristic sample grain size (e.g., Jacka and Maccagnan, 1984; Piazolo et al., 2013; Vaughan et al., 2017; Qi et al., 2017; Qi et al., 2019) it averages the recrystallised and remnant fractions. Lopez-Sanchez and Llana-Fúnez (2015) showed that the frequency peak ($D_{peak}$) of a grain size distribution provides a robust measure of the recrystallized grain size from the study of deformed rock samples. In our data, the population of grains smaller than $D_{peak}$ is too small, in many samples, to provide representative data. Instead, we define, for each temperature series, a threshold grain size, equal to the $D_{SMR}$ of the sample deformed to ~12% strain. Grains with the grain sizes greater than the threshold are classified as "big" grains and grains smaller than or equal to the threshold are classified as "small" grains.

Stereological artefacts inevitably arise from looking at microstructures on two-dimensional sections. Here we analyse two distinct (albeit related) stereological issues. The first issue relates to the *misidentification* of "small" grains, as these could

appear from slices cut close to the perimeter of a large grain in 3-D (Underwood, 1973). The second issue relates to the *oversampling* of grains that have highly irregular, branching shapes in 3-D and appear more than once on a 2-D surface (Hooke and Hudleston, 1980; Monz et al, 2020).

It is clear that 2-D grain size measurements will always underestimate the "true" 3-D grain size (Underwood, 1973; Berger et al., 2011). A trickier problem lies in understanding how, specifically, grain size distributions in two dimensions relate to those in three dimensions. As we have categorized grains in 2-D maps as "big" and "small" grain, we need to assess the likelihood of a "small" grain in 2-D being a slice through of a "big" grain in 3-D. One way to estimate this is to further flatten the two-dimensional data into one-dimension and measure grain sizes along a line. From this analysis, we can evaluate the likelihood that a "small" 1-D grain is indeed a "small" grain in 2-D. This analysis is presented in section S3 of the supplementary material. At ~20% strain the percentage of "small" grains on a 1-D line that correspond to "small" grains in the 2-D EBSD map is 64%, 76% and 43% at -30, -20 and -10 °C, respectively. These data suggest that at 20% strain the presence of "small" grains in 3-D is likely, with the confidence in this statement increasing at reduced temperatures. Another observation supports this: at 20% strain many "small" grains have "small" grain neighbours (Fig. 4-7). At -30 °C and -20 °C some "small" grains are entirely surrounded by other "small" grains. At -10 °C there are lines of "small" grains in contact along the boundary between "large" grains. It is very difficult (and at -30 °C impossible) to have all of these "small" grains linked to large grains in the third dimension whilst maintaining a microstructure (e.g. in an orthogonal plane) that looks like the microstructures in these maps. This is the case at 20% strain. At lower strains the percentage of "small" 1-D segments that correspond to "small" 2-D grains is lower so the confidence with which we can define "small" grains is reduced.

The linear intercept analyses described in the previous paragraph also allow a crude assessment of grain oversampling—in other words, how likely are we to measure a large, branching grain more than once? In all samples >90% of 2-D grains along an arbitrary line are unique (that is, they are cut only once). Of course, with lines in multiple directions the percentage of unique grains might decrease. Using EBSD crystal orientation data, we can assess the likelihood of nearby grains in the 2-D map belonging to the same grain in 3-D (Monz et al., 2020). For every grain identified within a given EBSD map, we searched for all the nearby grains misoriented by less than a 10°, within a 1mm radius. These thresholds probably overestimate the number of grains connected in 3D. 1mm is close to double the size of the largest grain and 10° is more than twice the median and significantly larger than the higher quartile in mis2mean data (the misorientation angle between all pixels in a grain and the mean orientation of that grain) for all samples.

Full details are outlined in section S4 of the supplementary material and key outcomes are listed in Table 3. The percentage of "unique" grains (that only appear at the surface once) relative to all grains in a 2D map is higher than 68% at all temperatures and strains (Table S2, Table 3). The procedure outlined in the last paragraph allows us to estimate the number of "distinct" grains (where all 2-D grains attributed to the same 3-D grain are counted as one grain) in each map and from this, the number density (grain number per unit area) of "distinct" grains. The number density of "distinct" grains within all deformed samples at all temperatures is greater by a factor of 3 than that in the starting material: reaching values > 6 times the starting material at -10 °C and >11 times the starting material at lower temperatures (Fig. 11(c), Table 3). The number density of "distinct"

grains is generally higher at strains of ε ≥~12% than at strains of ε ≤~8% at all temperatures, and it is generally higher in samples deformed at -20 and -30 °C than samples deformed at -10 °C (Fig. 11(c), Table 3).

The analyses above provide some confidence that in all the experiments the number density of grains has increased relative to the starting material and increases with strain. If we couple this to the grain size statistics presented and the analysis of whether we are misidentifying small grains, the weight of evidence suggests that we have a real population of smaller grains. Our confidence in this statement increases with reducing temperature and increasing strain. Now we come back to the issue of how we distinguish "big" and "small" grains. Our scheme for separating "big" and "small" grains, using $D_{SMR}$ of the sample deformed to ~12% strain as the threshold, is not perfect, but it does provide a fast and repeatable way of looking at the possible differences in microstructures and CPO of smaller and larger grains. The grain size threshold chosen (peak, mean, median and SMR) to separate "big" and "small" grains has little impact on the CPOs of the "big" and "small" grain populations (see section S5 of the supplementary material).

### 3.3.3 Subgrain size

Subgrain size distributions (Fig. 4(e), 5(e) and 6(e)) are similar to the grain size distributions (Fig. 4(d), 5(d) and 6(d)), but the median and mean subgrain sizes are smaller than median and mean grain sizes (Table 3, Fig. 11(b)). In many cases, particularly at lower temperatures, the peak corresponds to the lower grain size resolution (cut off) indicating that we could be missing smaller subgrains. For this reason, the peak subgrain sizes are not useful and the median and mean subgrain sizes probably represent overestimates. The interquartile range (IQR) of subgrain sizes, with upper and lower limits bounded by higher quartile and lower quartile subgrain sizes, respectively, generally decreases with an increasing strain at all temperatures (Fig. 11(b)). IQR covers a wide range of subgrain size for samples deformed at -10 °C, but it narrows as the temperature decreases (Fig. 11(b)).

### 3.3.4 Misorientation

Misorientation angle distributions are presented for misorientations between 2° and 20° (Fig. 4(f), 5(f), 6(f)). Random-pair misorientation angle distributions show the misorientations expected for the measured CPO. It is important to identify differences between neighbour- and random-pair distributions, as these can be attributed to orientation inheritance, among other processes (Wheeler et al., 2001). Neighbour- and random-pair distributions at misorientation angles greater than 20° (not shown) are very similar in all samples, indicating that these are simply a function of the CPO. In all deformed samples there is a large peak at 2° in neighbour-pair data that is not present in random-pair data. The difference between neighbour-pair and random-pair frequency lessens as misorientation angle increases. The misorientation angle at which neighbour-pair frequency has reduced to be equal to the random-pair frequency increases with decreasing temperature. At -10°C, it is at 10° to 14° and does not change substantially with strain. At -20°C this angle is 10° to 14° at low strain but increases to around 18°-20° at 12% and 20% strain. At -30°C this angle is 16° to 18° at 3% strain, 18° to 20° at 5% strain and 20° at higher strain.

Neighbour-pair misorientation axes at misorientation angles 5°-10° show primary maxima lying in the basal plane (Fig. 4 (g), 5(g), 6(g)) for all deformed samples. Misorientation axes below 5° are omitted from these plots as the axes have relatively high angular errors (Prior, 1999). Misorientation axes from 2°-5° (not shown for all: an example is in Fig. 8(c)) also lie dominantly in the basal plane but have lower intensities, which we attributed to higher angular error. Grain boundary (>10°) misorientation axes for neighbouring grains are not strongly aligned. There is a very slight preference for misorientation axes lying in the basal plane (except PIL007) and this preference is slightly stronger at colder temperatures.

Figure 7 illustrates misorientation analyses of a typical "core-and-mantle" structure characterized by "small" grains (illustrated with thin boundaries) arranged along boundaries of "big" reference grains (ref1 to ref5 illustrated with thick boundaries) in sample PIL268 ( -30 °C, ~20% strain). The *c*-axes of reference grains are dispersed in a complex way, with the *c*-axes within an individual reference grain varying by up to ~20°. The complex dispersions include some data that lie along great circles and maybe some small circles. Great circle dispersions indicate rotation axes in the basal plane, consistent with neighbour-pair misorientation axes of low angle boundaries, which have a primary maximum parallel with poles to *m*-planes ([-1100]) (Fig. 7(c)). The *c*-axes of "small" grains are dispersed around the *c*-axes of "big" reference grains (Fig. 7(b)); the small grains occupy a much wider range of orientations than the large grains. Some of the small grains have *c*-axes within the single grain that are dispersed in a great circle smear, with up to ~5° of *c*-axis orientation variation. The distributions of misorientation axes between each of the reference grains and it's neighbouring small grains show no particular pattern apart from an absence of misorientation axes close to [0001]. These are all high-angle (>10°) misorientation so the axis errors will be small (Prior, 1999). The boundary misorientation axes between neighbouring "small" grains are distributed relatively uniformly (Fig. 7(d)), apart from an absence of data close to [0001].

## 3.4 Crystallographic preferred orientations

The contoured *c*-axes, *a*-axes and poles to *m*-planes pole figures are illustrated in Fig. 8-10. The *c*-axes figures are presented with (1) the compression axis vertical and (2) the compression axis perpendicular to the page. These two reference frames, which are commonly used by different communities, enable different elements of symmetry to be illustrated. At all temperatures CPO intensity increases with strain.

### 3.4.1 -10 °C series

The CPO of the sample (PIL176) deformed to ~3% strain at -10 °C is characterized by several weak maxima of *c*-axes with similar angles relative to the compression direction, and random distributions of *a*-axes and poles to *m*-planes. As the strain increases from ~5%, the CPO becomes clearer, with *c*-axes aligned in a cone (small circle). The cone is incomplete, with distinct maxima that are distributed along a small circle and individually elongated along the small circle trajectory. The *a*-axes and poles to *m*-planes align in a broad swath along the plane perpendicular to the compression axis and bound by the *c*-axis cone.

### 3.4.2 -20 °C series

The CPO of the sample (PIL183) deformed to ~3% strain at -20 °C is very weak. At ~5% strain (PIL182), the CPO is characterized by a blurred cone formed by several weak maxima of *c*-axes, and randomly distributed *a*-axes and poles to *m*-planes. As the strain increases from ~8% to ~12%, the CPO becomes clearer, with *c*-axes aligned in distinct clusters superposed on a blurred broad small circle cone. The *a*-axes and poles to *m*-planes align in weak a broad swath along the plane perpendicular to the compression axis and bound by the *c*-axis small cone. At ~20% strain, the *c*-axes align in two clusters that lie in a cone (small circle), and the *a*-axes and poles to *m*-planes align in broad swath along the plane perpendicular to the compression axis and bound by the *c*-axis small cone.

### 3.4.3 -30 °C series

The CPOs of the samples (PIL165, PIL162) deformed to ~3% and ~5% strain at -30 °C are very weak. As the strain increases from ~8% to ~12%, the *c*-axis CPO exhibits a pattern of a distinct narrow cone superposed on an overall broad cluster, the *a*-axes and poles to *m*-planes align in a broad swath along the plane perpendicular to the compression axis. At ~20% strain, the *c*-axes align in distinct clusters superposed on an overall broad cluster, and the *a*-axes and poles to *m*-planes align in a broad swath along the plane perpendicular to the compression axis and bound by the *c*-axis narrow cone.

### 3.4.4 CPOs of different grain size fractions

"Big" and "small" grains have similar patterns of *c*-axes (i.e. maxima in approximately the same places)—samples deformed to ~12% strain illustrate this (Fig. 12). The data are taken from smaller area maps (Table 2) with a 5 μm step size; the CPOs for all grains are comparable to data from larger areas using a 30 μm step size (compare Fig. 12a with Fig. 8-10), with CPOs from the 30 μm maps being slightly weaker. At -10 °C, the CPO intensity of "small" grains is slightly lower than "big" grains (Fig. 12(b-c)). This contrast becomes strengthened as the temperature decreases. At -30 °C the CPO intensity of "small" grains is much lower than "big" grains (Fig. 12(b-c)). CPO intensity is not significantly affected by the number of grains used to calculate M-index—we verified this by calculating M-index for a subpopulation of small grains, of the same size as the "big" grain population (Fig. 12(d)).

To show how CPO strength differs for "big" and "small" grains for the whole data set we plot the M-indices for the grain size categories against strain (Fig. 11(d)). For all the deformed samples, the M-indices of "big" grains have the same pattern with strain as the complete data set (all grains). The "small" grains generally have lower M-indices at strains of $\varepsilon \geq$ ~5%. The grain size threshold ($\overline{D}$, $D_{SMR}$, $D_{median}$ and $D_{peak}$) chosen to separate "big" and "small" grains has a minor impact on CPO, with no significant change in CPO pattern or intensity (see section S5 of the supplementary material for the test).

### 3.4.5 The opening-angle of the *c*-axis cone

To increase our understanding of the processes that might control the *c*-axes cone opening-angle $\theta$, we plotted data from this study and previous studies in a diagram of $\theta$ as a function of strain with data subdivided with different temperatures and strain rates (Table 4 and Fig. 13). For data from the literature, we digitised the *c*-axis orientations from published stereonets (Jacka and Maccagnan, 1984; Jacka and Li, 2000) and calculated $\theta$ using the same method described in Section Two (Method). For data from Montagnat and others (2015) and Craw and others (2018), we measured the values of $\theta$ directly from contoured *c*-axis CPO figures. For data from Vaughan and others (2017), we calculated the values of $\theta$ from raw EBSD data. The values of $\theta$ from Hooke and Hudleston (1981), Piazolo and others (2013), Qi and others (2017) and Wilson and others (2020) are taken directly from these papers. The experiments reported by Piazolo and others (2013) were conducted on D₂O ice at -7 °C, which is a direct analogue for deforming H₂O ice at −10 °C (Wilson et al., 2019). These angles were analysed using methods similar to ours. In order to make a direct comparison with the data reported from this study and Qi and others (2017), we converted the reported axial engineering strain ($e$) and strain rate ($\dot{e}$) (Piazolo et al., 2013; Montagnat et al., 2015; Vaughan et al., 2017) to true axial strain ($\varepsilon$) and strain rate ($\dot{\varepsilon}$) using the equations:

$$\varepsilon = -\ln(1-e) \tag{4}$$

$$\dot{\varepsilon} = \frac{\dot{e}}{1-e} \tag{5}$$

Equation (6) and (7) were used to forward model axial engineering strain ($e$) and strain rate ($\dot{e}$) from octahedral shear strain ($\gamma$) and strain rate ($\dot{\gamma}$) (Jacka and Maccagnan, 1984; Jacka and Li, 2000).

$$\gamma = \frac{\sqrt{2}}{3}\left(e + \frac{1}{\sqrt{1-e}} - 1\right) \tag{6}$$

$$\dot{\gamma} = \frac{\sqrt{2}}{3}\left(\frac{1}{2(1-e)^{\frac{3}{2}}} + 1\right)\dot{e} \tag{7}$$

After that, the axial engineering strain ($e$) and strain rate ($\dot{e}$) were converted to true axial strain ($\varepsilon$) and strain rate ($\dot{\varepsilon}$) using Eq. (4) and Eq. (5).

For natural ice samples (top of the south dome, Barnes Ice Cap, Baffin Island) from Hooke and Hudleston (1981), Eq. (8) was used to calculate axial engineering strain ($e$) from natural octahedral unit shear strain ($\bar{\gamma}_{oc}$). Values of $\bar{\gamma}_{oc}$ were taken from Fig. 4 of Hooke and Hudleston (1981) based on the assumption that ice was deformed under uniaxial compression. After that, the axial engineering strains ($e$) were converted to true axial strains ($\varepsilon$) using Eq. (4). Hooke and Hudleston (1981) assumed their natural ice samples were deformed under a constant vertical strain rate, $\dot{e}$, of $5.71 \times 10^{-11} s^{-1}$, which converted to true axial strain rate ($\dot{\varepsilon}$) using Eq. (5). The derivations of Eq. (4-8) are shown in section S2 of the supplementary material.

$$\bar{\gamma}_{oc} = \frac{2\sqrt{2}}{3}\left(e + \frac{1}{\sqrt{1-e}} - 1\right) \tag{8}$$

To our knowledge, Fig. 13 contains data from all published 3-D uniaxial compression ice experiments and deformed natural ice that present *c*-axis CPOs as a function of strain. 2-D experiments, involving deformation on a microscope stage (e.g. Peternell et al., 2014; Peternell and Wilson, 2016) are excluded as these have different kinematics. There are numerous other high temperature and low strain rate axial compression experiments to strains of ~10% to 30% where *c*-axis cones have

opening-angles of ~35 degrees (e.g., Wilson and Russell-Head, 1982; Gao and Jacka, 1987; Treverrow et al., 2012; Wilson et al., 2019). These data are consistent with the pattern shown in Fig. 13 but are not part of a strain series and are not added to the diagram to maintain clarity. There are comparatively few CPOs from samples at low temperatures (< -15 °C) so we have included all published data from experiments at < -15 °C irrespective of whether these are part of a strain series. The values of $\theta$ are scattered between 0° and 42° for all experiments. Experiments to low strains have random CPOs where a cone angle

cannot be defined, and these data are not shown on Fig. 13. For experimental data, the evolution pattern of $\theta$ as a function of strain at temperatures warmer than -15 °C show $\theta$ decreases with increasing strain up to ~20% true axial strain. The only two data points of $\theta$ from samples deformed to the strain of ~50% are at 30°. There is little difference as a function of temperature at $\geq$ -15 °C. For natural ice deformed under temperatures of -4 to -6 °C, $\theta$ generally decreases with increasing strain for both "coarse" ice (>0.15 cm2) and "fine" ice (<0.1 cm2).

Samples deformed at temperatures colder than -20 °C have lower $\theta$ values compared with samples deformed at warmer temperatures at similar strains. At -30 °C, the opening-angle of the *c*-axis cone decreases to ~0° at strains of ~20%. The strain corresponding to the formation of a clear *c*-axis cone (non-random CPO) increases with decreasing temperature.

## 4 Discussion

### 4.1 Deformation mechanisms

**4.1.1 Inferences from mechanical evolution**

All stress-strain curves (Fig. 3) show stress rising to the peak stress and then relaxing, with the rate of stress drop decreasing with strain. This pattern matches published constant-displacement-rate experiments (Mellor and Cole, 1982; Durham et al., 1983; Durham et al., 1992; Piazolo et al., 2013; Vaughan et al., 2017; Qi et al., 2017; Craw et al., 2018; Qi et al., 2019), and has an approximate inverse relationship (Mellor and Cole, 1982, 1983; Weertman, 1983) to constant-load experiments (Budd

and Jacka, 1989; Jacka and Li, 2000; Treverrow et al, 2012; Wilson and Peternell, 2012) where strain rate first decreases to a minimum and then increases to approach a near-constant strain rate.

Stress-strain curves of all experimental runs show a smooth and continuous increase of stress as a function of strain before reaching the peak. Approximately linear portions of the stress-strain data prior to peak have been termed quasi-elastic (Kirby et al., 1987). Slopes of ~1GPa are significantly below the published value of Young's modulus (~9GPa: Gammon et al, 1983)

and indicate that there is significant dissipative deformation here. This likely includes anelastic deformation related to intergranular stress redistribution used to explain primary creep in constant load experiments (Duval et al, 1983; Castelnau et

al., 2008). The curvature of the stress-strain line at the start of each experiment may relate to initial porosity loss as suggested by rapid increases in ultrasonic p-wave velocity in comparable experiments by Vaughan and others (2017).

As our experiments are all at the same approximate strain rate, we cannot calculate the stress dependency of strain rate (the stress exponent, n). Qi and others (2017) calculate a peak stress n value of 3 and flow stress n value of 3.9 for comparable experiments (including PIL007 used here) at -10 °C. Craw and others (2018) calculate a peak stress n value of 4.1 for comparable experiments at -30 °C. Our experiments show higher peak and final stress values at colder temperatures than at warmer temperatures. This phenomenon is well known, and the temperature dependence of the creep rate is commonly parameterised using an Arrhenius relationship with an activation energy (Homer and Glen, 1978; Durham et al., 1983, 2010; Budd and Jacka, 1989; Cuffey and Paterson, 2010; Scapozza and Bartelt, 2003). Our peak and final stress data can be used to calculate the activation energy by assumption of a value of stress exponent, n (see section S6 of the supplementary material for the calculation). A best fit to all data (-10, -20 and -30 °C) give activation energy of 98 kJ/mol and 103 kJ/mol from peak and final stress data assuming n=3 and 131 kJ/mol and 138 kJ/mol from peak and flow stress data assuming n=4. These numbers are consistent with published values (64-250 kJ/mol) at relatively high temperature (Glen, 1955; Goldsby, 2001; Budd and Jacka, 1989; Cuffey and Paterson, 2010; Durham et al., 2010; Kuiper et al., 2020a, 2020b).

### 4.1.2 Inferences from microstructure

#### 4.1.2.1. Nucleation

The number density (number of grains per unit area) of "distinct" grains (counting 2-D grains attributed to the same 3-D grain as one: section S4 of supplementary material) increases by more than a factor of 3 over that of the starting material in all deformed samples at all temperatures (Table 3). We can be reasonably confident that the number of grains in the samples has increased as a function of deformation, which requires a process of nucleation to create new grains. For all the deformed samples, the grain size distributions are characterised by peaks at finer grain sizes, and a smaller mean/median grain size compared with the undeformed sample (Fig. 2, 4-6, Table 3). The smallest grains in the deformed samples were not present in the starting material. These observations suggest that nucleation generates the grains with smaller sizes. The number of grains per unit area (accounting for multiple occurrences of the same 3-D grain) generally increases and all measures of 2-D grain size decrease with strain (Table 3), at all temperatures, suggesting that nucleation operates continuously as part of the recrystallisation process throughout the deformation.

#### 4.1.2.2. Dislocation activity, recovery, subgrain rotation and subgrain rotation recrystallisation

Microstructure maps show subgrain boundaries in all deformed ice samples (Fig. 4(a-c), 5(a-c) and 6(a-c)). The subgrain boundary geometry is comparable with other experimentally or naturally deformed rock and metal samples, e.g. quartz (Cross et al., 2017a; Killian and Heilbronner, 2017), olivine (Hansen et al., 2012), Magnox alloy (Wheeler, 2009) and zircon (MacDonald et al., 2013). The misorientation axes for subgrain boundaries are generally rotations around rational crystallographic axes, particularly directions in the basal plane, suggesting that the boundaries may represent arrays of

dislocations (Humphreys and Hatherley, 2008; Shigematsu et al., 2006). There is much higher frequency of low angle (Particularly < 10°) neighbour-pair misorientations than are expected from the CPO (as shown by the random-pair misorientation angles). The subgrain boundaries and the pattern of misorientation angles are commonly interpreted as the result of dynamic recovery of dislocations generated during deformation and subsequent subgrain rotation related to ongoing

recovery (Guillope and Poirier, 1979; Trimby et al., 1998; Fliervoet et al., 1999; Wheeler et al., 2001) and has been observed from ice deformation experiments previously and interpreted in this way (e.g. Montagnat et al., 2015; Qi et al., 2017; Seidemann at al., 2020). The misorientation angle at which neighbour-pair frequency has reduced to be equal to the random-pair frequency increases with decreasing temperature (Fig. 4 (f), 5(f), 6(f)). This observation suggest intragranular distortion is more significant at lower temperatures.

Subgrain rotation is a process that involves an increase in the misorientation across a subgrain boundary forming via the progressive addition of dislocations (White., 1979; Lallemant, 1985). New grains will form as the misorientation across the subgrain boundary becomes large enough, with the subgrain boundary eventually dividing its parent grain (Poirier and Nicolas, 1975; Guillope and Poirier, 1979; Urai et al., 1986; Halfpenny et al., 2006; Gomez-Rivas et al., 2017). This process is known as subgrain rotation recrystallization (Hirth and Tullis, 1992; Stipp et al., 2002; Passchier and Trouw, 2005). When subgrain

rotation recrystallization is responsible for nucleation, the recrystallized "daughter" grains should be initially of a similar size to the internal subgrain size of the "parent" grain (Urai et al., 1986). At all temperatures and strains the mean, median, lower quartile and upper quartile values of subgrain sizes are smaller than that of grain sizes (Table 3). This indicates that the subgrain rotation recrystallization could be the nucleation mechanism that generates the "small" grain population. Previous studies on deformed metals and quartzites describe the structure of smaller grains encircling larger grains as "core-and-mantle" structure

(Gifkins, 1976; White, 1976). The production of smaller grains that form the "mantle" region was considered as a result of continual rotation of subgrains to develop small strain-free grains (White, 1976; Urai et al., 1986; Jacka and Li, 2000). The network of smaller grains that encircle bigger grains at strains higher than 12% at -20 and -30 °C is consistent with the operation of subgrain rotation recrystallization. The network of finer grains encircling larger grains has been observed in deformed metals, and it is named the "necklace structure" in the material science literature (e.g. Ponge and Gottstein, 1998; Jafari and

Najafizadeh, 2009; Eleti et al., 2020).

For all deformed samples, neighbour-pair misorientation axes at misorientation angles of 5°-10° show primary maxima lying in the basal plane (Fig. 4(g), 5(g), 6(g)). Similarly, misorientation axes of 5°-10° boundaries within the large reference grains of the "core-and-mantle" structure have primary maxima lying in the basal plane (Fig. 7(c)). These observations suggest low-angle boundaries (5°-10°) are crystallographically controlled, consistent with them being subgrain boundaries that comprise

arrays of dislocations.

The general understanding of the subgrain rotation recrystallisation process (Poirier, 1972; Poirier and Nicolas, 1975), is that a new "nucleus" forms when all of the enclosing subgrain boundaries, comprising arrays of dislocations, are transformed to grain boundaries, that have no formally defined structure (Rollett et al., 2017). In simplified terms there is a limit to the misorientation that can be sustained by an array of dislocations (related to limits of the Read-Shockley equation: Read and

Shockley, 1950; Rollett et al., 2017) so at some point continued addition of dislocations to a boundary prompts the change in boundary structure. There is no mechanism understood by which the misorientation of a subgrain boundary changes when it transforms to a grain boundary. Thus, high-angle grain boundaries of nuclei generated by subgrain rotation recrystallization should have the same orientation as the subgrain that forms the nucleus and should also be crystallographically controlled (Rollett et al., 2017), if additional processes are not activated after their production.

In all deformed samples, neighbour-pair misorientation axes at misorientation angles larger than 10° have a distribution close to random (Fig. 4(g), 5(g), 6(g)), suggesting high-angle grain boundaries lack a crystallographic control. The "core-and-mantle" structures observed at lower temperatures (-20 °C and -30 °C) have misorientations axes of boundaries between small grains and their large or small neighbours that do not lie in the basal plane and have no preferred nor rational crystallographic orientation (Fig. 7(d)). High-angle grain boundaries of "small" grains lack a crystallographic control within deformed samples. Subgrain rotation recrystallisation can be considered a nucleation mechanism, but alone it cannot explain the misorientation axes of high angle boundaries. Either a different nucleation mechanism is needed or an additional process is needed to change the orientation of grains during or after nucleation. This discussion continues in the context of the CPO data in section 4.1.4.

All measures of grain and subgrain sizes are smaller at lower temperatures (Table 3, Fig. 11(b)). This is likely a consequence of the higher stresses of the lower temperature experiments resulting in smaller subgrain and recrystallised grain sizes through a piezometer or similar relationship (Derby, 1991; Austin and Evans, 2007; Lopez Sanchez and Llana Funez, 2015; Cross et al., 2017a). Jacka and Li (1994) show an inverse relationship between ice grain size and stress from deformed ice samples that reach tertiary creep.

### 4.1.2.3. Grain boundary migration

Lobate grain boundaries are commonly interpreted as the result of strain-induced grain boundary migration (GBM) (Urai et al., 1986; Jessell, 1986; Duval and Castelnau., 1995). Samples deformed at -10 and -20 °C show more grains with lobate boundaries at higher strains (>~3%), suggesting more widespread strain-induced GBM with an increasing strain. The proportion of repeated (i.e. interconnected and highly lobate) grains is generally higher in the higher-temperature experiments (Table 2, section S4 of the supplementary material). This observation suggests that GBM is also more widespread at higher temperatures. It is worth noticing that data presented in this study is not sufficient enough to quantify the rate of GBM at high and low temperatures. Because the rate of GBM depends on two parameters; the boundary mobility, which is a function of temperature, and the driving force, conventionally the dislocation density difference (Humphreys and Hatherley, 2008). The dislocation density difference is likely to be controlled by stress (Bailey and Hirsch, 1960; Ajaja, 1991). Under a lower temperature, the grain boundary mobility is likely to reduce. However, the differential stress for a given strain rate increases (Fig. 3). Consequently, the dislocation density difference is likely to be higher at a lower temperature.

### 4.1.3 CPO development

The CPO intensity and opening-angle of the *c*-axis CPO decrease as the temperature drops. Previous studies suggest the CPO development is mainly controlled by the deformation and recrystallization mechanisms (Alley, 1992; Qi et al., 2017). Fig. 14 explains how key processes (Fig. 14(b)) involved in the deformation and recrystallization mechanisms (Fig. 14(a)) may affect the CPO development as a function of strain and temperature (Fig. 14(c)). Many deformed samples exhibit an incompleteness of *c*-axes cone (lack of cylindrical symmetry) (Fig. 8-10). The incompleteness of *c*-axes cone is more severe for 5 µm EBSD maps collected from a much smaller area than 30 µm EBSD maps (Fig. 12). These phenomena are common to all ice CPOs from measurements on a single sample planes (by EBSD or optical methods: see any of the papers cited), but are not so apparent in neutron diffraction data (Piazolo et al., 2013; Wilson et al., 2019), that sample a larger volume, suggesting that a single plane through a deformed sample does not generally contain sufficient grains for a fully representative CPO.

Our -10 °C series CPO data show a monotonic increase in CPO intensity as indicated by M-index, and a clearer cone-shaped pattern of the *c*-axes with increasing strain. Similar observations were made in previous ice deformation experiments (e.g. Jacka and Maccagnan, 1984; Piazolo et al., 2013; Montagnat et al., 2015; Vaughan et al., 2017; Qi et al., 2017). Cone-shaped *c*-axes CPOs have been related to strain-induced GBM favouring the growth of grains with easy slip orientations (high Schmid Factors) (Duval and Castelnau., 1995; Little et al., 2015; Vaughan et al., 2017; Qi et al., 2017). Linked to this is the idea that grains with hard slip orientations should have greater internal distortions (Duval and Castelnau., 1995; Bestmann and Prior 2003), and therefore store higher internal strain energy. If this is correct then hard slip grains are likely to be consumed by grains with easy slip orientations through GBM (Duval and Castelnau., 1995; Castelnau et al., 1996; Bestmann and Prior, 2003; Piazolo et al., 2006; Killian et al., 2011; Qi et al., 2017; Xia et al., 2018). However, we have to re-evaluate the detail of this idea, as recent studies on deformed ice samples show there is no systematic relation between orientation and strain localisation at low strain (Grennerat et al. 2012). Furthermore, studies of high-strain shear samples find no clear difference in the geometrically necessary dislocation density within the two maxima that develop in simple shear (Journaux et al. 2019). An alternative, and as yet incomplete, explanation from Kamb (1959) relates recrystallisation directly to the elastic anisotropy of crystals and through this to the orientation of the stress field. At this stage the observation that ice CPOs developed at relatively high temperature and particularly at low strain correspond to high Schmid factor orientations remains robust. The underlying mechanisms will need continual review as we collect new data.

At temperatures greater than -10° the cone opening angle ($\theta$) from experiments decreases from 42° at ~3% strain to ~ 30° at 20-50% strain. Hooke and Huddleston's (1981) data from Barnes ice cap suggest it may reduce further to ~18° at ~143% strain (Table 4, Fig. 13). The cone opening angle does not stabilise at the easiest slip orientation of 45°, suggesting that GBM alone cannot be the mechanism that controls the CPO development. Previous studies suggest CPO evolves through the parallel operation of rotation and selective growth (e.g. Kamb, 1972; Qi et al., 2017). The narrowing of cone-shaped *c*-axis CPO has been explained by an activation of grain rotation (Jacka and Li., 2000; Qi et al., 2017). Jacka and Maccagnan (1984) show that the *c*-axis cone narrows in compression and opens in extension, consistent with the expected kinematics of grain rotation.

Neighbour-pair misorientation axes at misorientation angles of 5°-10° show primary maxima lying in the basal plane (Fig. 4(g), 5(g), 6(g)) at both low and high strains. Therefore, we infer the decreasing of opening-angle $\theta$ as a function of strain is likely to result from more active grain rotation driven by intracrystalline glide on the basal plane. For all deformed samples in this study, there is a large peak at 2° in neighbour-pair misorientation angle distribution, which is not present in the random-

or neighbour-pair data of the starting material (Fig. 4(f), 5(f), 6(f)). Moreover, neighbour-pair misorientation angles show much higher frequencies between 2° and 10° than random-pair data. These observations suggest recovery and subgrain rotation operated in parallel with strain-induced GBM. The dislocation activity required to generate subgrain structures and to provide the strain energy driving force for strain-induced GBM is likely the primary control on grain rotation (Duval and Castelnau, 1995; Llorens et al., 2016).

The opening-angle $\theta$ of the $c$-axes cone as well as the CPO intensity decrease with decreasing temperature (Table 2, 4; Fig. 8-10, 11(c), 13). Earlier studies have inferred that the selective growth of the grains oriented for easy slip orientations becomes less active due to the reduction of GBM activity at lower temperatures (Qi et al., 2017, 2019). Lower temperatures, for constant displacement rate experiments, correspond to higher stresses. Previous studies in deformed metals suggest a higher stress is likely a cause of higher dislocation densities (Bailey and Hirsch, 1960; Ajaja, 1991), that in turn will require kinematically

more lattice rotation. The misorientation angle at which neighbour-pair frequency reduces to be equal to the random-pair frequency increases with decreasing temperature (Fig. 4(f), 5(f), 6(f)). Moreover, neighbour-pair misorientation axes at misorientation angles of 5°-10° show primary maxima lying in the basal plane (Fig. 4 (g), 5(g), 6(g)) for all deformed samples. These observations support the hypothesis that grain rotation becomes more prominent at lower temperatures (Jacka and Li., 2000), and it is dominantly driven by intracrystalline glide on the basal plane. More active grain rotation can lead to a closure

of $c$-axis cone at lower temperatures: maxima parallel to compression are characteristic of strains $\geq 20\%$ at temperatures colder than -30 °C (Craw et al., 2018; Prior et al., 2015).

### 4.1.4 CPO development: differences related to grain size

The CPO intensity (as indicated by M-index) of "small" grains is generally lower than "big" grains, and this contrast strengthens with decreasing temperature (Fig. 11(d)). At ~12% strain, the CPO pattern of "big" grains is clearer than "small"

grains, at all temperatures (Fig. 12). These observations suggest a mechanism that weakens the CPO development may be associated with the "small" grains. Microstructural analyses suggest subgrain rotation and subgrain rotation recrystallization are dominant processes responsible for the production of "small" grains within samples deformed at both high and low temperatures (see section 4.1.2.2). Subgrain rotation and subgrain rotation recrystallization will disperse orientations away from the parent grain orientation and small grains, if representative of nuclei formed by subgrain rotation recrystallization will

represent the most extreme dispersion from the parent orientation. However, subgrain rotation recrystallization alone should produce nuclei with crystallographically controlled grain boundaries linked to the parent grain orientation. As this is not the case (see section 4.1.2.2) it is difficult to explain the weakening of the CPO by subgrain rotation recrystallization alone; additional processes must be responsible for a weaker CPO within small grains. Data from our experiments suggest such

hypothesis works better for experiments at -20 and -30 °C. At -10 °C the contrast of CPO intensity between "big" and "small" grains becomes less clear, and the "core-and-mantle" structure is absent. One way to explain these data is that the modification of grain boundary topology via grain boundary migration (GBM) could be more widespread at a warmer temperature, which could obscure evidence of the additional process responsible for a weaker CPO in smaller grains. The interaction of GBS with

dislocation processes and GBM is developed a little in the ice literature (Hondoh and Higashi, 1983; Liu et al, 1993; 1995; Duval, 1985; Faria et al., 2014) and experiments that show offsets of grain boundaries whilst GBM occurs (Drury and Humphreys 1988; Ree, 1994) suggest that GBS may not leave a clear microstructural signal when GBS and GBM coexist. There are two candidate processes that may explain weakening of CPO in smaller grains and the lack of rational crystallographic misorientation axes of small grains relative to each other or to larger grains.

1.  Previous rock deformation studies reported small recrystallized grains having CPOs that are randomly dispersed equivalents of the stronger parent grain CPOs (Jiang et al., 2000; Bestmann and Prior, 2003; Storey and Prior, 2005; Warren and Hirth, 2006). These observations are commonly interpreted as the result of an increase in the contribution of grain boundary sliding (GBS) in fine grains right after their formation via subgrain rotation recrystallization. Craw and others (2018) reported similar interpretations in uniaxially deformed Antarctic ice, and the reduction of CPO

intensity in grains with finer sizes was attributed to GBS. Bestmann and Prior (2003) suggest a randomization of boundary misorientation axes among "small" grains can result from sliding along boundaries of newly formed grains. GBS can accompany grain shape change by dislocation or diffusion processes (Raj and Ashby, 1971; Crossman and Ashby, 1975; Langdon 1994, 2006, 2009). Recently, Cross and others (2017b) found evidence for specific orientations (i.e., those lying in the plane of maximum vorticity) being randomized by GBS. GBS is inherently grain

size sensitive (Raj and Ashby, 1971; Gifkins 1976; Langdon, 1994; Warren and Hirth, 2006) and interpretation of GBS processes in fine polycrystalline ice was initially made by identification of grain size sensitivity of mechanical data for the deformation of fine-grained ice (Goldsby and Kohlstedt, 1997, 2001).

        2.  Microstructural studies of metals deformed under high homologous temperatures suggest a possibility of nucleation of grains with random orientations at the tip of irregular boundaries of "parent" grains (Hasegawa and Fukutomi,

2002; Hasegawa et al., 2003). Such hypothesis is similar to "spontaneous" nucleation (Duval et al., 2012), a process considered driven by the relaxation of the dislocation-related internal stress field that may produce nuclei with orientations not related to their corresponding parent grains. Adoption of this interpretation would require that small grains are not generated by subgrain rotation recrystallisation. Falus and others (2011) suggests bulging, which allows for higher misorientations between parent and recrystallized grains, could explain a higher dispersion of CPOs.

The two hypotheses—GBS and nucleation of grains with random orientations—can explain a weakening of CPO in "small" grains and these two ideas are not mutually exclusive. Further work is needed to test both hypotheses. Most critical are experiments where nuclei can be observed whilst they are very small and subsequent misorientations can be documented, as might be possible with 3-D microscopy methods (Lauridsen et al., 2003; Poulson et al., 2004), and experiments where fiducial

markers are used to confirm the physical existence of offsets on grain boundaries (Schmid et al, 1977; Spiers, 1979 ; Beeré, 1978; Drury and Humphreys, 1988; Ree, 1994; Maruyama and Hiraga, 2017; Eleti et al., 2020).

## 4.2 Future work: implications for enhancement (weakening)

The mechanical weakening, i.e. stress drop after peak in constant strain rate experiments and strain rate enhancement from
secondary to tertiary creep in constant load experiments, has been associated with: (1) the softening owing to the reduction of stored strain energy by dynamic recrystallisation processes such as nucleation and grain boundary migration (Duval, 1979; Weertman, 1983; Derby and Ashby, 1987; Humphreys and Hatherley, 2008; Rollett et al 2017), (2) increased contribution of grain size sensitive deformation mechanisms due to grain size reduction resulting from dynamic recrystallization (De Bresser et al., 2001), and (3) development of strong CPO in viscously anisotropic materials (Durham and Goetze, 1977; Hansen et al.,
2012) such as ice. The microstructural data as discussed in section 4.1 will enable us to comment on the potential contribution of different mechanisms to the weakening in deformed polycrystalline ice.

All experiments show weakening after peak stress (Fig. 3). The percentage of stress drop from peak to flow stress is similar between high and low temperatures. Weakening is classically observed during dynamic recrystallization; the mechanical evolution from peak to flow stress in constant displacement rate experiments or from secondary to tertiary creep stage in
constant load experiments was attributed to a balance between processes favouring strain hardening (e.g. accumulation of stored strain energy through dislocation) and weakening (e.g. reduction of stored strain energy through dynamic recrystallisation) (Duval, 1979; Weertman, 1983; Derby and Ashby, 1987; Humphreys and Hatherley, 2008). In this study, mean and median ice grain size reduces with strain at all temperatures (Table 3, Fig. 11(a)). Grain size is commonly reduced during rock deformation in the laboratory (e.g. Pieri et al., 2001; Hansen et al., 2012) and in nature (Trimby et al., 1998;
Bestmann and Prior, 2003). At smaller grain sizes the strain-rate contribution of grain size sensitive (GSS) mechanisms increases or the stress required to drive a given strain rate contribution of GSS decreases. For this reason, grain size reduction has been proposed as a weakening mechanism (Rutter and Brodie, 1988; De Bresser et al 2001; Kilian et al., 2011; Campbell and Menegon, 2019). On the other hand, many published papers on ice sheet mechanics imply that enhancement (weakening) is caused by anisotropy development and there are analytical numerical models that seek to quantify this relationship (Azuma,
1995; Morland and Staroszczyk, 2009; Placidi et al., 2010). At -10 °C, the CPO development includes many grains with basal plane orientations that would facilitate further axial shortening and it is intuitive that CPO development could provide a cause for the weakening. However, at -30 °C the CPO developed at high strain is a narrow cone or cluster with many basal planes sub-perpendicular to compression. Therefore, further studies are required to quantify the contribution of candidate weakening (enhancement) mechanisms including (1) dynamic recrystallisation processes such as nucleation and GBM, (2) grain size
reduction and the resulting contribution of GBS, (3) CPO development to the mechanical weakening observed during ice deformation.

## 5 Conclusions

1. We deformed isotropic polycrystalline pure water ice to successive strains (~3%, 5%, 8%, 12% and 20%) under a constant displacement rate (strain rate $\sim 1.0 \times 10^{-5} s^{-1}$) at -10, -20 and -30 °C. For all deformed samples, stress first rises to a peak at ~1-4% strain and then drops to lower stresses at higher strains. Samples deformed at colder temperatures show higher peak and final stresses, as expected for the temperature dependency of creep. Microstructural and CPO analyses were conducted on deformed ice samples using cryo-EBSD.

2. All deformed samples develop distinct subgrain boundaries and show a peak at 2°-3° in the neighbour-pair misorientation angle distribution. Mean/median subgrain size is smaller than mean/median grain size. These observations suggest that dislocation glide and associated recovery and subgrain rotation were active in all deformed samples. Neighbour-pair low-angle (5°-10°) misorientation axes show primary maxima lying in the basal plane, for all samples, suggesting that basal glide dominated intragranular deformation processes. Subgrain boundary misorientation distributions extend to higher misorientation angles with strain and with decreasing temperature, suggesting that subgrain rotation develops progressively and is more effective at lower temperatures.

3. All deformed samples have skewed grain size distributions with a strong peak at small (<100 μm) sizes and a tail to larger sizes. The grain size peak is smaller than the grain size of the starting material (~297 μm) and a stereological analysis suggests that many of the small grains, measured in 2-D, are also small in 3-D. The number density of "distinct" grains (counting 2-D grains that are out of the analysis plane to the same 3-D grain) is more than 3 times that in the starting material for all deformed samples and the number density increases with strain. These data indicate that nucleation is involved in dynamic recrystallization. "Core-and-mantle" structures (small grains surrounding larger grains) are observed at high strains and are clearest at -20 and -30 °C, suggesting that subgrain rotation recrystallization has occurred and is more important at lower temperatures. Lobate grain boundaries suggest that strain-induced grain boundary migration has occurred in all samples.

4. Many of the deformed samples have CPOs defined by open cones (small circles) of $c$-axes. The cone opening-angle decreases with strain. The CPO intensity and $c$-axis opening-angle both decrease as the temperature drops from -10 to -30 °C. At -30 °C and 20% strain the $c$-axes define a cluster with maximum parallel to compression, rather than an open cone. We interpret that the open $c$-axis cone develops because strain-induced GBM favours the growth of grains in easy slip orientations. The closure of the $c$-axes cone with strain is interpreted primarily as the result of grain rotation related to intragranular dislocation glide on the basal plane. We infer that grain rotation becomes more prominent at lower temperatures, whilst GBM is more widespread at higher temperatures.

5. Small grains have a weaker CPO than large grains. This distinction is slight at -10 °C, but becomes much clearer at lower temperatures. Neighbour-pair high-angle ($\geq 10°$) misorientation axes, corresponding to grain boundaries are not strongly aligned in the basal plane, nor with any other crystal direction. An additional process is needed to explain these

observations. We identify two candidate processes; (1) grain boundary sliding causing rotation of grains without crystallographic control on the rotation axes, and (2) nucleation of grains with random orientations.

*Author contributions.* DJP, DLG and SF designed the research. DJP, SF, TH, AJC and CQ performed experiments. SF, DJP and MN collected the cryo-EBSD data. SF, DJP and JW analysed data. SF and DJP wrote the draft. All authors edited the paper.

*Acknowledgements.* We are thankful to Pat Langhorne for providing the cold room facility at University of Otago. This work was supported by a NASA fund (NNX15AM69G) to DLG and two Marsden Funds of the Royal Society of New Zealand (UOO1116 and UOO052) to DJP. SF was supported by the University of Otago doctoral scholarship, the Antarctica New Zealand doctoral scholarship and the University of Otago PERT (Polar Environment Research Theme) seed funding. We thank the journal reviewers Maurine Montagnat and Olivier Castelnau for very helpful reviews of the paper and Roger LeB. Hooke and Christopher J. L. Wilson for helpful additional comments.

*Competing interests.* The authors declare that they have no conflict of interest.

*Data availability.* Data available on request from the authors.

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

**Table 1** Summary of experiments

| Sample No. | Temperature | Initial length | True axial strain | Peak stress (corrected) | Strain rate at peak stress | True axial strain at peak stress | Final stress (corrected) | Final strain rate |
|---|---|---|---|---|---|---|---|---|
| | $T$ (°C) | $L_0$ (mm) | ($\varepsilon$) | $\sigma_p$ (MPa) | $\dot{\varepsilon}_p$ (s-1) | $\varepsilon_p$ | $\sigma_f$ (MPa) | $\dot{\varepsilon}_f$ (s-1) |
| PIL176 | -10 | 30.480 | 0.03 | 1.78 | $1.03 \times 10^{-5}$ | 0.02 | 1.70 | $1.04 \times 10^{-5}$ |
| PIL163 | -10 | 48.768 | 0.05 | 2.92 | $1.03 \times 10^{-5}$ | 0.01 | 2.42 | $1.06 \times 10^{-5}$ |
| PIL178 | -10 | 39.624 | 0.08 | 2.54 | $1.11 \times 10^{-5}$ | 0.02 | 1.97 | $1.19 \times 10^{-5}$ |
| PIL177 | -10 | 40.640 | 0.12 | 2.85 | $1.11 \times 10^{-5}$ | 0.03 | 1.90 | $1.21 \times 10^{-5}$ |
| [1]PIL007 | -10 | 63.754 | 0.19 | 2.13 | $1.03 \times 10^{-5}$ | 0.02 | 1.33 | $1.22 \times 10^{-5}$ |
| PIL254 | -20 | 39.624 | 0.03 | 4.33 | $1.05 \times 10^{-5}$ | 0.02 | 4.25 | $1.06 \times 10^{-5}$ |
| PIL182 | -20 | 46.990 | 0.04 | 4.88 | $8.09 \times 10^{-6}$ | 0.02 | 4.44 | $8.94 \times 10^{-6}$ |
| PIL184 | -20 | 31.242 | 0.08 | 3.64 | $1.13 \times 10^{-5}$ | 0.04 | 3.24 | $1.17 \times 10^{-5}$ |
| PIL185 | -20 | 41.656 | 0.12 | 4.69 | $1.09 \times 10^{-5}$ | 0.03 | 3.68 | $1.19 \times 10^{-5}$ |
| PIL255 | -20 | 49.530 | 0.20 | 4.66 | $1.10 \times 10^{-5}$ | 0.03 | 2.93 | $1.28 \times 10^{-5}$ |
| PIL165 | -30 | 37.846 | 0.03 | 8.24 | $1.08 \times 10^{-5}$ | 0.03 | 8.15 | $1.09 \times 10^{-5}$ |
| PIL162 | -30 | 50.546 | 0.05 | 8.71 | $1.07 \times 10^{-5}$ | 0.03 | 7.87 | $1.10 \times 10^{-5}$ |
| PIL164 | -30 | 45.974 | 0.07 | 8.93 | $1.03 \times 10^{-5}$ | 0.03 | 7.31 | $1.07 \times 10^{-5}$ |
| PIL166 | -30 | 45.466 | 0.12 | 7.60 | $1.11 \times 10^{-5}$ | 0.03 | 6.45 | $1.20 \times 10^{-5}$ |
| PIL268 | -30 | 47.240 | 0.21 | 7.82 | $1.10 \times 10^{-5}$ | 0.02 | 5.00 | $1.31 \times 10^{-5}$ |

[1] Experiment from study by Qi and others (2017).

**Table 2** Summary of EBSD analyses

| Sample No. | T (°C) | Data with 30 µm step size | | | | Data with 5 µm step size | | | |
|---|---|---|---|---|---|---|---|---|---|
| | | Map area (Width × Length (mm)) | No. indexed | No. grains | M-index for all indexed pixels | Map area (Width × Length (mm)) | No. indexed | No. grains | M-index for all indexed pixels |
| undeformed | - | 33.18 × 20.55 | 545323 | 11318 | 0.00370 | 24.47 × 7.86 | 4444599 | 1242 | 0.004500 |
| PIL176 | -10 | 25.00 × 25.00 | 353781 | 4728 | 0.00119 | 5.41 × 4.00 | 785025 | 694 | 0.010244 |
| PIL163 | -10 | 24.53 × 10.28 | 201134 | 4851 | 0.00858 | 6.80 × 4.16 | 992513 | 1494 | 0.008886 |
| PIL178 | -10 | 16.20 × 20.30 | 235789 | 6270 | 0.05765 | 5.50 × 4.11 | 690117 | 1028 | 0.046907 |
| PIL177 | -10 | 16.67 × 15.38 | 163507 | 5018 | 0.04068 | 5.49 × 4.14 | 645076 | 1507 | 0.040403 |
| [1]PIL007 | -10 | 13.10 × 5.87 | 91830 | 1655 | 0.12457 | 1.88 × 12.43 | 1010898 | 1789 | 0.118133 |
| PIL254 | -20 | 34.26 × 9.33 | 166929 | 2735 | 0.00227 | 5.41 × 4.24 | 641292 | 903 | 0.006909 |
| PIL182 | -20 | 36.58 × 6.04 | 213919 | 4053 | 0.00540 | 5.48 × 4.28 | 691817 | 907 | 0.004948 |
| PIL184 | -20 | 21.09 × 7.14 | 120209 | 2440 | 0.01296 | 5.50 × 4.13 | 665454 | 1157 | 0.010872 |
| PIL185 | -20 | 26.36 × 7.92 | 121589 | 3127 | 0.01541 | 5.56 × 4.23 | 625128 | 3023 | 0.019941 |
| PIL255 | -20 | 12.42 × 7.95 | 25644 | 1213 | 0.101764 | 3.41 × 4.20 | 472774 | 3057 | 0.106619 |
| PIL165 | -30 | 19.57 × 14.78 | 258779 | 4728 | 0.00077 | 5.45 × 3.07 | 594671 | 589 | 0.006147 |
| PIL162 | -30 | 25.96 × 10.00 | 191672 | 4833 | 0.00442 | 8.11 × 3.97 | 937793 | 2399 | 0.004555 |
| PIL164 | -30 | 18.22 × 22.56 | 229261 | 6087 | 0.02164 | 4.04 × 5.55 | 598744 | 1515 | 0.017329 |
| PIL166 | -30 | 31.26 × 18.29 | 415185 | 8878 | 0.02334 | 8.08 × 3.98 | 1043672 | 6036 | 0.020205 |
| PIL268 | -30 | 5.76 × 20.76 | 93394 | 1039 | 0.101730 | 5.69 × 10.18 | 1664877 | 8215 | 0.063540 |

[1] Experiment from study by Qi and others (2017).

**Table 3** Summary of subgrain and grain sizes

| Sample No. | $T$ (°C) | $2\varepsilon$ | [3]Percentage of repeat counted grains in 2-D (%) | [4]Number density of "distinct" grains ($\mu m^{-2}$) | Number density of "distinct" grains as ratio to starting material | [5]$\bar{D}$ ($\mu m$) | [6]$D_{median}$ ($\mu m$) | [7]$D_{q,25\%}$ ($\mu m$) | [8]$D_{q,75\%}$ ($\mu m$) | [9]$D_{SMR}$ ($\mu m$) | [10]$\overline{D_{big}}$ ($\mu m$) | [11]$\overline{D_{small}}$ ($\mu m$) | [12]$D_{peak}$ ($\mu m$) | [13]$\bar{d}$/[14]$d_{median}$ ($\mu m$) | [15]$d_{q,25\%}$/[16]$d_{q,75\%}$ ($\mu m$) | [17]$d_{peak}$ ($\mu m$) |
|---|---|---|---|---|---|---|---|---|---|---|---|---|---|---|---|---|
| | | | | | | | | | | | | | | | [18]$\varphi \geq 2°$ | |
| undeformed | - | - | 1.90 | 9.97E-06 | 1.00 | 297 | 291 | 165 | 413 | 274 | - | - | 300 | 291/280 | 161/392 | - |
| PIL176 | -10 | 0.03 | 9.45 | 3.24E-05 | 3.25 | 156 | 117 | 48 | 250 | 132 | 250 | 51 | 30 | 134/79 | 39/219 | 20 |
| PIL163 | -10 | 0.05 | 11.71 | 4.75E-05 | 4.76 | 125 | 98 | 54 | 171 | 110 | 197 | 58 | 35 | 104/77 | 51/162 | 25 |
| PIL178 | -10 | 0.08 | 13.47 | 3.82E-05 | 3.83 | 140 | 119 | 72 | 188 | 127 | 194 | 63 | 55 | 127/108 | 62/170 | 50 |
| PIL177 | -10 | 0.12 | 14.19 | 5.14E-05 | 5.15 | 114 | 90 | 54 | 155 | 101 | 184 | 59 | 40 | 96/77 | 45/129 | 30 |
| [1]PIL007 | -10 | 0.19 | 13.07 | 6.25E-05 | 6.27 | 106 | 88 | 51 | 143 | 96 | 174 | 58 | 50 | 96/78 | 46/129 | 45 |
| PIL254 | -20 | 0.03 | 7.40 | 5.75E-05 | 5.77 | 114 | 62 | 36 | 174 | 93 | 197 | 38 | 25 | 91/46 | 29/106 | 20 |
| PIL182 | -20 | 0.04 | 5.30 | 3.97E-05 | 3.98 | 148 | 122 | 62 | 220 | 131 | 188 | 42 | 30 | 103/67 | 33/146 | 25 |
| PIL184 | -20 | 0.08 | 10.61 | 4.73E-05 | 4.74 | 122 | 89 | 48 | 164 | 105 | 169 | 42 | 45 | 88/58 | 36/109 | 20 |
| PIL185 | -20 | 0.12 | 7.76 | 1.05E-04 | 10.49 | 75 | 53 | 36 | 85 | 66 | 132 | 41 | 30 | 55/40 | 28/63 | 20 |
| PIL255 | -20 | 0.20 | 12.29 | 1.28E-04 | 12.85 | 64 | 53 | 36 | 81 | 59 | 106 | 41 | 30 | 55/46 | 32/68 | 25 |
| PIL165 | -30 | 0.03 | 2.07 | 3.15E-05 | 3.16 | 149 | 108 | 48 | 241 | 126 | 203 | 38 | 40 | 108/60 | 32/152 | 20 |
| PIL162 | -30 | 0.05 | 4.87 | 7.27E-05 | 7.29 | 103 | 76 | 45 | 135 | 91 | 144 | 40 | 35 | 70/49 | 31/86 | 20 |
| PIL164 | -30 | 0.07 | 5.58 | 6.67E-05 | 6.69 | 98 | 61 | 39 | 113 | 82 | 158 | 39 | 30 | 59/38 | 27/65 | 20 |
| PIL166 | -30 | 0.12 | 6.01 | 1.34E-04 | 13.45 | 67 | 54 | 37 | 79 | 61 | 104 | 70 | 35 | 57/47 | 32/69 | 25 |
| PIL268 | -30 | 0.21 | 5.66 | 1.18E-04 | 11.88 | 60 | 37 | 29 | 53 | 50 | 158 | 35 | 30 | 42/30 | 24/41 | 20 |

[1]Experiment from study by Qi and others (2017). [2]True axial strain. [3]See section S4 of supplementary material for method. 2-D grains attributed to the same 3-D grain are selected by a critical misorientation angle threshold of 10°. [4]Number density of "distinct" grains, which is calculated from number of "distinct" grains divided by total grain area. "Distinct" grains are calculated by counting 2-D grains attributed to the same 3-D grain as one. 2-D grains attributed to the same 3-D grain are selected by a critical misorientation angle threshold of 10° (section 3.3.2 and section S4 in supplementary material). [5]Mean grain size. [6]Median grain size. [7]Lower quartiles, which split off the lowest 25% of the grain sizes from the highest 75%. [8]Higher quartiles, which split off the highest 25% of the grain sizes from the lowest 75%. [9]Square mean root grain size. [10]Mean grain size of "big grains". [11]Mean grain size of "small grains". [12]Peak grain size in grain size distribution. [13]Mean subgrain size (with $\varphi \geq 2°$). [14]Median subgrain size (with $\varphi \geq 2°$). [15]Lower quartiles (with $\varphi \geq 2°$), which split off the lowest 25% of the subgrain sizes from the highest 75%. [16]Higher quartiles (with $\varphi \geq 2°$), which split off the highest 25% of the subgrain sizes from the lowest 75%. [17]Peak subgrain size in subgrain size (with $\varphi \geq 2°$) distribution. [18]Boundary misorientation angle.

**Table 4** Summary of the open half-angle of the $c$-axis cone ($\theta$) from this study and the literature

| Reference | Name | Mat-erial | T (°C) | No. of $c$-axes | $\theta$ (°) | Conditions* | | True axial strain rate converted (s-1) | True axial strain converted (%) |
|---|---|---|---|---|---|---|---|---|---|
| This study | PIL163 | $H_2O$ | -10 | 353781 | 40 | | $\varepsilon = 5\%$ | $1.06 \times 10^{-5}$ | 5.0 |
| | PIL178 | $H_2O$ | -10 | 201134 | 36 | | $\varepsilon = 8\%$ | $1.19 \times 10^{-5}$ | 8.0 |
| | PIL177 | $H_2O$ | -10 | 235789 | 36 | | $\varepsilon = 12\%$ | $1.21 \times 10^{-5}$ | 12.0 |
| | PIL007 | $H_2O$ | -10 | 163507 | 34 | | $\varepsilon = 19\%$ | $1.22 \times 10^{-5}$ | 19.0 |
| | PIL182 | $H_2O$ | -20 | 213919 | 30 | Constant displacement rate | $\varepsilon = 4\%$ | $8.94 \times 10^{-6}$ | 4.0 |
| | PIL184 | $H_2O$ | -20 | 120209 | 26 | $\dot{\varepsilon} = \sim 1 \times 10^{-5} s^{-1}$ | $\varepsilon = 8\%$ | $1.17 \times 10^{-5}$ | 8.0 |
| | PIL185 | $H_2O$ | -20 | 121589 | 28 | | $\varepsilon = 12\%$ | $1.19 \times 10^{-5}$ | 12.0 |
| | PIL255 | $H_2O$ | -20 | 25644 | 32 | | $\varepsilon = 20\%$ | $1.28 \times 10^{-5}$ | 20.0 |
| | PIL164 | $H_2O$ | -30 | 229261 | 14 | | $\varepsilon = 7\%$ | $1.07 \times 10^{-5}$ | 7.0 |
| | PIL166 | $H_2O$ | -30 | 415185 | 16 | | $\varepsilon = 12\%$ | $1.20 \times 10^{-5}$ | 12.0 |
| | PIL268 | $H_2O$ | -30 | 93394 | 8 | | $\varepsilon = 21\%$ | $1.31 \times 10^{-5}$ | 21.0 |
| Jacka and Maccagnan (1984) | A2 | $H_2O$ | -3 | 132 | 42 | | $\dot{\gamma} = 3.6 \times 10^{-8} s^{-1}, \gamma = 2.4\%$ | $5.17 \times 10^{-8}$ | 3.4 |
| | A3 | $H_2O$ | -3 | 98 | 36 | | $\dot{\gamma} = 4.0 \times 10^{-8} s^{-1}, \gamma = 2.9\%$ | $5.77 \times 10^{-8}$ | 4.1 |
| | A4 | $H_2O$ | -3 | 111 | 28 | Constant load | $\dot{\gamma} = 6.1 \times 10^{-8} s^{-1}, \gamma = 6.8\%$ | $9.04 \times 10^{-8}$ | 9.8 |
| | A5 | $H_2O$ | -3 | 95 | 36 | $\sigma = \sim 0.2$ MPa | $\dot{\gamma} = 6.3 \times 10^{-8} s^{-1}, \gamma = 7.3\%$ | $9.37 \times 10^{-8}$ | 10.6 |
| | A6 | $H_2O$ | -3 | 108 | 26 | | $\dot{\gamma} = 6.1 \times 10^{-8} s^{-1}, \gamma = 15.0\%$ | $9.53 \times 10^{-8}$ | 22.3 |
| | A7 | $H_2O$ | -3 | 96 | 30 | | $\dot{\gamma} = 6.0 \times 10^{-8} s^{-1}, \gamma = 32.5\%$ | $1.02 \times 10^{-7}$ | 51.0 |
| Jacka and Li (2000) | N/A | $H_2O$ | -5 | 87 | 26 | Constant load, $\sigma = 0.2$ MPa | $\dot{\gamma} = 3.4 \times 10^{-8} s^{-1}, \gamma = 11.0\%$ | $5.18 \times 10^{-8}$ | 16.2 |
| | N/A | $H_2O$ | -10 | 100 | 32 | Constant load, $\sigma = 0.2$ MPa | $\dot{\gamma} = 6.6 \times 10^{-9} s^{-1}, \gamma = 10.0\%$ | $9.99 \times 10^{-9}$ | 14.6 |
| | N/A | $H_2O$ | -15 | 173 | 38 | Constant load, $\sigma = 0.5$ MPa | $\dot{\gamma} = 7.5 \times 10^{-8} s^{-1}, \gamma = 11.0\%$ | $1.14 \times 10^{-7}$ | 16.2 |
| | N/A | $H_2O$ | -15 | 199 | 32 | Constant load, $\sigma = 0.4$ MPa | $\dot{\gamma} = 3.6 \times 10^{-8} s^{-1}, \gamma = 11.0\%$ | $5.49 \times 10^{-8}$ | 16.2 |
| Piazolo et al (2013) | MD6 | $D_2O$ | -7 | N/A | 35 | | $\dot{e} = 6 \times 10^{-7} s^{-1}, e = 10\%$ | $6.67 \times 10^{-7}$ | 11.0 |
| | MD10 | $D_2O$ | -7 | N/A | 35 | | $\dot{e} = 2.5 \times 10^{-6} s^{-1}, e = 10\%$ | $2.78 \times 10^{-6}$ | 11.0 |
| | MD3 | $D_2O$ | -7 | N/A | 35 | Constant displacement rate | $\dot{e} = 2.5 \times 10^{-6} s^{-1}, e = 20\%$ | $3.13 \times 10^{-6}$ | 22.0 |
| | MD12 | $D_2O$ | -7 | N/A | 35 | | $\dot{e} = 1.0 \times 10^{-5} s^{-1}, e = 10\%$ | $1.11 \times 10^{-5}$ | 11.0 |
| | MD4 | $D_2O$ | -7 | N/A | 35 | | $\dot{e} = 1.0 \times 10^{-5} s^{-1}, e = 20\%$ | $1.25 \times 10^{-5}$ | 22.0 |
| | MD22 | $D_2O$ | -7 | N/A | 30 | | $\dot{e} = 1.0 \times 10^{-5} s^{-1}, e = 40\%$ | $1.67 \times 10^{-5}$ | 51.0 |
| Montagnat et al (2015) | N/A | $H_2O$ | -5 | 2838 | 40 | Constant load, $\sigma = 0.8$ MPa | $\dot{e} = 1.2 \times 10^{-7} s^{-1}, e = 7\%$ | $1.30 \times 10^{-7}$ | 7.0 |
| | N/A | $H_2O$ | -5 | N/A | 35 | Constant load, $\sigma = 0.75$ | $\dot{e} = 3.9 \times 10^{-7} s^{-1}, e = 12\%$ | $4.43 \times 10^{-7}$ | 12.8 |
| | N/A | $H_2O$ | -5 | 1862 | 35 | Constant load, $\sigma = 0.7$ MPa | $\dot{e} = 3.8 \times 10^{-7} s^{-1}, e = 13\%$ | $4.37 \times 10^{-7}$ | 13.9 |
| | N/A | $H_2O$ | -5 | 830 | 33 | Constant load, $\sigma = 0.8$ MPa | $\dot{e} = 3.8 \times 10^{-7} s^{-1}, e = 18\%$ | $4.63 \times 10^{-7}$ | 19.9 |

| Reference | Sample | Medium | Temp | | | Method | Conditions | | |
|---|---|---|---|---|---|---|---|---|---|
| Qi et al (2017) | PIL7 | $H_2O$ | -10 | N/A | 37 | Constant displacement rate | $\dot{\varepsilon} = \sim1 \times 10^{-5}s^{-1}, \varepsilon = 18\%$ | $1.10 \times 10^{-5}$ | 18.0 |
| | PIL32 | $H_2O$ | -10 | N/A | 34 | | $\dot{\varepsilon} = \sim2 \times 10^{-6}s^{-1}, \varepsilon = 21\%$ | $2.31 \times 10^{-6}$ | 21.0 |
| | PIL33 | $H_2O$ | -10 | N/A | 26 | | $\dot{\varepsilon} = \sim2 \times 10^{-4}s^{-1}, \varepsilon = 22\%$ | $2.42 \times 10^{-4}$ | 22.0 |
| | PIL35 | $H_2O$ | -10 | N/A | 35 | | $\dot{\varepsilon} = \sim1 \times 10^{-5}s^{-1}, \varepsilon = 13\%$ | $1.35 \times 10^{-5}$ | 13.0 |
| | PIL36 | $H_2O$ | -10 | N/A | 34 | | $\dot{\varepsilon} = \sim5 \times 10^{-5}s^{-1}, \varepsilon = 19\%$ | $5.02 \times 10^{-5}$ | 19.0 |
| Vaughan et al (2017) | def013 | $H_2O$ | -5 | 206641 | 42 | Constant displacement rate $\dot{e} = \sim1 \times 10^{-6}s^{-1}$ | $e = 3\%$ | $1.03 \times 10^{-6}$ | 3.0 |
| | def012 | $H_2O$ | -5 | 309428 | 36 | | $e = 5\%$ | $1.05 \times 10^{-6}$ | 5.1 |
| | def011 | $H_2O$ | -5 | 218653 | 38 | | $e = 7.5\%$ | $1.08 \times 10^{-6}$ | 7.8 |
| | def010 | $H_2O$ | -5 | 335722 | 34 | | $e = 10\%$ | $1.11 \times 10^{-6}$ | 10.5 |
| Craw et al (2018) | PIL133 | $H_2O$ | -30 | N/A | 0 | Constant displacement rate | $\dot{\varepsilon} = \sim2 \times 10^{-6}s^{-1}, \varepsilon = 20\%$ | $2.60 \times 10^{-6}$ | 20 |
| | PIL141 | $H_2O$ | -30 | N/A | 0 | | $\dot{\varepsilon} = \sim5 \times 10^{-6}s^{-1}, \varepsilon = 23\%$ | $7.20 \times 10^{-6}$ | 23 |
| | PIL132 | $H_2O$ | -30 | N/A | 0 | | $\dot{\varepsilon} = \sim2 \times 10^{-5}s^{-1}, \varepsilon = 20\%$ | $2.80 \times 10^{-5}$ | 20 |
| Wilson et al (2020) | MD7 | $D_2O$ | -3 | N/A | 34 | Constant displacement rate | $\dot{e} = 1.0 \times 10^{-5}s^{-1}, e = 20\%$ | $1.25 \times 10^{-5}$ | 22 |
| | MD9 | $D_2O$ | -10 | N/A | 33 | | $\dot{e} = 2.5 \times 10^{-6}s^{-1}, e = 20\%$ | $3.13 \times 10^{-6}$ | 22 |
| | DH24 | $D_2O$ | -20 | N/A | 30 | | $\dot{e} = 2.5 \times 10^{-6}s^{-1}, e = 20\%$ | $3.13 \times 10^{-6}$ | 22 |
| | D1_5 | $D_2O$ | -3 | N/A | 34 | | $\dot{e} = 2.5 \times 10^{-6}s^{-1}, e = 20\%$ | $3.13 \times 10^{-6}$ | 22 |
| | D1_1 | $D_2O$ | -1 | N/A | 36 | | $\dot{e} = 2.5 \times 10^{-6}s^{-1}, e = 20\%$ | $3.13 \times 10^{-6}$ | 22 |
| Hooke and Hudleston (1981) | Coarse -100m | Natu-ral ice | -4 ~ -6 | 65 | 22-32 | Constant uniaxial strain rate $\dot{e} = 5.7 \times 10^{-11}s^{-1}$ | $\bar{\gamma}_{oc} = 40\%$ | $7.71 \times 10^{-11}$ | 30 |
| | Coarse -125m | | | 53 | 28-38 | | $\bar{\gamma}_{oc} = 50\%$ | $8.37 \times 10^{-11}$ | 38 |
| | Coarse -154m | | | 82 | 22-32 | | $\bar{\gamma}_{oc} = 60\%$ | $9.09 \times 10^{-11}$ | 47 |
| | Coarse -175m | | | 56 | 20-30 | | $\bar{\gamma}_{oc} = 70\%$ | $9.90 \times 10^{-11}$ | 55 |
| | Coarse -191m | | | 93 | 19-29 | | $\bar{\gamma}_{oc} = 80\%$ | $1.08 \times 10^{-10}$ | 64 |
| | Coarse -215m | | | 155 | 18-28 | | $\bar{\gamma}_{oc} = 90\%$ | $1.18 \times 10^{-10}$ | 73 |
| | Coarse -238m | | | 61 | 15-25 | | $\bar{\gamma}_{oc} = 110\%$ | $1.41 \times 10^{-10}$ | 90 |
| | Coarse -291m | | | 119 | 14-24 | | $\bar{\gamma}_{oc} = 150\%$ | $2.00 \times 10^{-10}$ | 126 |
| | Coarse -315m | | | 102 | 13-23 | | $\bar{\gamma}_{oc} = 170\%$ | $2.38 \times 10^{-10}$ | 143 |
| | Fine -125m | | | 52 | 26 | | $\bar{\gamma}_{oc} = 40\%$ | $7.71 \times 10^{-11}$ | 30 |
| | Fine -150m | | | 89 | 21 | | $\bar{\gamma}_{oc} = 50\%$ | $8.37 \times 10^{-11}$ | 38 |
| | Fine -175m | | | 65 | 28 | | $\bar{\gamma}_{oc} = 60\%$ | $9.09 \times 10^{-11}$ | 47 |

| | | | | | |
|---|---|---|---|---|---|
| Fine<br>-238m | 79 | 17 | $\bar{\gamma}_{oc} = 110\%$ | $1.41 \times 10^{-10}$ | 90 |

\* $\dot{\varepsilon}$ is the true axial strain rate, $\varepsilon$ is the true axial strain, $\dot{\gamma}$ is the octahedral shear strain rate, $\gamma$ is the octahedral shear strain, $\dot{e}$ is the engineering axial strain rate, $e$ is the engineering axial strain, $\sigma$ is the initial stress, $\bar{\gamma}_{oc}$ is the natural octahedral unit shear.

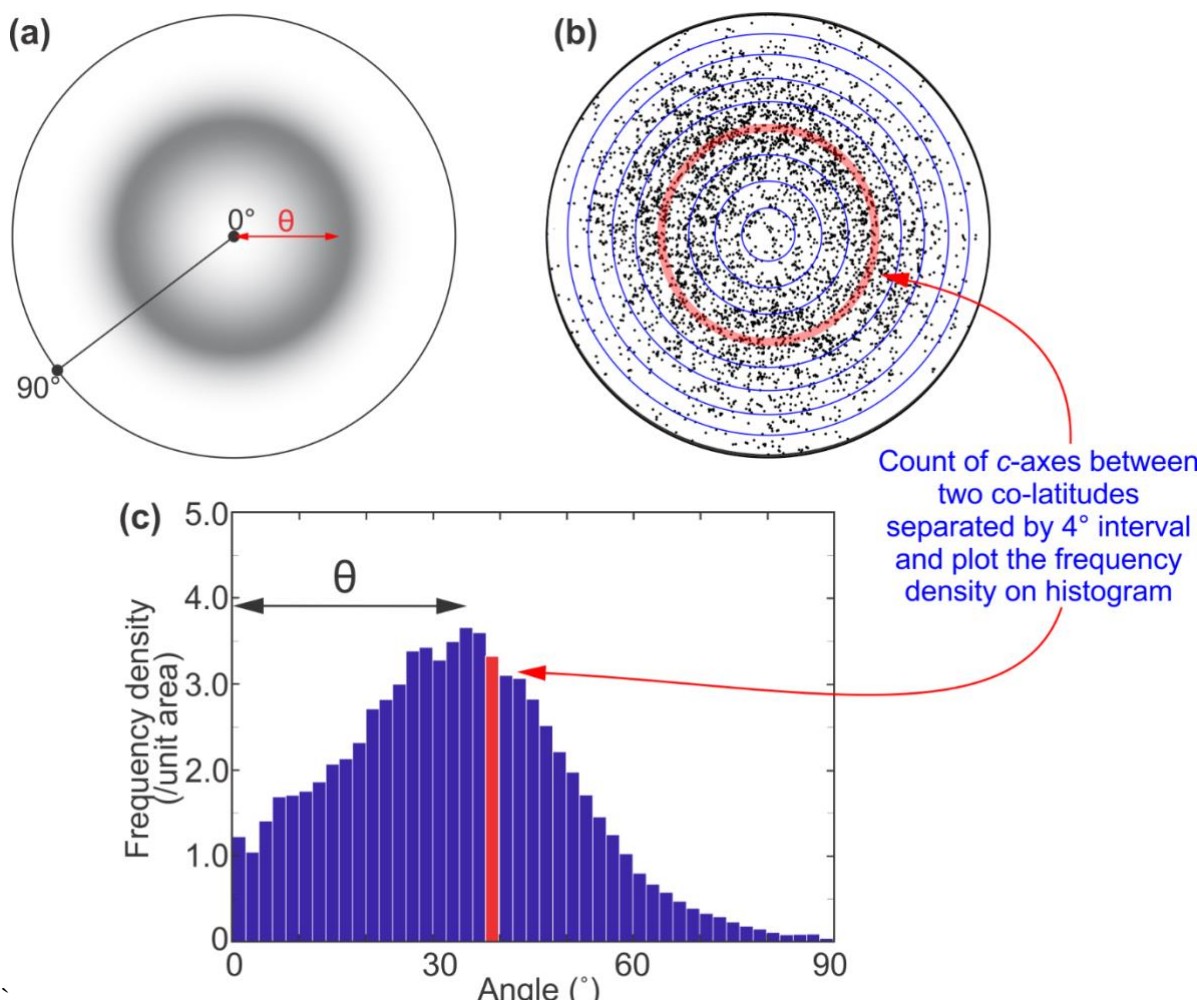

**Figure 1. (a)** Typical *c*-axes distribution at high temperatures with compression axis perpendicular to the page. **(b)** A schematic drawing explaining the method used to quantify the distribution of *c*-axes. The *c*-axes point pole figure taken from PIL178 is used as an example. The pole figure is plotted with lower hemisphere equal-area projection, and compression axis perpendicular to the page. Only 3000 points are plotted for demonstration purpose. At a given angle, red transparent circle covering co-latitudes separated by 4 degrees' interval is drawn. The points lying between the given co-latitudes (covered by the red transparent circle) are counted. The frequency density of the points is calculated from the normalised counts divided by the normalised area between the given co-latitudes. **(c)** The distribution of *c*-axes frequency density as a function of angle to the compression axis. The angle corresponds to the peak in the distribution is taken as the opening half-angle $\theta$ for the cone (small circle) shaped *c*-axes distribution. Throughout the text this is referred to as the opening-angle.

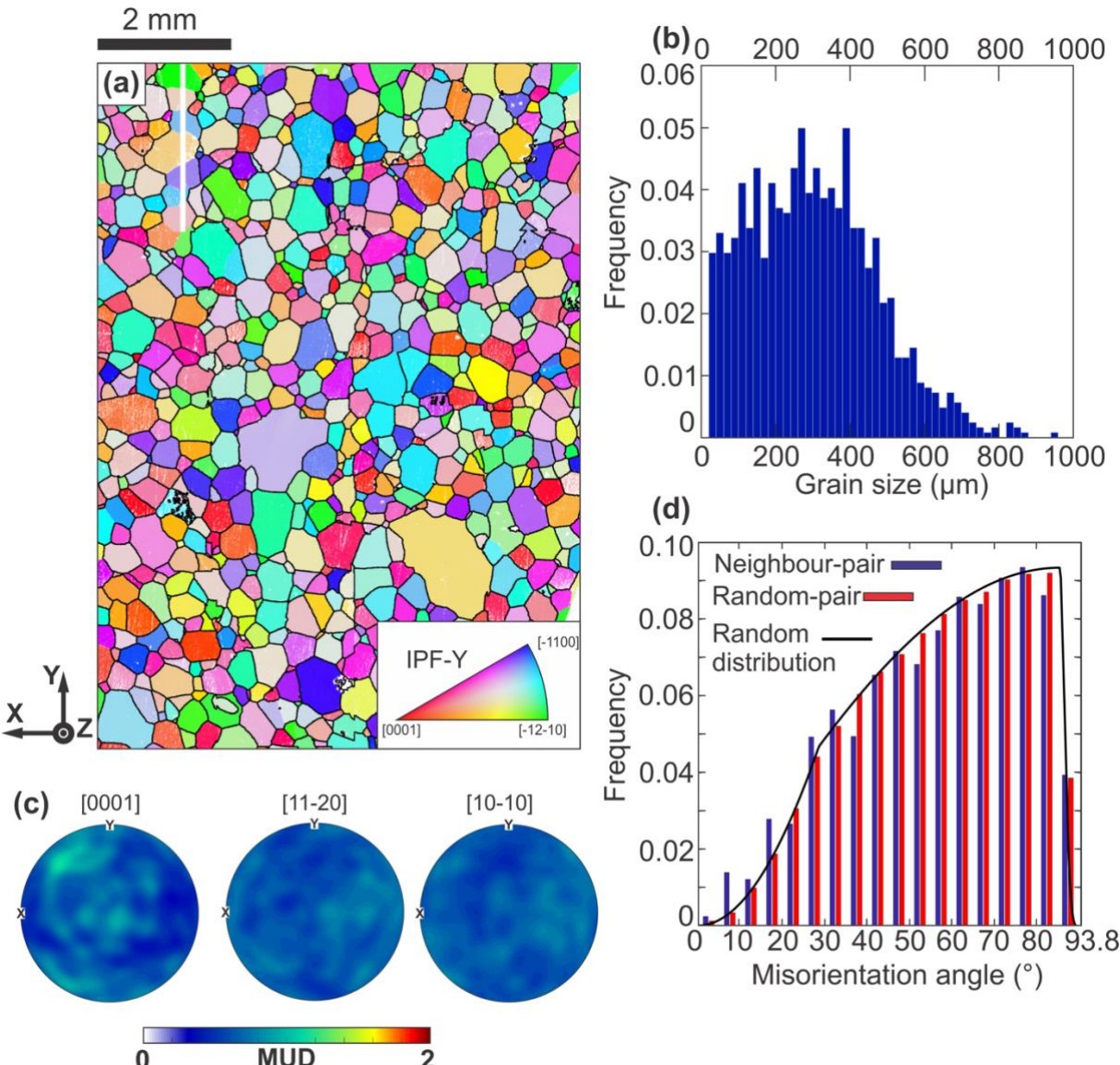

**Figure 2.** Microstructural details of undeformed standard ice. The EBSD data collected with 5 µm step size are presented as (**a**) Orientation maps coloured by IPF-Y, which uses the colour map to indicate the crystallographic axes that are parallel to the y-axis as shown by the black arrows. (**b**) Grain size distribution. (**c**) The distributions of orientations for [0001] (*c*-axes), [11-20] (*a*-axes) and [10-10] (poles to *m*-planes). (**d**) Misorientation angle distribution for (**a**). Neighbour-pair misorientation angle distribution is shown with blue bars. Random-pair misorientation angle distribution is shown with red bars. Misorientation angle distribution calculated for randomly distributed ice 1h crystals are shown with black line.

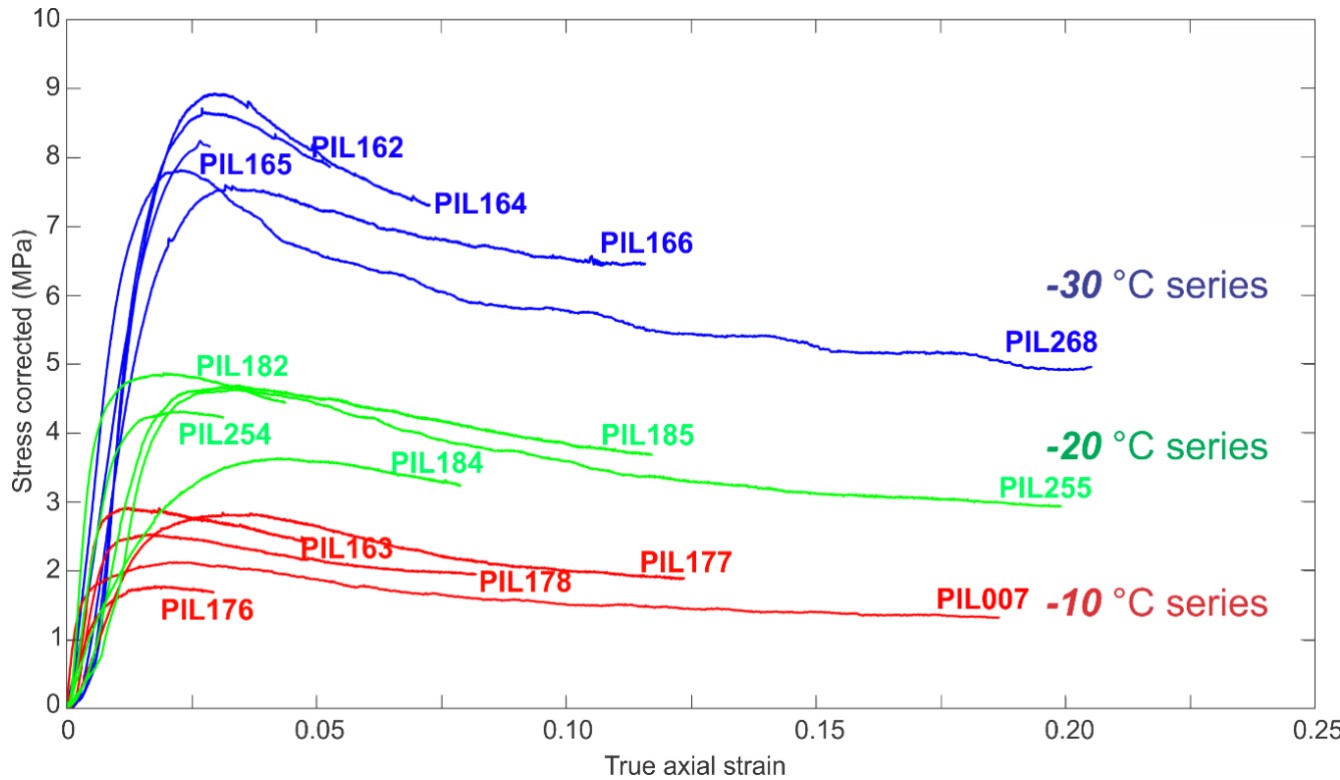

5  **Figure 3.** The stress-strain curves for all the deformed ice samples. The *x*-axis is the true axial strain (Eq. (2)). The *y*-axis is the uniaxial stress. The stress has been corrected for the change of sample cross-sectional area, assuming constant sample volume during the deformation.

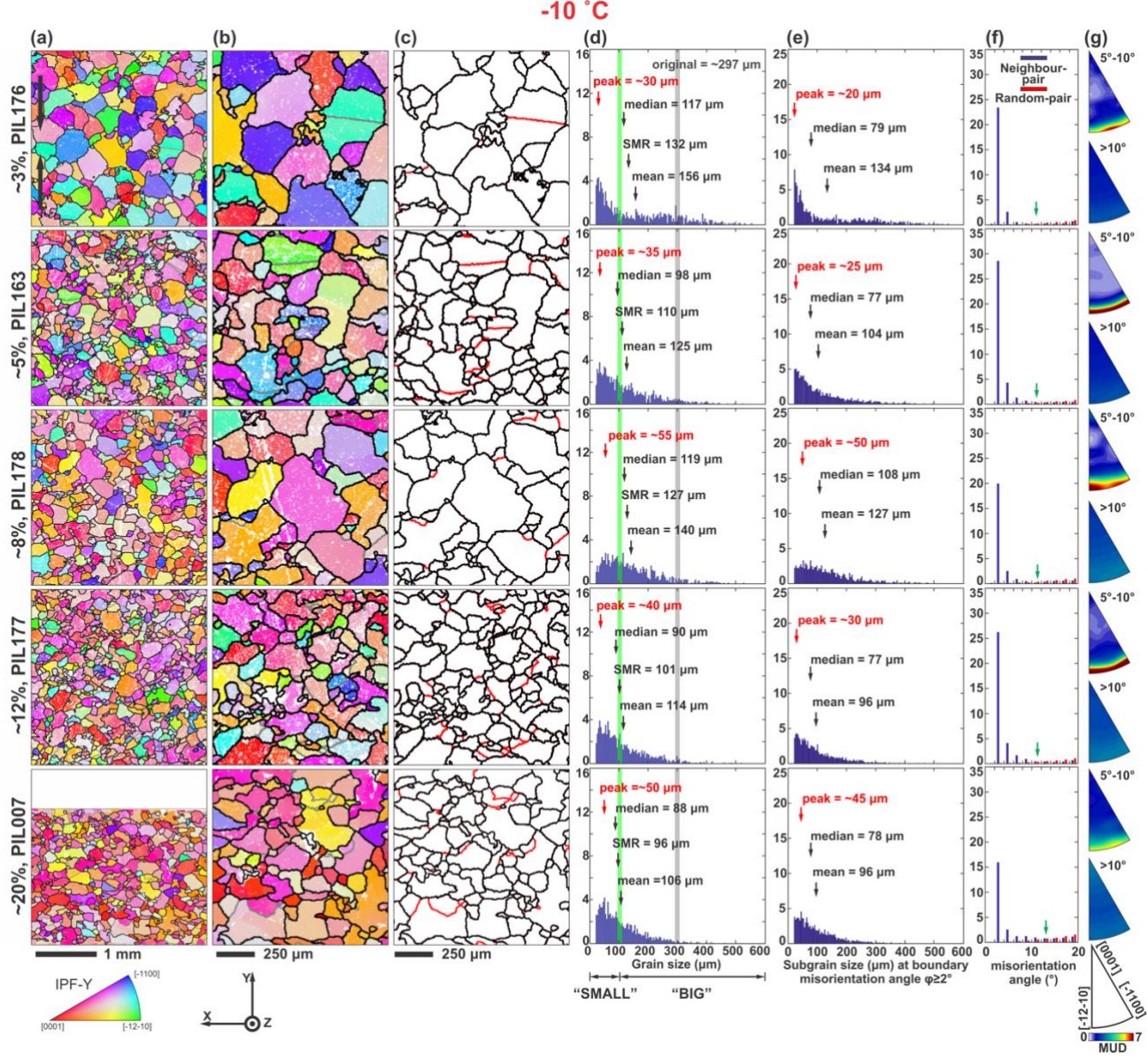

-10 °C

**Figure 4**. Microstructural analyses of deformed ice samples at -10 °C. Axial true strain increases from ~3% on top to ~20% to bottom. The EBSD data collected with 5 µm step size are presented as **(a)** orientation maps at low magnification and **(b)** orientation maps of selected areas at high magnification. Orientation maps are coloured by IPF-Y, which uses the colour map to indicate the crystallographic axes that are parallel to the vertical shortening direction as shown by the black arrows. Ice grain boundaries with a misorientation larger than 10° are shown black. Non-indexed pixels are shown white. Subgrain boundaries, where misorientation angles between neighbouring pixels are between 2° and 10°, are shown grey. Maps show data without interpolation. **(c)** Distribution of subgrain boundaries. Subgrain boundaries are shown red. Grain boundaries are shown black. **(d)** Distribution of ice grain size presented in 4 µm bins. Mean, median and square mean root (SMR) diameters are indicated by black arrows. The main peak of the grain size distribution is indicated by a red arrow. Vertical grey line marks the mean grain size of the starting material. Vertical green line marks the threshold grain size between "big grains" and "small grains" (see text). **(e)** Distribution of subgrain size presented in 4 µm bins. The subgrain size is calculated by applying the boundary misorientation angle of $\varphi \geq 2°$. **(f)** Distribution of neighbour-pair and random-pair misorientation angles. The misorientation angle at which neighbour-pair frequency reduces to be equal to the random-pair frequency is marked with a green arrow. **(g)**. Misorientation axes distribution plotted in crystal reference frame as contoured inverse pole figure (IPF). The contoured IPFs are coloured by MUD. Neighbour-pair misorientation axes for neighbouring pixels with misorientation angles of 5°-10° are presented in the upper box. Grain boundary (>10°) misorientation axes using pixels along the grain boundaries of neighbouring grains are presented in the lower box.

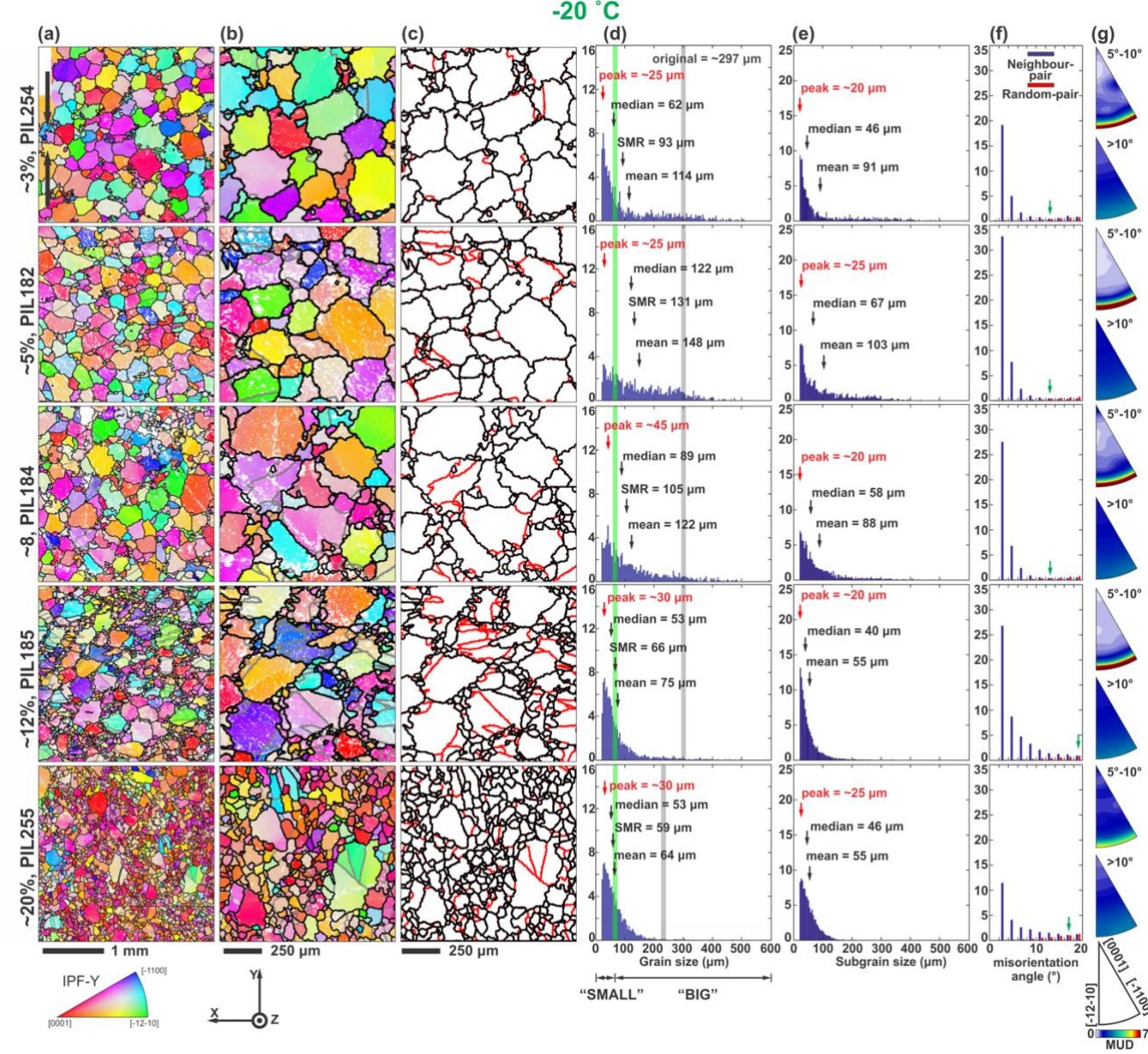

**Figure 5**. Microstructural analyses of deformed ice samples at -20 °C. Axial true strain increases from ~3% on top to ~20% to bottom. The descriptions of columns **(a)** to **(g)** are the same as in Fig. 4.

**-30 °C**

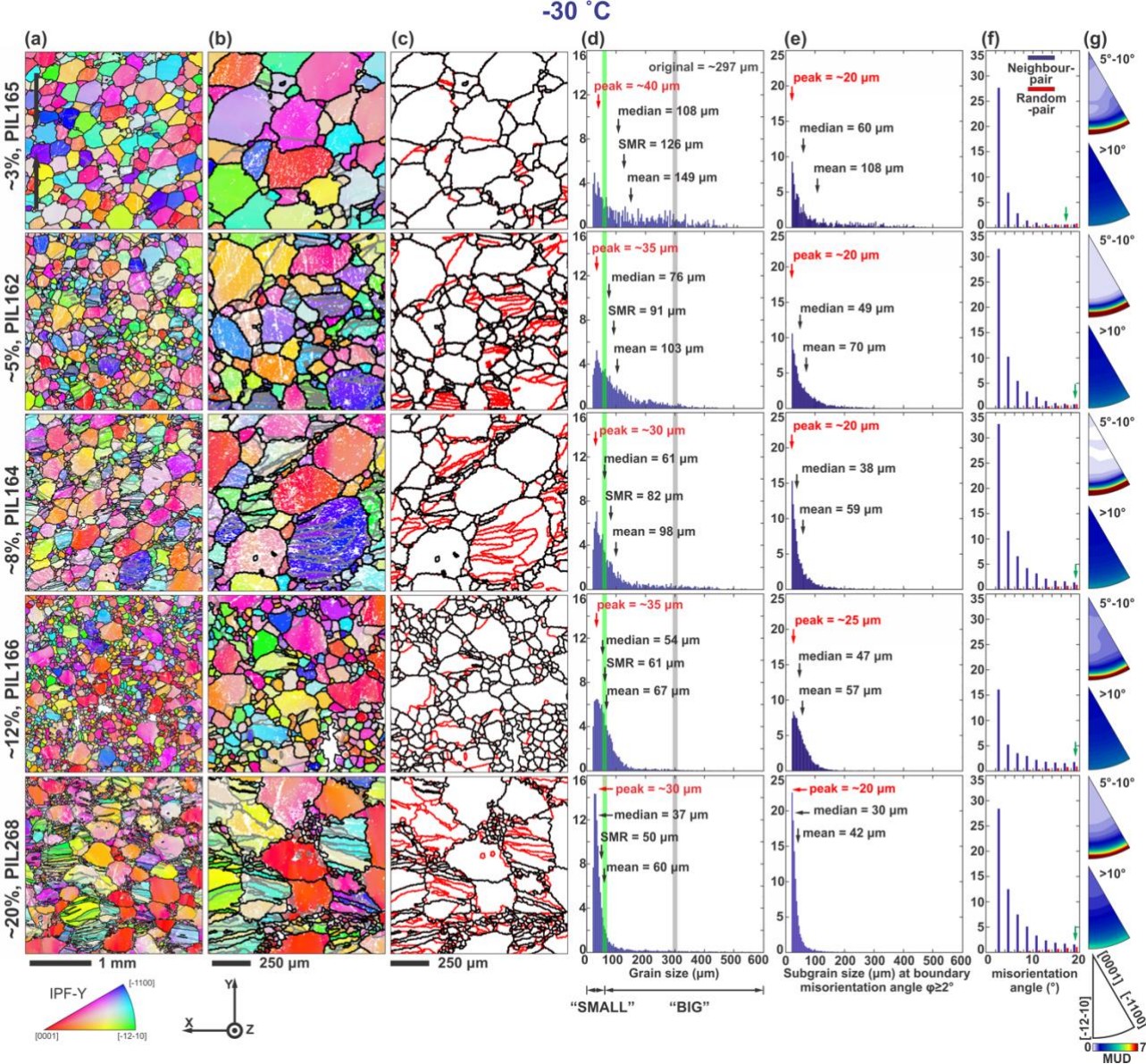

**Figure 6**. Microstructural analyses of deformed ice samples at -30 °C. Axial true strain increases from ~3% on top to ~20% to bottom. The descriptions of columns **(a)** to **(g)** are the same as in Fig. 4.

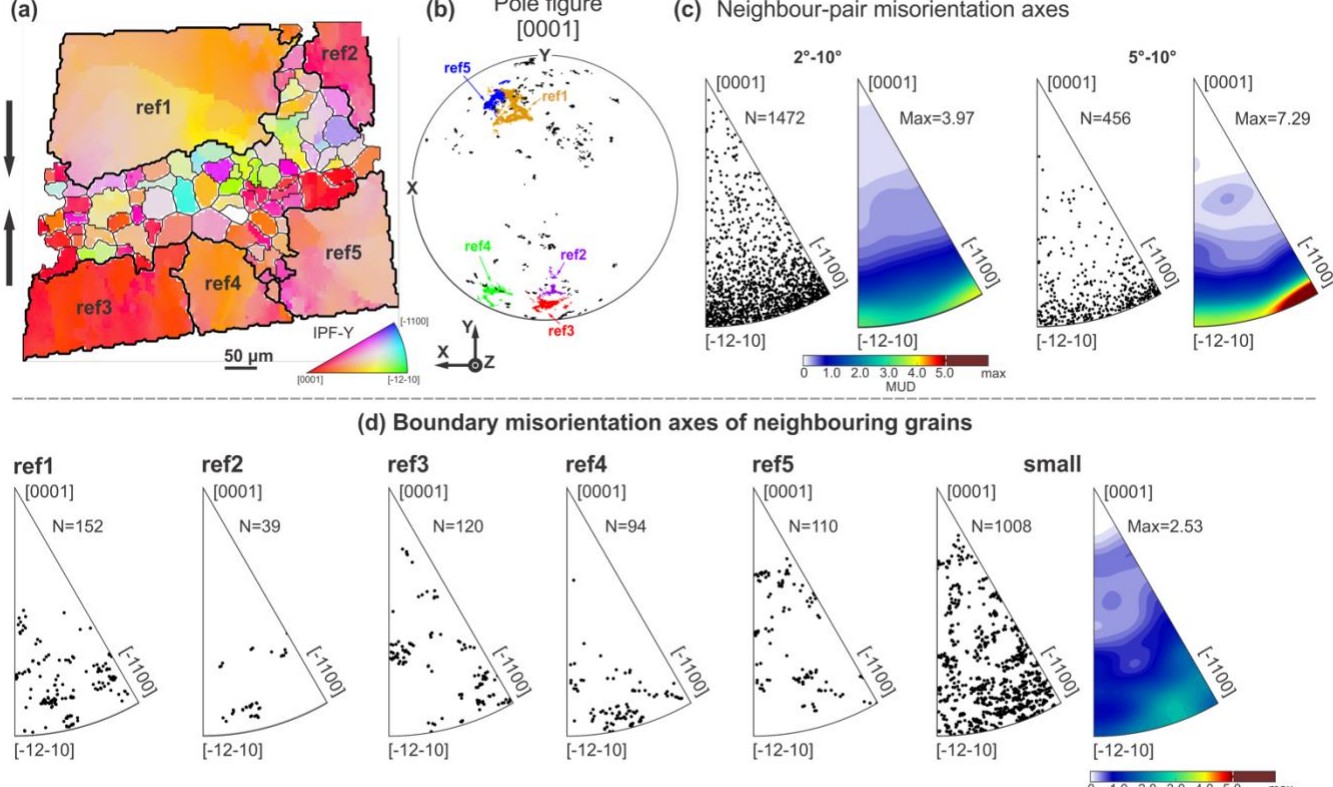

**Figure 7.** Misorientation axes analyses of a sub-area from the EBSD map of sample PIL268 (-30 °C, ~20% strain). **(a)** orientation map coloured by IPF-Y, which uses the colour map to indicate the crystallographic axes that are parallel to the vertical shortening direction as shown by the black arrows. Ice grain boundaries with a misorientation larger than 10° are shown black. "Big" reference grains are shown with thick black boundaries. "Small" grains are shown with thin black boundaries. **(b)** The pole figure of $c$-axes corresponding to orientations of all pixels of grains in **(a)**. $c$-axes of "small" grains are shown with black dots. $c$-axes of "big" reference grains are shown with dots coloured by non-black colours. **(c)** Neighbour-pair misorientation axes for neighbouring pixels with misorientation angles of 2°-10° and 5°-10° corresponding to **(a)**. The misorientation axes are plotted in crystal reference frame as inverse pole figure (IPF). The IPF either shows all points or coloured by MUD. The number of points or maximum value of MUD are given next to each IPF. **(d)** Boundary misorientation axes of "big" reference grains and corresponding neighbouring grains, and neighbouring grains among "small" grains. Grain boundary misorientation axes are calculated using pixels along the grain boundaries of neighbouring grains.

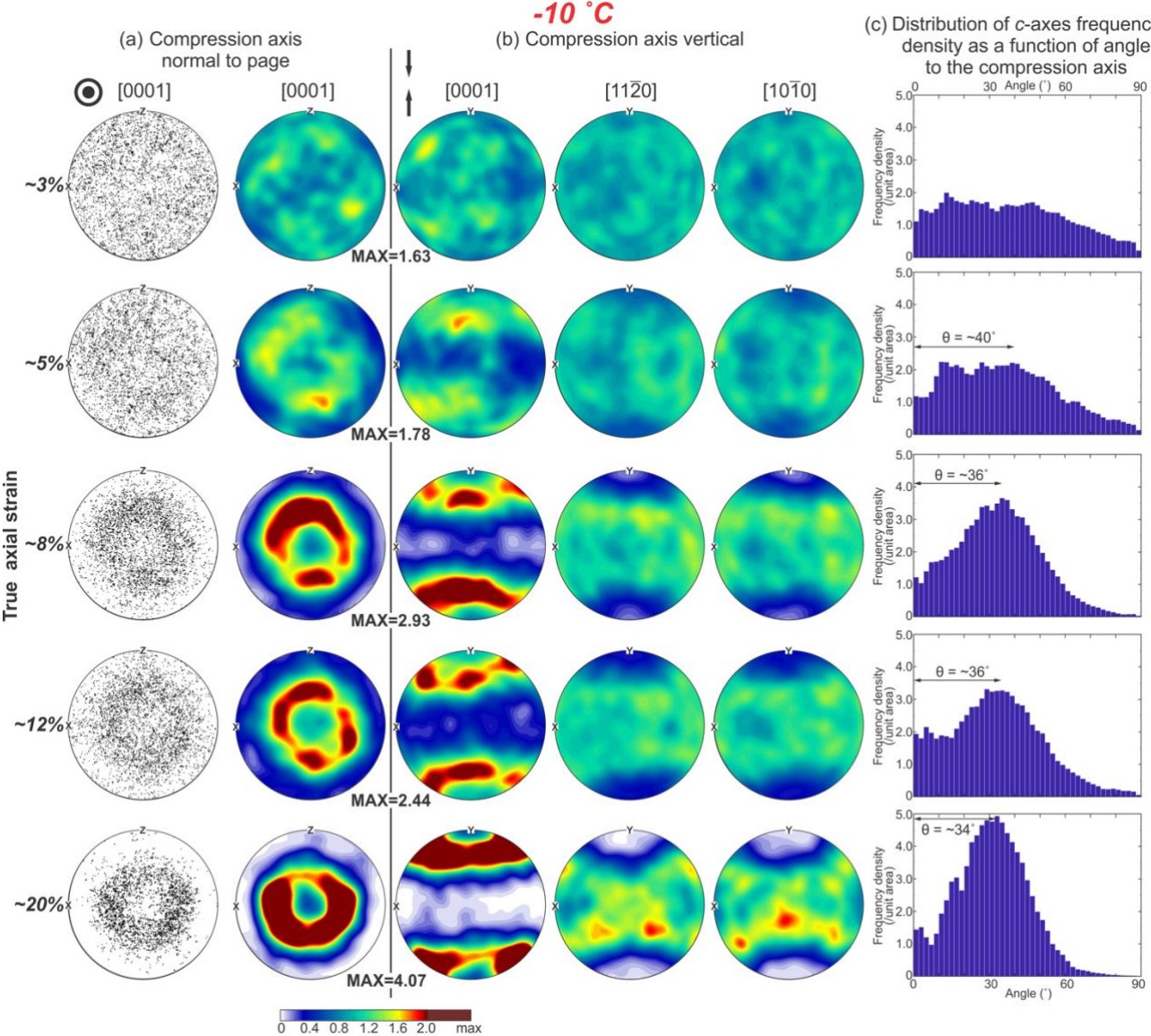

**Figure 8**. Crystallographic preferred orientations (CPOs) from EBSD data with 30 µm step size for ice samples deformed at -10 °C. Axial true strain increases from ~3% on top to ~20% to bottom. **(a)** The distributions of [0001] (*c*-axes) orientations plotted as point pole figures with 5000 randomly selected points and contoured pole figures. The compression axis is perpendicular to the page. **(b)** The distributions of orientations for [0001] (*c*-axes), [11-20] *a*-axes and [10-10] (poles to *m*-planes) plotted as contoured pole figures. The compression axis is vertical. Contoured pole figures are contoured based on MUD. The maximum value of MUD for the *c*-axis CPO of each sample is given between columns (a) and (b). **(c)** Distributions of the [0001] axes frequency density as a function of angle to the compression axis. Open half-angle θ of the cone (small circle) is presented on each histogram.

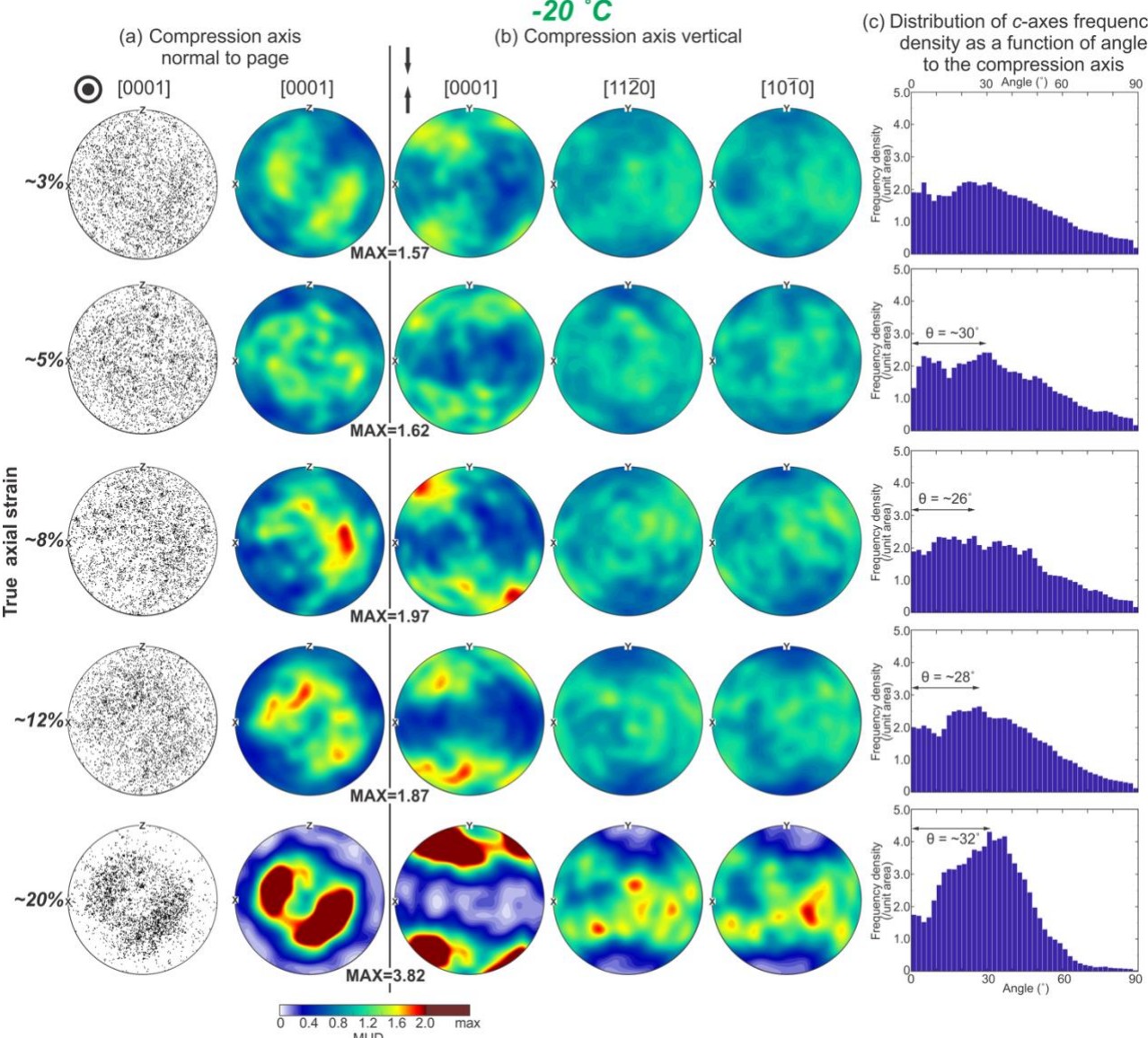

**Figure 9**. Crystallographic preferred orientations (CPOs) from EBSD data with 30 µm step size for ice samples deformed at -20 °C. Explanation of annotations and the descriptions of sections **(a)** to **(c)** are the same as in Fig. 8.

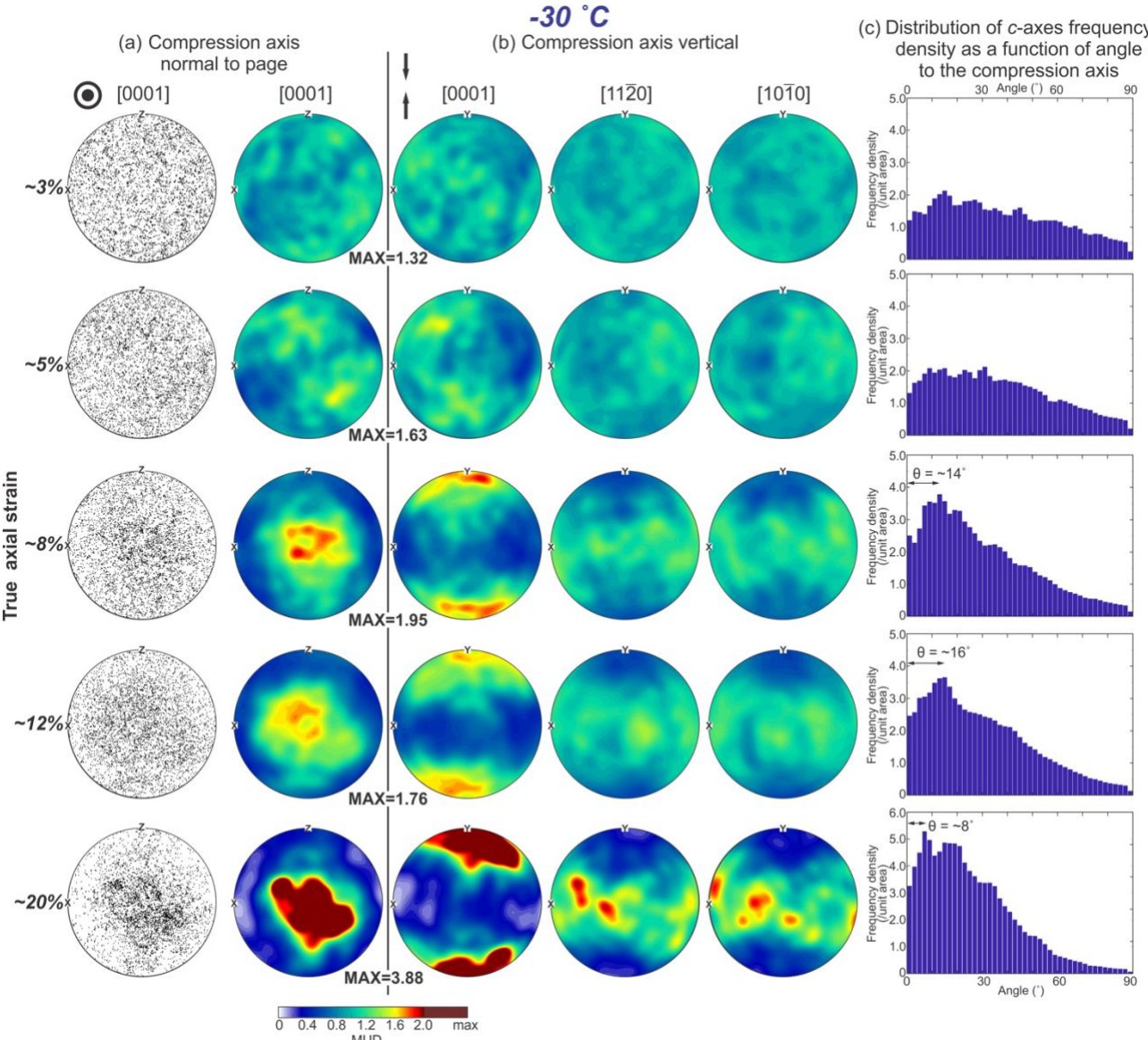

**Figure 10**. Crystallographic preferred orientations (CPOs) from EBSD data with 30 µm step size for ice samples deformed at -30 °C. Explanation of annotations and the descriptions of sections **(a)** to **(c)** are the same as in Fig. 8.

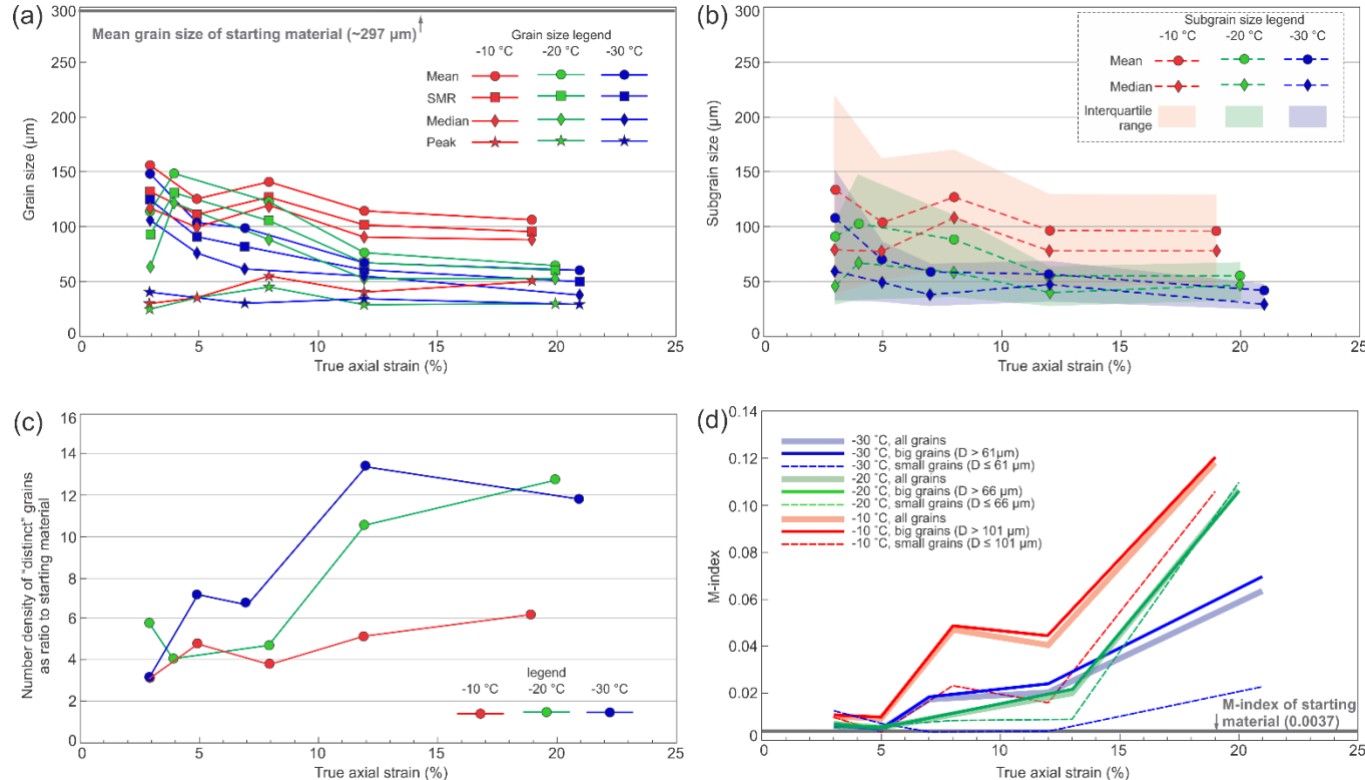

**Figure 11. (a)** Variation in the mean, SMR (square mean root), median and peak grain size as a function of true axial strain in each temperature series. **(b)** Variation in the mean, median subgrain grain size and interquartile range of subgrain size at boundary misorientation angle threshold of 2° as a function of true axial strain in each temperature series. The upper and lower bounds of interquartile range are constrained by lower quartile subgrain size and higher quartile subgrain size, respectively. **(c)** Variation in number density of "distinct" grains as ratio to starting material relative to true axial strain in each temperature series. **(d)** Variation in CPO strength (M-index) as a function of true axial strain for different grain size categories in each temperature series. M-indices are calculated from 5 µm EBSD data.

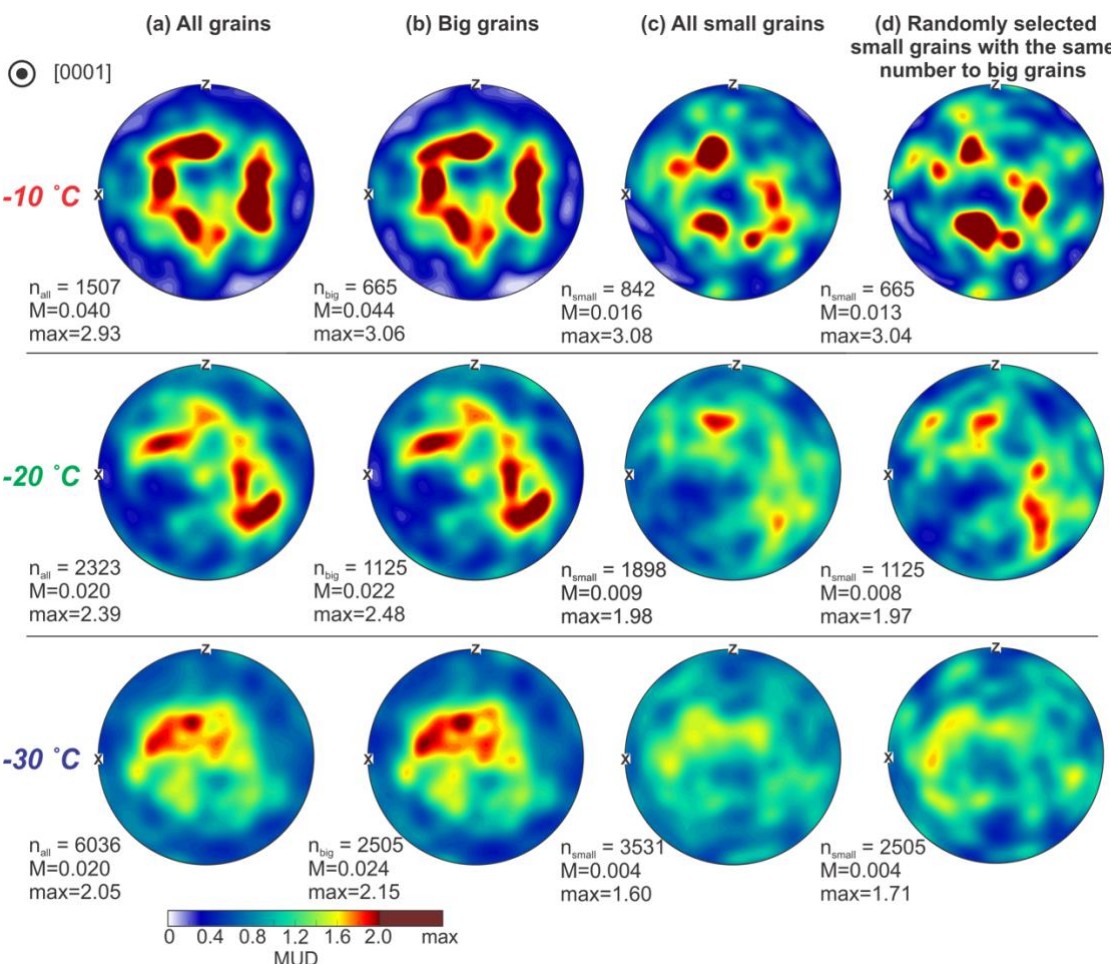

**Figure 12**. Contoured [0001] (*c*-axis) CPOs of **(a)** all grains, **(b)** "big grains", **(c)** "small grains" and **(d)** randomly selected "small" grains with the same grain number of "big" grains, for the samples deformed to ~12% strain at different temperatures. The number of grains, M-indices and max MUD values are marked on the bottom left of pole figures. The *c*-axis CPOs are calculated based on all pixels taken from the EBSD data with 5 µm step size. Compression axis is in the centre of the stereonets.

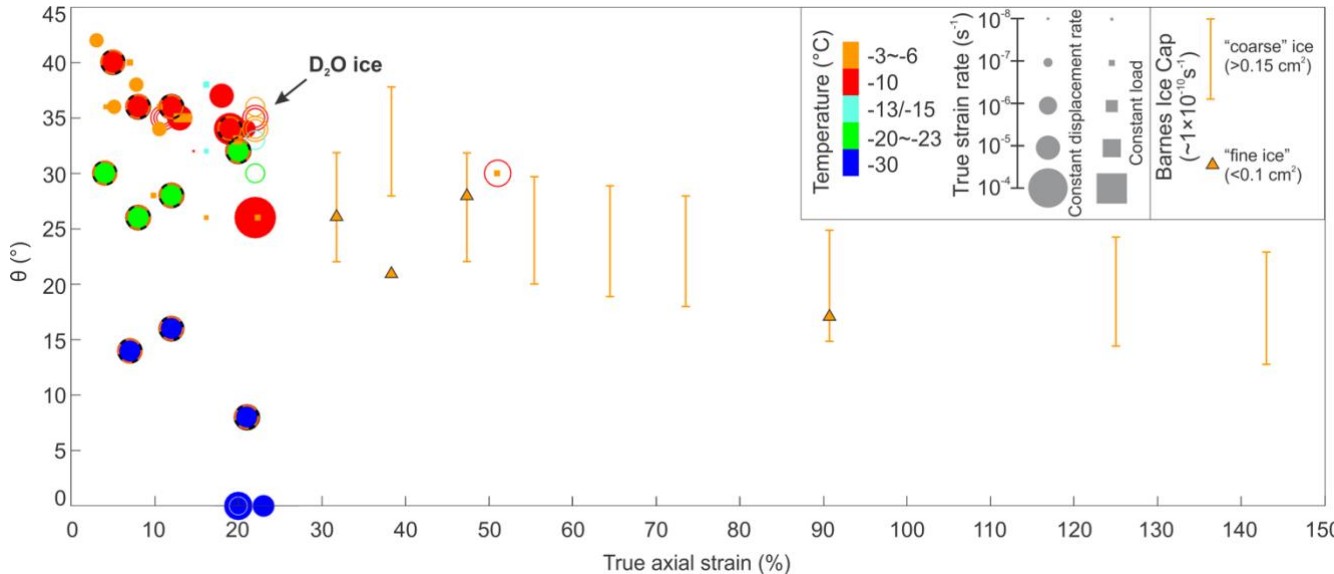

**Figure 13.** Plot of the relationship between the opening-angle, $\theta$, of the cone-shaped $c$-axis CPO and the true strain. The data come from this study and the literature (Table 4). The data from naturally deformed ice (Hooke and Hudleston, 1981) are
5   illustrated by bars with whiskers (cover uncertainty range of the open angle) for "coarse" ice and triangles with black edges for "fine" ice. The data from constant displacement rate experiments on $D_2O$ ice (Piazolo et al, 2013; Wilson et al., 2020) are illustrated by hollow circles. The deformation of $D_2O$ ice at -7 °C is a direct analogue for deforming $H_2O$ ice at −10 °C (Wilson et al., 2019). The data from constant displacement rate experiments on $H_2O$ ice (this study, Vaughan et al., 2017, Qi et al., 2017, Craw et al., 2018) are illustrated by filled circles. Data from this study are highlighted by orange-black edges. The data
10   from constant load experiments (Jacka and Maccagnan, 1984; Jacka and Li, 2000; Montagnat et al., 2015) are illustrated by solid squares. Each marker is sized and coloured by the corresponding true strain rate and temperature, respectively. For all experiments the strain rate shown is the strain rate at the end of the experiment.

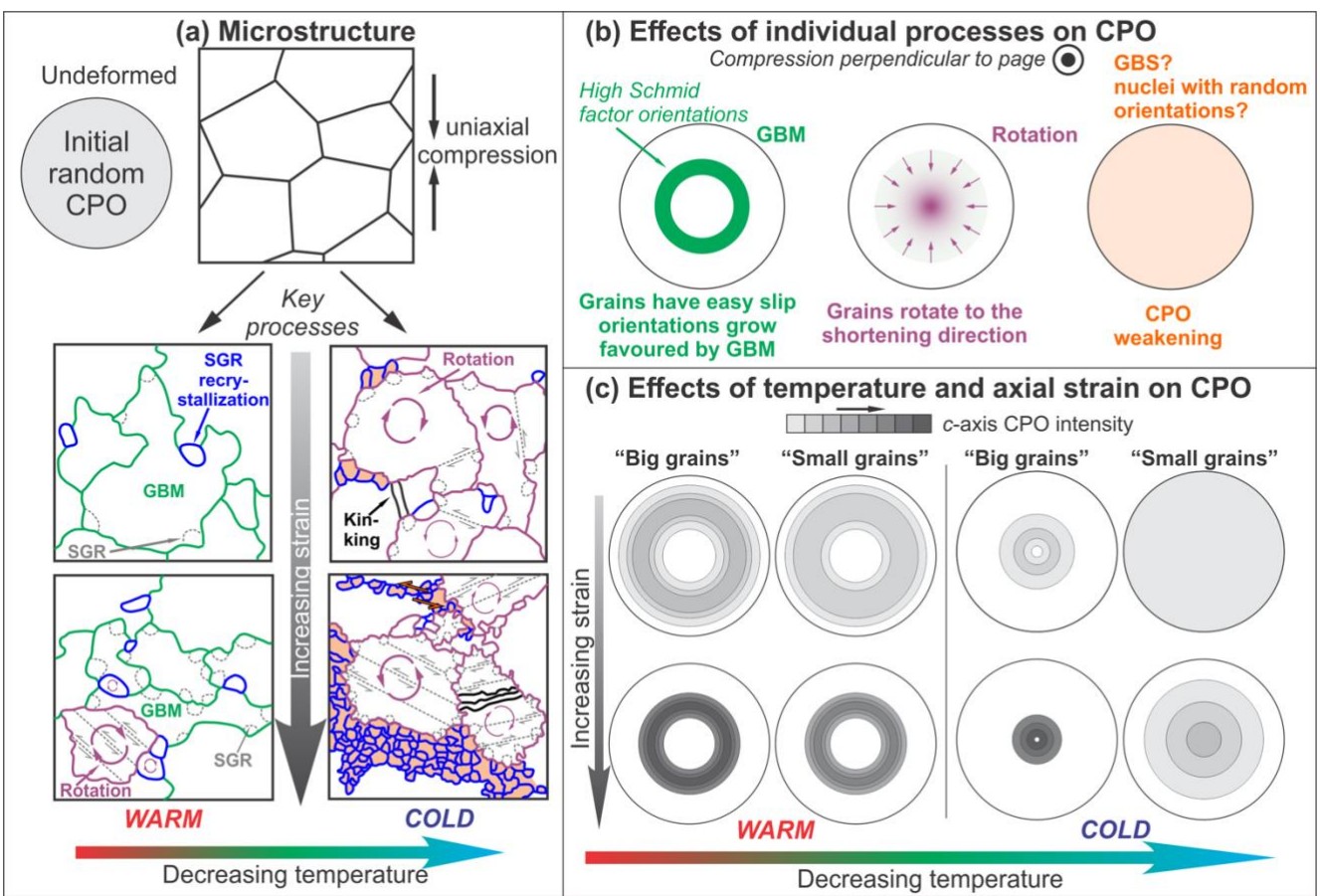

**Figure 14.** Schematic drawing of the microstructure and CPO development in ice deformed under uniaxial compression. **(a)** The effects of temperature and axial strain on the microstructural evolution. Grains undergoing different deformation processes are marked by different colours, with interpretations of the processes presented. **(b)** The effects of individual processes on CPO development. **(c)** The development of CPOs for "small grains" and "big grains" with strain at different temperatures. Starting point (shown in (a)) is a random CPO. SGR: subgrain rotation. GBM: grain boundary migration. GBS: grain boundary sliding.