# Peer review of "Temperature and strain controls on ice deformation mechanisms: insights from the microstructures of samples deformed to progressively higher strains at -10, -20 and -30 $^{\circ}$ C"

_The Cryosphere, 2020_

## Referee Comment (RC1) · Maurine Montagnat (Referee) · 19 Feb 2020

**"Temperature and strain controls on ice deformation mechanisms: insights from the microstructures of samples deformed to progressively higher strains at -10, -20 and -30°C.**

**The Cryosphere**
**February 2020**

In this report, I first provide specific comments following the course of the text, and I finish with some more general comments.

P3 l. 28: why should the sample be cooled in liquid nitrogen? Couldn't that induce some thermal stress due to the fast and strong temperature gradient? Changes in local microstructure, dislocation arrangements, etc. are expected to occur in the first minutes after the test... so this quenching should not avoid it.

P7 l. 8-10: in the paragraph just before it is mentioned that the grain size distributions are mostly bimodals, and therefore not gaussians... The mean and STD parameters are therefore not suited to described them, since they do not represent well the given statistics. I therefore suggest the authors to provide medians and quartile data instead to better fit the type of distributions observed.

P7 l. 30: couldn't it exist a bias link to the lack of resolution in step size and misorientation when getting toward smaller subgrains?

Part 3.2.3: this is not clear to me how is the subgrain size defined and calculated. Although we observe a clear grain boundary structure in the figures, there appears no clear subgrain structure (as one could observe in some minerals or metals for instance). On the contrary, subgrains appear more like straight tilt bands or kink bands, with, in some places, some variations around the straight shape.
I would be curious to see, for instance, how was measured this subgrain size in the sample deformed at 3% at -10°C, or at 12% at -20°C. I think that the author should clarify this technical aspect as they make a lot of explanation rely on such "average" parameters.
Furthermore, provided it is calculated properly, I doubt the distribution is normal, and I think that a metric other than the average would fit best.
In particular, the following assertion is questionable:
"because subgrain rotation recrystallization should produce grains that have similar sizes with subgrains, while bulging nucleation should produce grains that have smaller sizes than subgrains" that rely on a parameter (subgrain size) that is ill measured here, since subgrain structure does not resemble at all the one observe in quartz (Halfpenny et al 2012).
On top of that, this expected hierarchy of grain size depending from the nucleation mechanism comes from one study on quartz and should not be taken as granted, see for instance Humphreys 2004 (Materials Science Forum) that shows clearly a bulged grain much larger that the surrounding subgrains. It will therefore depend on the material and its anisotropy, and on the resolution of the observations (ability to distinguish between a grain resulting from a bulge and one resulting from subgrain rotation...)

p8 l. 8-9: To consider $<D\_small>$ as a good representative of the mean recrystallized grain size is also a strong hypothesis that should be justified (either by some specific observations or by references from previous work). It will, in particular, depend on the relative effect of grain boundary migration compare to nucleation during recrystallization (and therefore on the temperature of the test) since an apparent small grain size at high GBM rate could well be a 2D cut of a strongly lobated grain, while, at lower temperature (lower GBM rate), small grains indeed are newly

recrystallized grains (see for instance the sample deformed at 8% at -10°C, could one certify that small grains observed on the 2D sections are indeed small grains?).
Here again appears the necessity of statistic metrics adapted to the observed distributions.

For the sake of clarity, I would suggest the authors not to mix result presentations and interpretations, and keep interpretations for the discussion part. In particular when interpretation requires additional hypotheses on top of direct observations and results.

P9 l. 13 "At -10 °C, the CPO intensity of "small grains" is lower than "big grains", and this contrast becomes strengthened as the temperature decreases."
This could also be related with the fact that it is less straightforward to distinguish small grains from big grains for these tests, this should be mentioned here.

P11 l. 26
The authors mention "much of the stress increase prior to peak stress relates to elastic strain", and, as they notice just after, this is not coherent with the known Young modulus of ice of 9 Gpa...
There is a broad literature, dating back to the 70's and 80's (Duval et al. 1983, Jacka 1984 for instance, and review by Schulson and Duval 2009) explaining that the transient behavior of ice is not elastic, but anelastic, and is related to the built of an internal stress field related to strain incompatibilities between grains. **I am therefore very astonished to read this sentence here, and I think that this should be corrected before publication**.
The "dissipative deformation" mentioned here is indeed plastic deformation related to intracrystalline dislocation slip, the porosity loss being very likely negligible.

Part 4.1.1: Discussion about GBS. The experimental results shown here present no evidence of a grain size sensitive mechanisms, since there is no initial grain size variation, no study of the influence of grain size on the stress – strain-rate relation. I therefore don't understand why GBS is mentioned here, since it is not necessary at all to explain the observations performed.
Indeed, all results presented here can be explained by intracrystalline dislocation slip accommodated by dynamic recrystallization mechanisms, as very well illustrated in the high quality EBSD observations performed.
Furthermore, there exist a large number of studies showing that GBS occurs significantly only in fine-grained materials (see Boullier and Gueguen 1975, Goldsby and Kohlstedt 1997) where grain boundary diffusion can play a role (Ashby 1973). Diffusion in ice is known to be very slow, that renders the hypothesis of a diffusion-controlled mechanism quite unlikely, especially for large grains, and high strain-rate conditions as encountered here.
The authors could try to calculate the strain-rate expected based on a GBS diffusion flow law (Nabarro-Coble for instance) for similar level of stress as the one of their experiences. They would likely see that the stress – strain-rate curves they obtained are not compatible with a GBS-influencing mechanism.

Part 4.1.2: In this part, the authors use the subgrain size measurements to estimate the role of subgrain rotation in the recrystallization mechanisms.
Once again, the subgrain structure observed here is very far from the ones in quartz, to enable using the paper mentioned here as a reference (Trimby et al. 1998), and I think the authors should be much clearer about the way they evaluate the subgrain size before getting to strong an interpretation from this parameter.
Ice behavior, and in particular in the experiments presented here, is very different from the one of more isotropic materials in the sense that the dislocation substructures are not characterized by subgrain cells as observed in Al or Quartz for instance. This is due to the fact that subgrain substructures as observed in Quartz results from equivalent activity of several slip systems.

Although one observe some c-dislocations in the microstructure, slip system activity in ice remains dominated by basal slip, and resulting subgrains have mostly the shape of large tilt and kink bands. Only close to GB and triple junctions will we find more complex substructures. Is it enough to evaluate an "average" subgrain size? Care must therefore be taken before using interpretations coming from these more isotropic materials. And explanation should be given about how is this subgrain size measured here.

P13 l. 1-2: Indeed, Jacka and Li Jun 1994 evidenced a linear relationship between grain size and stress during dynamic recrystallization of polycrystalline ice (creep experiments, tertiary creep). I think that this should be mentioned here.

P13. l. 5: here again, caution must be taken with making use of the subgrain size as it is still ill defined... and the ice case can not be compared straightforward to Halpenny et al. studies! Indeed, observation given l. 16-17 goes in the direction of my remark... Subgrain size, if measurable here, can not be used similarly as in the other studies mentioned since there is no clear subgrain substructure. But, still, subgrain rotation could explain part of the recrystallization by, for instance, closing the bulges (see Chauve et al. 2017, Phil Trans), or by separating grains via highly misoriented tilt or kink bands. But, indeed, one can not talk about "continous" recrystallization as observed in Al for instance (see Sakai et al. Progress in Materials Science, 60(0):130–207, 3 2014 for a review).

Part 4.1.3
p 14 l. 11: "Because grains with hard slip orientations should have greater internal distortions", there is absolutely no proof of that in ice, and some recent work tend to show that there is no systematic relation between orientation and strain localisation (see Grennerat et a. 2012 for instance) or between orientation and subgrains density (see Journaux et al. 2019 for instance). I think it should not be considered as granted, in particular when not shown directly in your experiments.
Have you tried, for instance, to measure the density of GNDs as a function of grain orientation?

P14 l. 20: GBM instead of GMB

About GBS and apparent texture weakening in small grains: to my point of view, this apparent texture weakening could be related to the nucleation process itself, and the fact that close to GBs, local misorientation can be high, and induce nucleation orientations varying from parent grains orientations (by bulging or subgrain rotation). This process would be enough to justify the small difference in texture concentration in small grains (that could also be due to more spread in data as there are less pixels measured in small grains, since GBs are interfering with the measurement, reducing its quality in small grain areas ?). See for instance the work of Falus et al. 2011 about Olivine for rotation recrystallization or Chauve et al. 2017 for the orientation of nucleus formed by bulging.
The work of Qi et al. 2017 mentioned several times in this part concluded that "the dominant mechanism of CPO development occurs with increasing stress, from GBM, which consumes grains with low Schmid factors, at low stress, to the rotation of basal slip planes to an orientation normal to the compression axis at high stress, due to dislocation glide." I didn't find any mention of "grain size sensitive mechanism" as certified l. 25...
Such a grain size sensitive mechanism should be verified by varying grain size during the experiments and evaluate its effect on a given parameter, such as peak stress, strain-rate or so.
I maintain that there is no proof of such a GSS mechanism in the experiments presented here, and therefore the interpretation should be cleared about that.
That GBS is more active in smaller grains is well known since Boullier and Gueguen work! It does not mean that it should occur in the specific case here, unless otherwise proven...

The hypothesis that GBM being less active at low temperature, the impact of grain rotation driven by intracrystalline slip prevails is much clearer, especially since it is very coherent with the observations that the cone angle is reduced, and more orientations are found close to the vertical. This assertion is, indeed, justified by the experimental observations.

This is, in fact, the main "novelty" of the presented work and should be emphasised more. Speculation about GBS tends to lessen this message, and also the interest of the good quality observations performed in this work.

Part 4.2:
During dynamic recrystallization, weakening is classically (see Humphreys and Haterly 2001 or 2004 for instance, Sakai et al. 2014) attributed to the reduction of hardening based on GBM and nucleation of grains, both reducing the stored strain energy associated with dislocation pile-up or dislocation structures. Therefore dynamic recrystallization induced weakening does not require the interplay of CPO or grain-size sensitive mechanism to be explained.
Another point for this consideration about weakening: the relative weakening at about 20% strain is similar for every temperature cases, at about 35% (Sigma_p – Sigma_f/Sigma_p).  Therefore there is not more weakening with small grains that without... It should rule out the hypothesis of a grain-sensitive mechanism to explain weakening. Nucleation and GBM (each one having different relative influence depending on the temperature) are enough to explain the observed weakening, as expected from the dynamic recrystallization literature.

Part 5:

Point 2: from figure 2, the steady state is not so obviously reached, unless, maybe at -10°C. Maybe the authors should be more careful about it, especially about mentioning it in the conclusion.

Point 3: regarding my previous comments concerning the evaluation of a subgrain size, I think that either the authors explain very clearly how they evaluate this subgrain size, and show that it is meaningful based on their experimental observations (that they do observe a subgrain network, although it does not appear clearly in the given figures, from which extracting a subgrain size appears relevant), or this parameter, even if used in the discussion with care, should not appear in the conclusion.

Point 5: once again, this conclusion makes use of the subgrain size which measurement method is not clear, and therefore should not be used in the conclusion unless clarified.

Point 6: I think that there is nothing really new in this point... it has been demonstrated for many materials undergoing dynamic recrystallization, and it is a direct evidence from energy considerations... Should it really come as an important conclusion? At least, the authors should be care to mentioned "as already observed", or "as expected during dynamic recrystallization"...

Point 7: based on my comments concerning part 4.2, the mention of GBS to explain weakening should be removed. It is also surprising that an hypothesis that is only briefly mentioned in a very short paragraph (4.2), could come to an important conclusion point...

Point 8: same as point 7, and please note that weakening should be measured relatively to the peak stress value (for instance), and it therefore leads to very similar weakening for all temperature conditions (about 35%).

GENERAL COMMENTS:

- In general, there is a lack of references from the work done on recrystallization (on ice and other materials) by others authors than the authors' team.... this is especially true, for instance, in part 4.1.3, and this should be corrected. In particular when other's work do not come to similar conclusions as the authors...

- Maybe related to this lack of references, some assertions are given with too few justifications, that should come either from experimental observations or from previous works. This should be corrected, and the authors could specify that they are making hypotheses when there is no existing justifications.

- This work does not contain any significant novelty, but provides more detailed and accurate observations at the microstructure scale compared to previous (old!) measurements performed by Jacka and co-authors for instance.
Compared to the extensive literature about dynamic recrystallization at hot temperature (see for instance Humphreys and Haterly 2001 or 2004), there is no novelty, and this literature should be mentioned, especially within the discussion, in order to help the interpretation of the results.

- The high quality observations enable to assert more clearly some mechanisms as important in the case of recrystallization in ice as, for instance, the fact that at low temperature, intracrystalline rotation will prevail on GBM and therefore induces texture that are closer to the one observed along deep ice cores.

- It is not clear, all over the text,why the authors want or need to mention GBS as an impacting mechanisms since the experiments performed show absolutely no proof of it, neither in macroscopic data (dependance of peak stress on grain size for instance), nor in microscopic observations. The only observation of small grain necklaces (but limited in number) at the lowest temperature, and a weaker texture in this small grain population is not sufficient, to my point of view, to assert the occurrence of GBS. It could be mentioned as one of the hypothesis among others, but not come to the conclusion as the mechanism at play. In particular, the use of GBS is not necessary to explain stress weakening and does not appear coherent with the results.

- I raise again the point about the lack of proper explanation concerning the measurement of subgrain size in the specific case the presented experiments, since the figures shown do not reveal any proper subgrain structure that could be characterized by a dimension (as a mean size for instance). Since different conclusion are taken out of this subgrain size evaluation, it should be corrected before any publication.

- the authors make no use of their observations from the WBV method neither in the discussion, nor in the conclusion... Should it remain in the paper?

As a conclusion, I suggest that this paper should not be published before the authors first enhance the justifications of some strong hypotheses they are making in order to interpret their observations, and second highlight the novelty of their observations and results regarding already existing work.

Ref:

M. F. Ashby and R.A. Verrall. Diffusion-accommodated flow and superplasticity. Acta Metallurgica, 1973

[1] T. Chauve, M. Montagnat, F. Barou, K. Hidas, A. Tommasi, and D. Mainprice. Investigation of nucle- ation processes during dynamic recrystallization of ice using cryo-ebsd. Philosophical Transactions of the Royal Society A: Mathematical, Physical and Engineering Sciences, 375(2086), 12 2017.

[1] P. Duval, M. Ashby, and I. Anderman. Rate controlling processes in the creep of polycrystalline ice. J. Phys. Chem., 87(21):4066–4074, 1983.

[1] F. Humphreys. Nucleation in recrystallization. Materials Science Forum, 467-470:107–116, 2004.

[1] F. J. Humphreys and M. Hatherly. Recrystallization and related annealing phenomena. Pergamon, Oxford, Second edition, 2004.

[1] F. Grennerat, M. Montagnat, O. Castelnau, P. Vacher, H. Moulinec, P. Suquet, and P. Duval. Exper- imental characterization of the intragranular strain field in columnar ice during transient creep. Acta Materialia, 60(8):3655–3666, 5 2012.

[1] T. H. Jacka and M. Maccagnan. Ice crystallographic and strain rate changes with strain in compression and extension. Cold Reg. Sci. Technol., 8:269–286, 1984.

[1] B. Journaux, T. Chauve, M. Montagnat, A. Tommasi, F. Barou, D. Mainprice, and L. Gest. Recrystal- lization processes, microstructure and crystallographic preferred orientation evolution in polycrystalline ice during high-temperature simple shear. The Cryosphere, 13(5):1495–1511, 05 2019.

[1] T. Sakai, A. Belyakov, R. Kaibyshev, H. Miura, and J. J. Jonas. Dynamic and post-dynamic recrys- tallization under hot, cold and severe plastic deformation conditions. Progress in Materials Science, 60(0):130–207, 3 2014.

[1] E. M. Schulson and P. Duval. Creep and Fracture of Ice. Cambridge University Press, 2009.

---

## Referee Comment (RC2) · Olivier Castelnau (Referee) · 2 Mar 2020

This paper essentially presents a nice set of experimental data on polycrystalline ice specimens (synthetic specimens) deformed at constant strain-rate under uniaxial compression, at -10°C, -20°C and -30 °C. Besides mechanical tests, the authors provide a detailed analysis of the microstructure of ice grains and its evolution, largely based on EBSD performed in a dedicated scanning electron microscope. Grain size, grain morphology, spatial grain distribution, intragranular misorientation, statistical grain orientation (crystallographic texture) are investigated with respect to temperature and strain

(up to 20%). The authors want to put the focus on the effect of grain size on the rheology, as state in the last line of the abstract that can be considered as a summary of the findings of this study : "Grain size reduction, which can be observed in all deformed samples, is most likely to cause weakening (enhancement) and should be considered to have a significant control on the rheology of natural ice flow".

I suggest rejecting this paper for the following reasons :

** All along the paper, the authors state that grain boundary sliding (gbs) must be invoked to explain the observations. For my point of view, there is absolutely no proof here that gbs has been activated, even in the "small grains". The authors observe some correlations between grain size distribution, temperature, texture, but whether gbs is necessary to explain all that is another story. The evidences prone by the authors are highly speculative. The mechanical tests (figure 2) essentially show a dominant temperature effect (known since the early years of glaciology – do the associated activation energy, not calculated here, matches literature data ?) and a softening at strain larger than $\sim$0.03. There is no data in this paper relating rheology with grain size, although grain size effect is presented as a major conclusion. Authors try to explain that gbs is necessary using arguments based on microstructure evolution. But many other parameters coming in play should be also considered, and mostly those associated with recrystallization (for which the micrographs show direct evidences unlike gbs) such as gbm rate, nucleation rate, stored energy, etc and their evolution with temperature and strain for which our actual knowledge is very limited. To prove that gbs has been active in the specimen, I would suggest the authors to (i) provide direct evidence of a sliding boundary and/or (ii) show that the associated viscosity is compatible with the one of the specimen (as gbs in a polycrystalline aggregate required associated diffusion, which is slow) and/or (iii) model microstructure evolution due to deformation + dynamic recrystallization to show that the observed evolution cannot be explained by these only mechanisms.

** Along the same line, the sentence (p12 line 2) "gbs is kinematically required for all

grain size sensitive mechanisms" is incorrect. For example, the Hall-Petch mechanism is largely used in metallurgy to explain size effect observed in many nanometric grains metallic alloys. Hall-Petch is based on the mean free path of dislocations, it explain very well many observations, and does not require any other mechanisms than dislocation glide (no gbs!). Could the mean free path of mobile dislocations have an influence of ice rheology at low temperature ?

** Similarly, about the sentence (p15 line 6) "similarly, we suggest that grain size sensitivity of gbs favours a faster strain rate in small grains": I find no fact in the results supporting this assertion. Strain-rate in various sets of grains is not measured nor estimated here. And also, in section 4.2, the authors make a correlation between the softening observed at -30degC and the grain size, and conclude that the observed softening should likely be attributed to gbs. Gbs could be a possibility, but among many others. For example, what do we know about the density of mobile dislocations ?? If it increases, the stress would decreases as observed. Increase of dislocation density is often used to explain the peak stress for materials with low initial dislocation density (eg. Si, . . .).

**The statistical relevance of the performed mechanical tests and/or microstructural investigations can also be questioned. Figures 9, 10, 11, 13 show pole figures that do not, by far, exhibit the expected transverse isotropy (expected since the initial specimen are thought to exhibit random CPO with equiaxe grain shape, and since uniaxial compression is transverse isotropic). This severe lack of symmetry in the observed microstructure can originate from (i) initial samples that do not exhibit a random microstructure and/or (ii) mechanical tests that deviate from uniaxial compression (there could be many reasons for that) and/or (iii) the microstructure is not analysed on a sufficiently large material volume (volume smaller than the Representative Volume Element -RVE). Consequently, the global picture shown here (ex. texture strength as function of temperature, which is an interesting result) are probably correct, but I don't think that, with the results shown, authors can dig deeper into the interpretation of active deformation mechanisms. If the lack of texture symmetry is present in the specimens, then the applied axial strain-rate would generate significant shear stress (or shear-rate, depending on the experimental boundary conditions), affecting of course the texture and microstructure evolutions (so-called out-of-axis tests). Is there any connection with the large spread observed on the mechanical responses (figure 2) ? For example the peak stress at -10°C varies by almost a factor 2, which is considerable and should be discussed. One could expect some associated spread in the microstructure...

** The discussion in this paper relies on a separation of the grain size distribution between "small grains" and "large grains", invoking a "bimodal" (p7 line 4) grain size distribution. In figures 3, 4, 5, I do not see any bimodal grain size distribution, but rather a unimodal one with a long tail. Therefore the size threshold (p7 line 16) used to separate small and large grains is completely ad hoc, and I am not sure about the effect of this particular choice on the provided discussion. I also don't understand why the authors state that "The small grains are likely include all the recrystallized grains" (p7 line 19, p8 line 8, . . .) as (i) if GBM occurs, recrystallization can also lead to large grains and (ii) the grain size distribution of the initial microstructure is not shown.

** The discussion also largely relies of the size of subgrains. However, in figures 3, 4, 5, it is really hard to identify those subgrains in most of the grains. For example in figure 5 at 20% strain, one only sees some disconnected segments in the WBV map in the large yellow or pink-orange grains at the bottom of the micrograph. How do the authors identity the subgrains and calculate their size in such a case ?

Others remarks :

** This experimental study cannot be used without very special care to infer deformation mechanisms occuring in "terrestrial and planetary ice flow" (1st abstract line), as (i) the grain size investigated ($\sim$200 microns) is one order of magnitude smaller than the natural one, and (ii) the strain-rate used during the mechanical tests (10-5s-1) is 3 to 6 orders of magnitude larger than in cold regions of ice sheets.

[Figure]

\*\* I wonder whether there is no damage occurring at the high strain-rate considered, particularly at the smaller temperatures ?

\*\* p1 line 16 : "displacement rate" instead of "displacement"

\*\* p1 line 26 : invoking creep stages (secondary, tertiary) for the description of constant strain-rate experiments is misleading.

\*\* P5 line 26, I don't understand what is meant with "The CPO data were contoured with half-width of 7.5deg" ?

\*\* p7 line 26 : to the best of my knowledge, recovery, subgrain rotation and gbm are not deformation mechanisms ! If recovery and/or gbm are initiated, the specimen will not deform.

\*\* eq. 3, how is R (grain radius) estimated for non-spherical grains ??

\*\* p9, line 1 : I think that calling "m" the 10-10 direction is not standard (m-axes pole figures). Should be clarified ?

\*\* p11 line 26, the sentence "Much of the stress increase prior to peak stress relates to elastic strain" is wrong. First of all, there is no known yield stress for the high temperature rheology of ice, i.e. plastic strain starts as soon any stress is applied, as here in the first part of the loading prior to the peak stress. There are old published data (on single and polycrystals) showing that the initial slope depends on the strain-rate. Of course, there is always an elastic strain associated to the applied stress (Hooke's law). On top of that, the measured slope ($\sim$1GPa) very probably also accounts for the way strain is measured experimentally: if it is not measured directly on the specimen (eg. with an extensometer or strain-gage), it is well known that very small modulus are obtained, due to machine rigidity and other artefacts.

\*\* p 14 line17: why should there be an "acceleration of grain rotation rate due to intracrystalline basal glide" since the overall strain-rate (prescribed) is constant ?

[Figure]

** figures 3, 4, 5 : If I understand (this is not clear in the paper), the shown grain size distributions indicate the number of grains at a given size. It would be more instructive to show the volume fraction, not the number of grains, as the rheology is associated with the volume average of grain deformation.

** figure 14 is interesting, as it shows that the strain-rate seems to have little effect. To my understanding, this is not expected for thermally activated mechanisms such as recrystallization, where time comes in plays. This figure could be more largely discussed, to my point of view.

---

## Author Comment (AC2) · 15 Jun 2020

We thank Reviewer 2 for his thoughtful and helpful review of our paper. The comments have helped us improve the manuscript significantly. Our reply to reviewer comprises two parts: (1) some short general statements and (2) point-by-point reply to comments from reviewer. Please refer to supplement PDF for point-to-point reply.

Section one: general statements

1. This work contains data which are completely new. We would like to thank the

reviewer for one particular comment: "This paper essentially presents a nice set of experimental data.... the authors provide a detailed analysis of the microstructure of ice grains and its evolution...". We would like to emphasize that the sequence of microstructures and CPOs developed with increasing strain has not been documented before for ice deformed at cold temperatures (-20, -30 °C).

2. The reviewer suggests rejecting this paper mainly because the interpretation of grain boundary sliding (GBS). In our view, interpretations are not usually what make a scientific good paper. New data that is factually correct and will stand the test of time make a good paper. It is likely that the interpretations will change in the future as researchers gain new data or insight. We accept that the factual observations that we present and then to infer GBS could be interpreted in different ways. In the revision, we include some alternative interpretations (including "spontaneous" nucleation) of the data, with some discussion of the merits and drawbacks of each of these interpretations. We hope that we have kept the observations and interpretations clearly separated and we have reduced the emphasis on our preferred interpretation of GBS. We have also identified some of the tests that may facilitate distinguishing these different interpretations in the future. Some more details are included in answers to specific points.

The reviewer's comments highlight that our original manuscript did not really make clear that we do interpret intracrystalline dislocation glide that causes lattice rotation as one of the key processes controlling CPO development. We hope that we have made this much clearer in the revised manuscript. The operation of a GBS process, if this is correct, would be additional to the role of intracrystalline dislocation glide and associated recovery and recrystallisation processes.

Please also note the supplement to this comment:
https://www.the-cryosphere-discuss.net/tc-2020-2/tc-2020-2-AC2-supplement.pdf

**Supplement:**

**Response to Reviewer 2**

We thank Reviewer 2 for his thoughtful and helpful review of our paper. The comments have helped us improve the manuscript significantly. Our reply to reviewer comprises two parts: (1) some short general statements and (2) point-by-point reply to comments from reviewer. *Reviewers comments are in blue type. Extracts from our revised manuscript are in italics.*

**Section one: general statements**

**1. This work contains data which are completely new.**

We would like to thank the reviewer for one particular comment: *"This paper essentially presents a nice set of experimental data…. the authors provide a detailed analysis of the microstructure of ice grains and its evolution…".* We would like to emphasize that the sequence of microstructures and CPOs developed with increasing strain has not been documented before for ice deformed at cold temperatures (-20, -30 °C).

**2. The reviewer suggests rejecting this paper mainly because the interpretation of grain boundary sliding (GBS).**

In our view, interpretations are not usually what make a scientific good paper. New data that is factually correct and will stand the test of time make a good paper. It is likely that the interpretations will change in the future as researchers gain new data or insight. We accept that the factual observations that we present and then to infer GBS could be interpreted in different ways. In the revision, we include some alternative interpretations (including "spontaneous" nucleation) of the data, with some discussion of the merits and drawbacks of each of these interpretations. We hope that we have kept the observations and interpretations clearly separated and we have reduced the emphasis on our preferred interpretation of GBS. We have also identified some of the tests that may facilitate distinguishing these different interpretations in the future. Some more details are included in answers to specific points.

The reviewer's comments highlight that our original manuscript did not really make clear that we do interpret intracrystalline dislocation glide that causes lattice rotation as one of the key processes controlling CPO development. We hope that we have made this much clearer in the revised manuscript. The operation of a GBS process, if this is correct, would be additional to the role of intracrystalline dislocation glide and associated recovery and recrystallisation processes.

**Section two: point-by-point reply to comments**

1. All along the paper, the authors state that grain boundary sliding (gbs) must be invoked to explain the observations. For my point of view, there is absolutely no proof here that gbs has been activated, even in the "small grains". The authors observe some correlations between grain size distribution, temperature, texture, but whether gbs is necessary to explain all that is another story. The evidences prone by the authors are highly speculative. Authors try to explain that gbs is necessary using arguments based on microstructure evolution. But many other parameters coming in play should be also considered, and mostly those associated with recrystallization (for which the micrographs show direct evidences unlike gbs) such as gbm rate, nucleation rate, stored energy, etc and their evolution with temperature and strain for which our actual knowledge is very limited. To prove that gbs has been active in the specimen, I would suggest the authors to (i) provide direct evidence of a sliding boundary and/or (ii) show that the associated viscosity is compatible with the one of the specimen (as gbs in a polycrystalline aggregate required associated diffusion, which is slow) and/or (iii) model microstructure evolution due to deformation + dynamic recrystallization to show that the observed evolution cannot be explained by these only mechanisms.

The key objective of this paper is to report the detailed changes in microstructures and CPOs to progressively higher strains at low and high temperatures, with the very new data being at lower temperatures. The interpretation of GBS is not central to this and we have downplayed that in the revised manuscript. We still wish to explain the weakening of CPOs in finer grain sizes and have presented two alternative interpretations; GBS and spontaneous nucleation.

We agree with reviewer's comment that our data cannot prove the existence of grain boundary sliding. We would love to have fiducial marker evidence to show directly the GBS effect (e.g.(Eleti et al., 2020; Schmid et al., 1977; Spiers, 1979): this is a significant technical challenge for now. The particular set of experiments presented in our paper does not include variable initial grain size. However, comparable experiments do show grain size sensitivity. The set of -10 °C experiments published by Qi et al 2017 have two different initial grain sizes. A plot of strain rate against the peak stresses (Fig. 3, copied below as Fig. R2.1) shows two different best fit lines for the two initial grain sizes. At peak stress (~ equivalent to min strain rate) grain size is unlikely to have changed substantially from the starting material (and we have some new experiments to peak stress only that show this to be correct). The easiest interpretation of the Qi et al (2017) mechanical data is that there is grain size sensitivity, which is consistent with the operation of GBS.

[Figure]

**Figure R2.1.** Plot of strain rate versus stress on logarithmic scales using data from Qi and others (2017).

If GBS does occur, our interpretation is that this is in addition to dislocation glide (Drury et al., 1985; Gifkins, 1976, 1977; Goldsby and Kohlstedt, 1997, 2001; Hirth, 2002; Hirth and Kohlstedt, 2003; Kuiper et al., 2019a; Kuiper et al., 2019b; Langdon, 2006, 2009; Warren and Hirth, 2006). Some authors term this dislocation accommodated GBS or "disGBS". In this mechanism, the total strain rate is the addition of a dislocation process (that changes crystal shapes and causes lattice rotation and internal distortion) and a GBS process that is probably controlled by a "viscous" mechanism within grain boundaries (small path length diffusion and/ or asperity plasticity: idea originally from (Gifkins, 1976)). This is not the same as diffusion creep, irrespective of whether that is controlled by lattice diffusion (Nabarro-Herring creep) or grain boundary diffusion (Coble creep). GBS is required as an accompanying mechanism to polycrystalline diffusion creep, but in that case grain shape change is facilitated by the diffusive mass transfer process. In diffusion creep, grain size sensitivity comes primarily from the increased path length for diffusion meaning that the change of shape of bigger grains takes longer. In "disGBS" the GBS itself is the prime source of grain size sensitivity. If there is a "viscous" grain boundary volume then the rheology will depend on the volume proportion of the sample that comprises grain boundaries: this proportion will increase with decreasing grain size.

CPO models certainly do not match observations fully for shear (see discussion in Qi et al., 2019) and that paper speculates that GBS may bridge the gap between the results of laboratory experiments and numerical models. Indeed, there is currently a major effort (led by Sandra Piazolo and colleagues) among the community that use the ELLE modelling platform to incorporate GBS: a difficult task. Microstructural modelling is beyond the scope of our paper.

2. The mechanical tests (figure 2) essentially show a dominant temperature effect (known since the early years of glaciology – do the associated activation energy, not calculated here, matches literature data ?) and a softening at strain larger than ~0.03.

Yes, the mechanical data match literature data. We have added a calculation of activation energy to the supplementary information and have referred to this in the text. Best fit to all data (-10, -20 and -30 °C) give activation enthalpies of 98 kJ/mol and 103 kJ/mol from peak and final stress data assuming n=3 and 131 kJ/mol and 138 kJ/mol from peak and flow stress data assuming n=4. These values are close to reported Q values of 71-124 kJ/mol (-5 °C- -30 °C) from Budd and Jacka (1989) and ~133 kJ/mol (-1.5 °C- -12.8 °C) from Glen (1955) and 64-250 kJ/mol from Kuiper and others (2019a, 2019b). Note experiments in this study only cover three temperature values. Hence, the calculated Q values are prone to error. More data points are needed for a more accurate Q investigation.

3. Along the same line, the sentence (p12 line 2) "gbs is kinematically required for all grain size sensitive mechanisms" is incorrect. For example, the Hall-Petch mechanism is largely used in metallurgy to explain size effect observed in many nanometric grains metallic alloys. Hall-Petch is based on the mean free path of dislocations, it explain very well many observations, and does not require any other mechanisms than dislocation glide (no gbs!). Could the mean free path of mobile dislocations have an influence of ice rheology at low temperature ?

Our apologies; the reviewer is correct. That statement does not apply to the full breadth of GSS mechanisms including classic Hall-Petch and also mechanical twinning (Rowe and Rutter, 1990) and we have removed the statement.

As an aside there is a very interesting ongoing discussion of the Hall-Petch (Weertman, 1993) relationship (with strength increasing with grain size) and the inverse Hall-Petch relationship

(Masumura et al., 1998) in the materials science literature (Pande and Cooper, 2009; Ryou et al., 2018; Sheinerman et al., 2020). Modelling of the inverse Hall-Petch relationship requires coupling of GBS to intragranular dislocation activity (Carlton and Ferreira, 2007; Ehre and Chaim, 2008; Padmanabhan et al., 2007; Padmanabhan et al., 2014; Ryou et al., 2018; Sheinerman et al., 2020) and the relationships are not very different to those described elsewhere as GBS accommodated by dislocation creep (Goldsby and Kohlstedt, 1997; Hansen et al., 2011; Langdon, 2006, 2009). In minerals, the normal Hall-Petch relationship (increasing strength with decreasing grain size) has only been documented at low homologous temperatures (Hansen et al., 2019; Koizumi et al., 2020) whereas weakening with reduced grain size is the norm at higher temperatures and lower stresses (Brodie and Rutter, 2000; De Bresser et al., 2001; Hiraga et al., 2013; Hirth, 2002; Hirth and Kohlstedt, 2003; Schmid et al., 1977; Ter Heege et al., 2005; Walker et al., 1990). Materials science work defines a material-dependent threshold grain size, above which the Hall-Petch relationship holds and with the inverse Hall-Petch relationship at grain sizes below the threshold (Pande and Cooper, 2009; Ryou et al., 2018). Recent work suggests that the threshold moves to larger grain sizes at lower strain-rates or stresses (Somekawa and Mukai, 2015). The rates that are considered very slow in these metallurgical analysis (e.g. $1 \times 10^{-4}$ s$^{-1}$) are very fast in the context of geological or glaciological laboratory experiments and this may explain why we only see evidence of the Hall-Petch effect at low homologous T. Some recent work relates GBS associated with the inverse Hall-Petch relationship with amorphization of the grain boundaries (Guo et al., 2018) and a molecular dynamics modelling study of ice (Cao et al., 2018) generates an inverse Hall-Petch relationship that involves a combination of GBS, grain rotation, amorphization and recrystallization, phase transformation, and dislocation nucleation in both bicrystals and polycrystals.

4. Similarly, about the sentence (p15 line 6) "similarly, we suggest that grain size sensitivity of gbs favours a faster strain rate in small grains": I find no fact in the results supporting this assertion. Strain-rate in various sets of grains is not measured nor estimated here. And also, in section 4.2, the authors make a correlation between the softening observed at -30degC and the grain size, and conclude that the observed softening should likely be attributed to gbs. Gbs could be a possibility, but among many others. For example, what do we know about the density of mobile dislocations ?? If it increases, the stress would decreases as observed. Increase of dislocation density is often used to explain the peak stress for materials with low initial dislocation density (eg. Si, . . .).

Please see our answer to point 1.

5. The statistical relevance of the performed mechanical tests and/or microstructural investigations can also be questioned. Figures 9, 10, 11, 13 show pole figures that do not, by far, exhibit the expected transverse isotropy (expected since the initial specimen are thought to exhibit random CPO with equiaxe grain shape, and since uniaxial compression is transverse isotropic). This severe lack of symmetry in the observed microstructure can originate from (i) initial samples that do not exhibit a random microstructure and/or (ii) mechanical tests that deviate from uniaxial compression (there could be many reasons for that) and/or (iii) the microstructure is not analysed on a sufficiently large material volume (volume smaller than the Representative Volume Element-RVE). Consequently, the global picture shown here (ex. texture strength as function of temperature, which is an interesting result) are probably correct, but I don't think that, with the results shown, authors can dig deeper into the interpretation of active deformation mechanisms. If the lack of texture symmetry is present in the specimens, then the applied axial strain-rate would generate significant shear stress (or shear-rate, depending on the experimental boundary conditions), affecting of course the texture and microstructure evolutions (so-called out-of-axis tests). Is there any connection with the large

spread observed on the mechanical responses (figure 2) ? For example the peak stress at -10 C varies by almost a factor 2, which is considerable and should be discussed. One could expect some associated spread in the microstructure.

The reviewer is correct about symmetric incompleteness and we have added the following text to address this: *"Many deformed samples exhibit an incompleteness of c-axes cone (lack of cylindrical symmetry) (Fig. 8-10). The incompleteness of c-axes cone is more severe for 5 µm EBSD maps collected from a much smaller area than 30 µm EBSD maps (Fig. 12). These phenomena are common to all ice CPOs from measurements on a single sample planes (by EBSD or optical methods: see any of the papers cited), but are not so apparent in neutron diffraction data (Piazolo et al., 2013; Wilson et al., 2019), that sample a larger volume, suggesting that a single plane through a deformed sample does not generally contain sufficient grains for a fully representative CPO."*

The fact that neutron diffraction data gives CPOs that have close to the cylindrical symmetry, for samples that have fewer grains (initially) in an average cross section (Piazolo et al 2013 initial grain size 0.5mm whereas ours <0.3mm: samples in both cases 1 inch diameter) suggests that the sample as a whole has enough grains to b considered mechanically isotropic. In this case the incompleteness of CPOs is an analytical sampling issue and should not impact on mechanical data. A good example of where samples contain too few grains to be considered isotropic is the re-deformation of natural ice with a 20mm grain size (Craw et al., 2018): this gives rise to significant inconsistency in stress strain curves, although yield stress data correlate sensibly with strain rates.

The scatter of peak stress values we have is fairly typical of confined medium constant displacement rate experiments (data for comparison can be extracted from (Durham et al., 1983; Golding et al., 2020). Unconfined constant displacement rate experiments (Hammonds and Baker, 2016; Vaughan et al., 2017) have less variability and it is likely that some of the scatter in confined medium experiments relates to how stable the confining pressure is. Unconfined creep experiments (constant load) also show a range of minimum strain rates for a set of experiments at the same stress (Journaux et al., 2019; Montagnat et al., 2015; Treverrow et al., 2012). To compare constant rate vs constant load experiments, we can calculate the "viscosity" at peak stress/ minimum strain rate. Confined constant rate and unconfined creep tests both have "viscosities" that vary by up to about 2x for experiments at the same rate or stress. Unconfined constant rate experiments have peak stress "viscosities" that vary by up to about 1.1x. These statements are made on a relatively small data set as there seem to be few "repeat" experiments (in terms of load or rate) in the published literature. At the moment we don't have a full explanation as to what controls this variability. We have to account for the variability in studies where it becomes important (e.g. for calibrating flow laws). In this paper it is not so important and the aspect that is important to us – the curve shape with a peak stress followed by weakening is common to all experiments.

6. The discussion in this paper relies on a separation of the grain size distribution between "small grains" and "large grains", invoking a "bimodal" (p7 line 4) grain size distribution. In figures 3, 4, 5, I do not see any bimodal grain size distribution, but rather a unimodal one with a long tail. Therefore the size threshold (p7 line 16) used to separate small and large grains is completely ad hoc, and I am not sure about the effect of this particular choice on the provided discussion. I also don't understand why the authors state that "The small grains are likely include all the recrystallized grains" (p7 line 19, p8 line 8, . . .) as (i) if GBM occurs, recrystallization can also lead to large grains and (ii) the grain size distribution of the initial microstructure is not shown.

(1) We modified the description of grain size distribution in section 3.3.2: *"For samples deformed to ~3% strain, the grain size distributions are strongly skewed or possibly bimodal, with a clear main peak at finer grain sizes and a tail of coarser sizes with a broad, poorly defined secondary peak corresponding to the mean grain size of the starting material (Fig. 4(d), 5(d) and 6(d))."*

(2) We removed the statement of: "The small grains are likely include all the recrystallized grains." This is a very good point from the reviewer. We (who come from the rock deformation world) sometimes forget that at the high homologous temperatures in ice recrystallised grains can grow to a large size. In much lower homologous temperature experiments in quartz, for example, it is reasonable that recrystallised grains are small and remnant grains large (see for example (Cross et al., 2017; Hirth and Tullis, 1992). We still wish to segment the grain size on the basis of "big" and "small" grains and we hope that our presentation of this is now more robust and does not assign arbitrarily the status recrystallised or remnant on certain grain size populations. The precise threshold we use does not influence the difference in CPOs between "big" and "small" grains as shown in Fig. R2.2, extracted from new supplementary information.

[Figure]

**Figure R2.2**. The contoured *c*-axis CPOs of "big" and "small" grains in samples deformed at **(a)** -10, **(b)** -20 and **(c)** -30 °C to ~12% strain. "Big" and "small" grains are separated using the threshold of mean grain size (row 1), SMR (square mean root) grain size (row 2), median grain size (row 3) and peak grain size (row 4). Number of grains and M-index value are marked at the bottom left corner of the corresponding *c*-axis CPO.

7. The discussion also largely relies of the size of subgrains. However, in figures 3, 4, 5, it is really hard to identify those subgrains in most of the grains. For example in figure 5 at 20% strain, one only sees some disconnected segments in the WBV map in the large yellow or pink-orange grains at the bottom of the micrograph. How do the authors identity the subgrains and calculate their size in such a case ?

We agree with the observation from the reviewer that suggests many of the subgrain boundaries are straight tilt bands or kink bands. The subgrain structure was revealed by Weighted Burgers vector (WBV) method, which picks up pixels with the WBV magnitude ($\|\mathbf{WBV}\|$) higher than 0.0026 $\mu m^{-1}$ (equivalent to misorientation angle between neighbouring pixels higher than ~0.7°). Therefore, many of the subgrain boundaries lower than 2° were selected and they might

contain non-neglectable errors (Prior, 1999). Moreover, we didn't make it clear that the measurement of subgrain sizes were not based on the data of WBV, and they were based on the misorientation between adjacent pixels. Therefore, the new subgrain boundary plots corresponds to the original subgrain calculations. We kept the WBV analyses based on the thinking that they might contain more information for further comparison. But the WBV analyses have now been removed completely from this paper.

The new maps (Fig. 4(c), 5(c), 6(c)) that show subgrain boundaries correspond to the much simpler misorientation threshold. We modified statements on section 3.3.1 to: "Distinct sub-grain boundaries can be observed in all the samples (Fig. 4 (c), 5 (c) and 6 (c)). Many of the subgrain boundaries appear to be straight, some with slight curvature. A small number have strong curvature. Interconnected subgrain boundaries can be observed in some of the grains. Subgrain boundaries subdivide grains into subgrains."

8. This experimental study cannot be used without very special care to infer deformation mechanisms occuring in "terrestrial and planetary ice flow" (1st abstract line), as (i) the grain size investigated (~200 microns) is one order of magnitude smaller than the natural one, and (ii) the strain-rate used during the mechanical tests (10-5s-1) is 3 to 6 orders of magnitude larger than in cold regions of ice sheets.

We modified the first line in abstract to: "To understand better the ice deformation mechanisms…" The reviewer raises the key problem that we struggle with, when we are working in the laboratory with application to natural ice. The absolute fastest documented natural terrestrial strain rates are in lateral shear margins $\sim 10^{-9}$ s$^{-1}$ (Bindschadler et al., 1996; Jackson and Kamb, 1997). Rates in basal ice is harder to estimate; most models would have strain rate maxima also around $\sim 10^{-9}$ s$^{-1}$. Most parts of ice sheets and glaciers have strain rates that are up to 2 orders of magnitude slower than this. To run an experiment from to 10% strain (i.e something that may go from isotropic starting material to a "steady state" microstructure) will take three years at $\sim 10^{-9}$ s$^{-1}$. (Jacka and LI, 2000) did an amazing job running experiments for long durations at low rates (down to $4 \times 10^{-10}$ s$^{-1}$) but these are really the only experiments that achieve substantial strain at "natural" rates. Specific aspects of ice mechanics have been assessed by deforming natural samples to small strains (<1%) in the lab at relatively slow rates ($10^{-10}$ s$^{-1}$ to $10^{-8}$ s$^{-1}$) (Castelnau et al., 1998; DahlJensen et al., 1997; Jackson and Kamb, 1997). In general, it is virtually impossible to work at natural rates and we have to develop scaling relationships that involve strain rate, temperature and grain size.

9. I wonder whether there is no damage occurring at the high strain-rate considered, particularly at the smaller temperatures?

Stress-strain curves of all experimental runs show a smooth and continuous increase of stress as a function of strain before reaching the peak (Fig. 3). The stress-strain curves of experiments with a development of cracking during deformation normally show an initial yield point before reaching the peak stress (Mellor and Cole, 1982). The initial yield point is interpreted as a reflection of cracking on the mechanical data (Mellor and Cole, 1982). Such yield point is not observed in any of the experiments in this study.

The chief purpose of the confining pressure in these experiments is to suppress brittle phenomena including cracking and frictional sliding. Fig. R2.4 shows the experiment with the highest differential stress, plotted on a Mohr diagram for stress. The green circle shows the shear and normal stresses for surfaces of all orientations and the maximum ($\sigma_1$) and minimum ($\sigma_3 = \sigma_2$ = confining pressure) plot along the line of zero shear stress. Superposed are two failure envelopes. One is a Coulomb (frictional sliding) envelope using the friction coefficient

for ice-ice sliding from (McCarthy et al., 2017). Coulomb envelopes usually underestimate brittle strength. The second failure envelope is the composite envelope from (Beeman et al., 1988). Red and blue Mohr circles show the stress states needed for brittle failure at 20MPa pressure with each of these envelopes. Maximum differential stresses applied in our experiments are substantially below those for needed for brittle failure.

[Figure]

**Figure R2.3**. Typical stress-strain curve for deformed sample with cracking (from Mellor and Cole, 1982)

[Figure]

**Figure R2.4**. Mohr diagram showing stress state of sample PIL164 (the largest differential stress) in green. A coulomb failure envelope using a friction coefficient of 0.29 from (McCarthy et al., 2017) is shown with a red dashed line and the Mohr circle for failure at 20MPa confining pressure is shown in red. The blue lines show the (Beeman et al., 1988) failure envelope from and the Mohr circle for failure at 20MPa confining pressure is shown in blue.

10. p1 line 16 : "displacement rate" instead of "displacement"

Corrected.

11. p1 line 26 : invoking creep stages (secondary, tertiary) for the description of constant strain-rate experiments is misleading.

We have deleted these misleading wording.

12. P5 line 26, I don't understand what is meant with "The CPO data were contoured with half-width of 7.5deg" ?

We modified the statement in section 2.5.2: "The CPO data were contoured with a half-width of 7.5° based on the maximum of multiples of a uniform distribution (MUD) of the points, to more clearly show the CPO patterns."

12. p7 line 26 : to the best of my knowledge, recovery, subgrain rotation and gbm are not deformation mechanisms ! If recovery and/or gbm are initiated, the specimen will not deform.

Thank appreciate the reviewer for pointing out this mistake. We have removed this sentence since we removed boundary hierarchy analyses.

13. eq. 3, how is R (grain radius) estimated for non-spherical grains ??

We have removed grain boundary lobateness analyses.

14. p9, line 1 : I think that calling "m" the 10-10 direction is not standard (m-axes pole figures). Should be clarified ?

We corrected "*m*-axes" to "poles to the *m*-planes".

15. p11 line 26, the sentence "Much of the stress increase prior to peak stress relates to elastic strain" is wrong. First of all, there is no known yield stress for the high temperature rheology of ice, i.e. plastic strain starts as soon any stress is applied, as here in the first part of the loading prior to the peak stress. There are old published data (on single and polycrystals) showing that the initial slope depends on the strain-rate. Of course, there is always an elastic strain associated to the applied stress (Hooke's law). On top of that, the measured slope (~1GPa) very probably also accounts for the way strain is measured experimentally: if it is not measured directly on the specimen (eg. with an extensometer or strain-gage), it is well known that very small modulus are obtained, due to machine rigidity and other artefacts.

intracrystalline dislocation slip, the porosity loss being very likely negligible.

Published literature labelled the stress increase prior to peak stress in constant displacement rate experiments as: "normally elastic" (Cole, 1987) and "quasi-elastic" (Kirby, 1987). The deceleration during primary creep in constant stress experiments was interpreted as effected by a "delayed elasticity", with a recoverable component of time-dependent elastic strain and an irrecoverable viscous strain (Mellor and Cole, 1982), and "anelasticity" (Duval et al., 1983). The reason we chose to describe the behaviour as substantially elastic is that we have other experiments where we can show that this part of the deformation is recoverable. However, these other experiments are higher rate experiments with slopes on the stress strain curve approaching the 9GPa modulus. The reviewers are correct in pointing out that in the experiments presented in this paper the slope is substantially below modulus and the behaviour is not substantially elastic. We have modified the statement in section 4.1.1: "*This likely includes anelastic deformation related to intergranular stress redistribution used to explain primary creep in constant load experiments (Duval et al, 1983). The curvature of the stress strain line at the start of each experiment may relate to initial porosity loss as suggested by rapid increases in ultrasonic p-wave velocity in comparable experiments by Vaughan et al., (2017).*"

17. figures 3, 4, 5 : If I understand (this is not clear in the paper), the shown grain size distributions indicate the number of grains at a given size. It would be more instructive to show the volume fraction, not the number of grains, as the rheology is associated with the volume average of grain deformation.

Grain size distribution has been used to show generation of small grains after deformation. These grains are not observed in undeformed grains. We estimated grain volume for each grain

size class for modelling the effect of small grains on mechanical weakening. These grain volume data are subject to another paper.

18. figure 14 is interesting, as it shows that the strain-rate seems to have little effect. To my understanding, this is not expected for thermally activated mechanisms such as recrystallization, where time comes in plays. This figure could be more largely discussed, to my point of view.

We plotted data from this study and previous studies in a diagram of $\theta$ as a function of strain with data subdivided with different temperatures and strain rates to increase our understanding of the processes that might control the c-axes cone opening-angle (Table 4 and Fig. 13). The relation to strain rate within the broader data set in this figure is not very clear, because for any given temperature there is not a big range in strain rate. The exception is the data set plotted from (Qi et al., 2017) at -10 °C and ~ 20% strain which does show a rough decrease in $\theta$ as strain rate (or stress) increases (See Qi et al., 2017 fig 9. This fits with the Zener-Hollomon concept (Zener and Hollomon, 1944) that suggests that decreasing strain rate will have an equivalent effect to increasing temperature.

**References**

Beeman, M., Durham, W. B. and Kirby, S. H.: Friction of ice, Journal of Geophysical Research: Solid Earth, 93(B7), 7625–7633, https://doi.org/10.1029/JB093iB07p07625, 1988.

Bindschadler, R., Vornberger, P., Blankenship, D., Scambos, T. and Jacobel, R.: Surface velocity and mass balance of Ice Streams D and E, West Antarctica, Journal of Glaciology, 42(142), 461–475, https://doi.org/10.3189/S0022143000003452, 1996.

Brodie, K. H. and Rutter, E. H.: Deformation mechanisms and rheology: why marble is weaker than quartzite, Journal of the Geological Society, 157(6), 1093–1096, https://doi.org/10.1144/jgs.157.6.1093, 2000.

Budd, W. F. and Jacka, T. H.: A review of ice rheology for ice sheet modelling, Cold Regions Science and Technology, 16, 107–144, https://doi.org/10.1016/0165-232x(89)90014-1, 1989.

Cao, P., Wu, J., Zhang, Z., Fang, B., Peng, L., Li, T., Vlugt, T. J. H. and Ning, F.: Mechanical properties of bi- and poly-crystalline ice, AIP Advances, 8(12), 125108–23, https://doi.org/10.1063/1.5042725, 2018.

Carlton, C. E. and Ferreira, P. J.: What is behind the inverse Hall–Petch effect in nanocrystalline materials? Acta Materialia, 55(11), 3749–3756, https://doi.org/10.1016/j.actamat.2007.02.021, 2007.

Castelnau, O., Shoji, H., Mangeney, A., Milsch, H., Duval, P., Miyamoto, A., Kawada, K. and Watanabe, O.: Anisotropic behavior of GRIP ices and flow in Central Greenland, Earth and Planetary Science Letters, 154(1-4), 307–322, https://doi.org/10.1016/s0012-821x(97)00193-3, 1998.

Cole, D. M.: Strain-Rate and Grain–Size Effects in Ice, Journal of Glaciology, 33(115), 274–280, https://doi:10.3189/S0022143000008844, 1987.

Craw, L., Qi, C., Prior, D. J., Goldsby, D. L. and Kim, D.: Mechanics and microstructure of deformed natural anisotropic ice, Journal of Structural Geology, 115, 152–166, https://doi.org/10.1016/j.jsg.2018.07.014, 2018.

Cross, A. J., Prior, D. J., Stipp, M. and Kidder, S.: The recrystallized grain size piezometer for quartz: An EBSD-based calibration, Geophysical Research Letters, 44(13), 6667–6674, https://doi.org/10.1002/2017gl073836, 2017.

De Bresser, J., Heege, Ter, J. and Spiers, C.: Grain size reduction by dynamic recrystallization: can it result in major rheological weakening? Int J Earth Sci, 90(1), 28–45, https://doi.org/10.1007/s005310000149, 2001.

Drury, M. R., Humphreys, F. J. and White, S. H.: Large strain deformation studies using polycrystalline magnesium as a rock analogue. Part, Physics of the Earth and Planetary Interiors, 40(3), 208–222, https://doi.org/10.1016/0031-9201(85)90131-1, 1985.

Durham, W. B., Heard, H. C. and Kirby, S. H.: Experimental deformation of polycrystalline H2O ice at high pressure and low temperature: Preliminary results, Journal of Geophysical Research, 88(S01), B377–B392, https://doi.org/10.1029/JB088iS01p0B377, 1983.

Duval, P., Ashby, M. F. and Anderman, I.: Rate-controlling processes in the creep of polycrystalline ice, The Journal of Physical Chemistry, 87(21), 4066–4074, https://doi.org/10.1021/j100244a014, 1983.

Ehre, D. and Chaim, R.: Abnormal Hall–Petch behavior in nanocrystalline MgO ceramic, J Mater Sci, 43(18), 6139–6143, https://doi.org/10.1007/s10853-008-2936-z, 2008.

Eleti, R. R., Chokshi, A. H., Shibata, A. and Tsuji, N.: Unique high-temperature deformation dominated by grain boundary sliding in heterogeneous necklace structure formed by dynamic recrystallization in HfNbTaTiZr BCC refractory high entropy alloy, Acta Materialia, 183, 64–77, https://doi.org/10.1016/j.actamat.2019.11.001, 2020.

Gifkins, R. C.: Grain-boundary sliding and its accommodation during creep and superplasticity, MTA, 7(8), 1225–1232, https://doi.org/10.1007/BF02656607, 1976.

Gifkins, R. C.: The effect of grain size and stress upon grain-boundary sliding, MTA, 8(10), 1507–1516, https://doi.org/10.1007/bf02644853, 1977.

Glen, J. W.: The creep of polycrystalline ice. Proceedings of the Royal Society of London. Series A. Mathematical and Physical Sciences, 228, 519-538, https://doi.org/10.1098/rspa.1955.0066, 1955.

Golding, N., Durham, W. B., Prior, D. J. and Stern, L. A.: Plastic faulting in ice, Journal of Geophysical Research: Solid Earth, 1–45, https://doi.org/10.1029/2019JB018749, 2020.

Goldsby, D. L. and Kohlstedt, D. L.: Grain boundary sliding in fine-grained ice I, Scripta Materialia, 37(9), 1399–1406, https://doi.org/10.1016/s1359-6462(97)00246-7, 1997.

Goldsby, D. L. and Kohlstedt, D. L.: Superplastic deformation of ice: Experimental observations, Journal of Geophysical Research: Solid Earth, 106(B6), 11017–11030, https://doi.org/10.1029/2000JB900336, 2001.

Guo, D., Song, S., Luo, R., Goddard, W. A., III, Chen, M., Reddy, K. M. and An, Q.: Grain Boundary Sliding and Amorphization are Responsible for the Reverse Hall-Petch Relation in Superhard Nanocrystalline Boron Carbide, Phys. Rev. Lett., 121(14), 145504, https://doi.org/10.1103/PhysRevLett.121.145504, 2018.

Hammonds, K. and Baker, I.: The effects of Ca ++on the strength of polycrystalline ice, Journal of Glaciology, 62(235), 954–962, https://doi.org/10.1017/jog.2016.84, 2016.

Hansen, L. N., Zimmerman, M. E. and Kohlstedt, D. L.: Grain boundary sliding in San Carlos olivine: Flow law parameters and crystallographic-preferred orientation, Journal of Geophysical Research, 116(B8), 149–16, https://doi.org/10.1029/2011jb008220, 2011.

Hansen, L. N., Kumamoto, K. M., Thom, C. A., Wallis, D., Durham, W. B., Goldsby, D. L., Breithaupt, T., Meyers, C. D. and Kohlstedt, D. L.: Low‐Temperature Plasticity in Olivine: Grain Size, Strain Hardening, and the Strength of the Lithosphere, Journal of Geophysical Research: Solid Earth, 124(6), 5427–5449, https://doi.org/10.1029/2018JB016736, 2019.

Hiraga, T., Miyazaki, T., Yoshida, H. and Zimmerman, M. E.: Comparison of microstructures in superplastically deformed synthetic materials and natural mylonites: Mineral aggregation via grain boundary sliding, Geology, 41(9), 959–962, https://doi.org/10.1130/g34407.1, 2013.

Hirth, G. and Tullis, J.: Dislocation regimes in quartz aggregates, Journal of Structural Geology, 14(2), 145–159, https://doi.org/10.1016/0191-8141(92)90053-y, 1992.

Hirth, G.: Laboratory Constraints on the Rheology of the Upper Mantle, in Plastic Deformation of Minerals and Rocks, edited by S.-I. Karato and H.-R. Wenk, pp. 97–120, De Gruyter, Berlin, Boston. 2002.

Hirth, G. and Kohlstedt, D.: Rheology of the upper mantle and the mantle wedge: A view from the experimentalists, in Inside the Subduction Factory, vol. 138, pp. 83–105, American Geophysical Union, Washington, D. C., 2003.

Jacka, T. H. and Jun, L.: Flow rates and crystal orientation fabrics in compression of polycrystalline ice at low temperatures and stresses, Physics of Ice Core Records, 83–102 [online] Available from: http://hdl.handle.net/2115/32463, 2000.

Jackson, M. and Kamb, B.: The marginal shear stress of Ice Stream B, West Antarctica, Journal of Glaciology, 43(145), 415–426, https://doi.org/10.3189/S0022143000035000, 1997.

Jensen, D. D., Thorsteinsson, T., Alley, R. and Shoji, H.: Flow properties of the ice from the Greenland Ice Core Project ice core: The reason for folds? Journal of Geophysical Research: Solid Earth, 102(C12), 26831–26840, https://doi.org/10.1029/97JC01266, 1997.

Journaux, B., Chauve, T., Montagnat, M., Tommasi, A., Barou, F., Mainprice, D. and Gest, L.: Recrystallization processes, microstructure and crystallographic preferred orientation evolution in polycrystalline ice during high-temperature simple shear, The Cryosphere, 13(5), 1495–1511, https://doi.org/10.5194/tc-13-1495-2019, 2019.

Kirby, S. H., Durham, W. B., Beeman, M. L., Heard, H. C. and Daley, M. A.: Inelastic properties of ice Ih at low temperatures and high pressures, J. Phys. Colloques, 48(C1), C1–227–C1–232, https://doi.org/10.1051/jphyscol:1987131, 1987.

Koizumi, S., Hiraga, T. and Suzuki, T. S.: Vickers indentation tests on olivine: size effects, Physics and Chemistry of Minerals, 47(2), 1–14, https://doi.org/10.1007/s00269-019-01075-5, 2020.

Kuiper, E. J. N., Weikusat, I., de Bresser, J. H., Jansen, D., Pennock, G. M., and Drury, M. R.: Using a composite flow law to model deformation in the NEEM deep ice core, Greenland: Part 1 the role of grain size and grain size distribution on the deformation of Holocene and glacial ice, The Cryosphere Discuss, https://doi.org/10.5194/tc-2018-275, 2019a.

Kuiper, E. J. N., de Bresser, J. H. P., Drury, M. R., Eichler, J., Pennock, G. M. and Weikusat, I.: Using a composite flow law to model deformation in the NEEM deep ice core,

Greenland: Part 2 the role of grain size and premelting on ice deformation at high homologous temperature, The Cryosphere Discussions, https://doi:10.5194/tc-2018-275, 2019b.

Langdon, T. G.: Grain boundary sliding revisited: Developments in sliding over four decades, J Mater Sci, 41(3), 597–609, https://doi.org/10.1007/s10853-006-6476-0, 2006.

Langdon, T. G.: Seventy-five years of superplasticity: historic developments and new opportunities, J Mater Sci, 44(22), 5998–6010, https://doi.org/10.1007/s10853-009-3780-5, 2009.

Masumura, R. A., Hazzledine, P. M. and Pande, C. S.: Yield stress of fine grained materials, Acta Materialia, 46(13), 4527–4534, https://doi.org/10.1016/s1359-6454(98)00150-5, 1998.

McCarthy, C., Savage, H. and Nettles, M.: Temperature dependence of ice-on-rock friction at realistic glacier conditions, Phil. Trans. R. Soc. A, 375(2086), 20150348–20150348, https://doi.org/10.1098/rsta.2015.0348, 2017.

Mellor, M. and Cole, D. M.: Deformation and failure of ice under constant stress or constant strain-rate, Cold Regions Science and Technology, 5(3), 201–219, https://doi.org/10.1016/0165-232x(82)90015-5, 1982.

Mellor, M. and Cole, D. M.: Stress/strain/time relations for ice under uniaxial compression, Cold Regions Science and Technology, 6(3), 207–230, https://doi.org/10.1016/0165-232x(83)90043-5, 1983.

Montagnat, M., Chauve, T., Barou, F., Tommasi, A., Beausir, B. and Fressengeas, C.: Analysis of dynamic recrystallization of ice from EBSD orientation mapping, Front. Earth Sci., 3, 411–13, https://doi.org/10.3389/feart.2015.00081, 2015.

Padmanabhan, K. A., Dinda, G. P., Hahn, H. and Gleiter, H.: Inverse Hall–Petch effect and grain boundary sliding controlled flow in nanocrystalline materials, Materials Science & Engineering A, 452-453, 462–468, https://doi.org/10.1016/j.msea.2006.10.084, 2007.

Padmanabhan, K. A., Sripathi, S., Hahn, H. and Gleiter, H.: Inverse Hall–Petch effect in quasi- and nanocrystalline materials, Materials Letters, 133(C), 151–154, https://doi.org/10.1016/j.matlet.2014.06.153, 2014.

Pande, C. S. and Cooper, K. P.: Nanomechanics of Hall–Petch relationship in nanocrystalline materials, Progress in Materials Science, 54(6), 689–706, https://doi.org/10.1016/j.pmatsci.2009.03.008, 2009.

Prior, D. J.: Problems in determining the misorientation axes, for small angular misorientations, using electron backscatter diffraction in the SEM, Journal of Microscopy, 195(3), 217–225, https://doi:10.1046/j.1365-2818.1999.00572.x, 1999.

Qi, C., Goldsby, D. L. and Prior, D. J.: The down-stress transition from cluster to cone fabrics in experimentally deformed ice, Earth and Planetary Science Letters, 471, 136–147, https://doi.org/10.1016/j.epsl.2017.05.008, 2017.

Rowe, K. J. and Rutter, E. H.: Palaeostress estimation using calcite twinning: experimental calibration and application to nature, Journal of Structural Geology, 12(1), 1–17, https://doi.org/10.1016/0191-8141(90)90044-y, 1990.

Ryou, H., Drazin, J. W., Wahl, K. J., Qadri, S. B., Gorzkowski, E. P., Feigelson, B. N. and Wollmershauser, J. A.: Below the Hall–Petch Limit in Nanocrystalline Ceramics, ACS Nano, 12(4), 3083–3094, https://doi.org/10.1021/acsnano.7b07380, 2018.

Schmid, S. M., Boland, J. N., and Paterson, M. S.: Superplastic flow in finegrained limestone. Tectonophysics, 43(3-4), 257–291. https://doi.org/10.1016/0040-1951(77)90120-2, 1977

Seidemann, M., 2017, Microstructural evolution of polycrystalline ice during non-steady state creep [PhD thesis: University of Otago, p110].

Sheinerman, A. G., Castro, R. H. R. and Gutkin, M. Y.: A model for direct and inverse Hall-Petch relation for nanocrystalline ceramics, Materials Letters, 260, 126886, https://doi.org/10.1016/j.matlet.2019.126886, 2020.

Somekawa, H. and Mukai, T.: Hall–Petch Breakdown in Fine-Grained Pure Magnesium at Low Strain Rates, Metall and Mat Trans A, 46(2), 894–902, https://doi.org/10.1007/s11661-014-2641-2, 2015.

Spiers, C. J.: Fabric development in calcite polycrystals deformed at 400° C, bulmi, 102(2), 282–289, https://doi.org/10.3406/bulmi.1979.7289, 1979.

Ter Heege, J. H., De Bresser, J. H. P. and Spiers, C. J.: Rheological behaviour of synthetic rocksalt: the interplay between water, dynamic recrystallization and deformation mechanisms, Journal of Structural Geology, 27(6), 948–963, https://doi.org/10.1016/j.jsg.2005.04.008, 2005.

Treverrow, A., Budd, W. F., Jacka, T. H. and Warner, R. C.: The tertiary creep of polycrystalline ice: experimental evidence for stress-dependent levels of strain-rate enhancement, Journal of Glaciology, 58(208), 301–314, https://doi.org/10.3189/2012jog11j149, 2012.

Vaughan, M. J., Prior, D. J., Jefferd, M., Brantut, N., Mitchell, T. M. and Seidemann, M.: Insights into anisotropy development and weakening of ice from in situ P wave velocity monitoring during laboratory creep, Journal of Geophysical Research: Solid Earth, 122(9), 7076–7089, https://doi.org/10.1002/2017JB013964, 2017.

Walker, A. N., Rutter, E. H. and Brodie, K. H.: Experimental study of grain-size sensitive flow of synthetic, hot-pressed calcite rocks, Geological Society, London, Special Publications, 54(1), 259–284, https://doi.org/10.1144/gsl.sp.1990.054.01.24, 1990.

Warren, J. M. and Hirth, G.: Grain size sensitive deformation mechanisms in naturally deformed peridotites, Earth and Planetary Science Letters, 248(1-2), 438–450, https://doi.org/10.1016/j.epsl.2006.06.006, 2006.

Weertman, J. R.: Hall-Petch strengtheningin nanocrystalline metals, Materials Science & Engineering A, 16(1-2), 161–167, https://doi.org/10.1016/0921-5093(93)90319-a, 1993.

Zener, C. and Hollomon, J. H.: Effect of Strain Rate Upon Plastic Flow of Steel, Journal of Applied Physics, 15(1), 22–32, https://doi.org/10.1063/1.1707363, 1944.

---

## Author Response (AR1)

**This document is comprised by:**

**1. Response to reviewer 1 (P2-P23)**

**2. Response to reviewer 2 (P24-P38)**

**3. A list of all relevant changes made in the manuscript (P39-P50)**

**4. Revised manuscript with changes marked-up**

**Response to Reviewer 1**

We thank Reviewer 1 for her thoughtful and helpful review of our paper. The comments have helped us improve the manuscript significantly. Our reply to reviewer comprises two parts: (1) some short general statements and (2) point-by-point reply to comments from reviewer. Reviewers comments are in blue type. *Extracts from our revised manuscript are in italics.*

**Section one: general statements**

*1. This work contains data which are completely new.*

We would like to emphasize that the sequence of microstructures and CPOs developed with increasing strain has not been documented before for ice deformed at cold temperatures (-20, -30 °C). This is highlighted by one of the comments by the reviewer: "The hypothesis that GBM being less active at low temperature, the impact of grain rotation driven by intracrystalline slip prevails is much clearer, especially since it is very coherent with the observations that the cone angle is reduced, and more orientations are found close to the vertical. This assertion is, indeed, justified by the experimental observations. This is, in fact, the main "novelty" of the presented work and should be emphasised more."

Earlier work showing the up-strain microstructures and CPOs (e.g. Jacka & Macagnan, 1984; Montagnat 2015) are at warm conditions where the CPO evolves towards an open cone (small circle). Experiments at colder temperatures (-30 °C and colder) to strains of ~ 20% (Craw et al., 2018; Prior et al 2015) show CPOs have maxima of c-axes parallel to compression. No published work shows the up-strain evolution of microstructures or CPOs at -20 or -30 °C (or colder temperatures). Jacka & Li (2000) show CPOs at ~10% strain at ~ -15 and -20 °C and ~ 3% strain at -45 °C but include no microstructural data and do not explore the up-strain evolution. Recent work by Wilson et al (2019) shows CPOs at -15 and -20 °C at 20% strain, but show no up-strain evolution. In this paper the up-strain sequence at -30°C documents the evolution towards a cluster CPO, the sequence at -10 °C the evolution towards an open cone CPO and the sequence at -20 °C something between these two. Understanding how and why different CPOs develop as a function of temperature should give a better insight into the mechanisms that control CPO development and mechanical behaviour.

*2. Different interpretations can be made from the same observation.*

One of the reviewer's objections relates to our interpretation of the microstructural development as involving grain boundary sliding (GBS). We accept that the factual observations could be interpreted in different ways and in the revision, we include some alternative interpretations (including "spontaneous" nucleation) of the data, with some discussion of the merits and drawbacks of each of these interpretations. We hope that we have kept the observations and interpretations clearly separated and we have reduced the emphasis on our preferred interpretation of GBS. We have also identified some of the tests that may facilitate distinguishing these different interpretations in the future. Some more details are included in answers to specific points.

The reviewer's comments highlight that our original manuscript did not really make clear that we do interpret intracrystalline dislocation glide that causes lattice rotation as one of the key processes controlling CPO development. We hope that we have made this much clearer in the revised manuscript. The operation of a GBS process, if this is correct, would be additional to the role of intracrystalline dislocation glide and associated recovery and recrystallisation processes.

**Section two: point-by-point reply to comments**

**1.** P3 l. 28: why should the sample be cooled in liquid nitrogen? Couldn't that induce some thermal stress due to the fast and strong temperature gradient? Changes in local microstructure, dislocation arrangements, etc. are expected to occur in the first minutes after the test... so this quenching should not avoid it.

The purpose of cooling deformed ice samples in liquid nitrogen is to preserve the ice microstructure during a long-term storage and also for intercontinental transfer (using a nitrogen dry shipper). The time interval between ice deformation work and cryo-EBSD analyses is normally longer than one month. Very cold storage has several advantages:

1. The colder temperatures minimise the chance of long-term microstructure change.
2. The storage is much more reliable than storing in an electric freezer. The dewar usually lasts one to two months between liquid nitrogen top ups and can easily stay cold for > 6 months if fully filled (as has just been done for COVID 19 lock down).
3. The storage solution is much cheaper than a very cold (-80C) freezer.

Thermal shock is a risk with liquid nitrogen storage. Plunging a -20°C sample directly into liquid nitrogen will cause the sample to shatter. More careful handling prevents any fracturing. We have some samples (e.g. MIT666 (Prior et al., 2012)) that have been cycled between liquid nitrogen and much warmer temperatures (e.g. for grain growth experiments (Becroft, 2015)) with no discernible changes to structure or microstructure. We applied a staged cool-down method to progressively cool deformed samples to a liquid nitrogen temperature:

1. Firstly, samples are cooled down to -30 °C in a chest freezer for ~5 minutes while the indium jacket is being peeled off.

2. Secondly, samples are transferred into liquid nitrogen mist at ~ -100 °C for ~10 minutes.

3. Finally, samples are transferred to a liquid nitrogen dewar for a long-term storage.

We use the staged cool-down process to prevent a drastic temperature change, which might lead to thermal stress in sample. The staged cool-down method was not clarified in the manuscript and it has been clarified in the modified version.

We always worry that there could be some post-deformational changes that change the sample microstructure after load has been removed but before the sample is "quenched". Our procedures try to minimise this and as a minimum, ensure all samples are treated in a similar way. After each deformation run ended, we drove back the driving piston, depressurized the pressure vessel, and extracted the sample from deformation rig in 10 to 30 minutes. Each sample was exposed at room temperature for less than 30 seconds for taking photos. Soon after, the samples were cooled down to a liquid nitrogen temperature using the staged cool-down method.

Static annealing of the ice microstructure during the sample extraction is a potential issue in all ice deformation experiments. The experiments shown by Hidas and others (2017) quantified the ice microstructural changes during thermal annealing at -5 to -2 °C. They show no significant ice microstructural change in pre-deformed samples over the time scales of our sample extraction process. More specifically, Hidas and others (2017) shows deformation-induced tilt boundaries and kink bands remain stable during early stages of annealing. It takes >24 h of annealing to start to erase these microstructures.

We have added a statement into section 2.2: "*Minor static recovery of the ice microstructures may happen on this timescale (Hidas et al., 2017), but significant change in CPO or grain size is unlikely.*"

**2.** P7 l. 8-10: in the paragraph just before it is mentioned that the grain size distributions are mostly bimodals, and therefore not gaussians... The mean and STD parameters are therefore not suited to described them, since they do not represent well the given statistics. I therefore suggest the authors to provide medians and quartile data instead to better fit the type of distributions observed.

The reviewer has a good point and we have addressed this by including a wider range of grain size measurements, that may reflect better a scalar representation of a skewed distribution; for example, we have added median and quartile grain size values to Table 3. We have also kept the values of mean grain size in the paper, as this is a common measure used in microstructural studies and provides some comparability to those studies.

**3.** P7 l. 30: couldn't it exist a bias link to the lack of resolution in step size and misorientation when getting toward smaller subgrains?

We'll discuss the two issues separately:

The data filtering process removes grains with area equivalent diameters lower than 20 µm. Thus, there is an artificial lower cut off to the grain size and sub grain distributions (as there always is for any microscopic method). Our grain size distributions show peaks above the 20 µm cut off, these peaks are unlikely to change even though we may miss some smaller grains. The subgrain peak in all cases is ~ 20 µm (i.e. at th resolution cut-off), particularly at lower temperatures. The true peak subgrain size is therefore likely to be < 20 µm and it is probable that we are missing a substantial population of smaller subgrains in our analyses. We acknowledge this limitation and have added a statement into section 3.3.3: "*In many cases, particularly at lower temperatures, the peak corresponds to the lower grain size resolution (cut off) indicating that we could be missing smaller subgrains. For this reason, the peak subgrain sizes are not useful and the median and mean subgrain sizes probably represent overestimates.*"

The orientation resolution in these EBSD maps is ~0.5°, so that we cannot reliably identify misorientations of ~1°. The misorientation threshold chosen for identifying subgrain boundaries is ≥ 2°, the lowest angle that returns reliable results for our data. Lower angle subgrain boundaries could exist and at least one comparison of TEM and EBSD has shown that this is the case (in quartz :(Shigematsu et al., 2006)). We have added a statement into section 2.4: "*The angular resolution (error of crystallographic orientation measurement of each pixel) of the EBSD data is ~0.5°.*"

**4-1.** Part 3.2.3: this is not clear to me how is the subgrain size defined and calculated.

Thanks for pointing our lack of clarity in defining how subgrain size was defined and calculated.

We have added a statement in section 2.5.1: "*Deformed ice is often characterised by a development of subgrain boundaries where the misorientations between neighbouring pixels are lower than the misorientation angle threshold of grain boundaries (e.g. Montagnat et al., 2015; Weikusat et al., 2017). An ice grain can be separated into several subgrains by one or more subgrain boundaries. We calculated subgrain size using boundary misorientation thresholds of ≥ 2°. Grain size and subgrain size were calculated as the diameter of a circle with the area equal to the measured area of each grain or subgrain.*"

**4-2.** Although we observe a clear grain boundary structure in the figures, there appears no clear subgrain structure (as one could observe in some minerals or metals for instance). On the contrary, subgrains appear more like straight tilt bands or kink bands, with, in some places, some variations around the straight shape.

We agree with the observation from the reviewer that suggests many of the subgrain boundaries are straight tilt bands or kink bands. The subgrain structure in the submitted manuscript version was revealed by the Weighted Burgers vector (WBV) method. We didn't make it clear that the measurement of subgrain sizes were not based on the WBV data, but on the misorientations between adjacent pixels. The WBV was a legacy of a much earlier manuscript and the reviewer's comments have highlighted that it is better removed. The new maps (Fig. 4(c), 5(c), 6(c)) show subgrain boundaries that correspond to the much simpler misorientation threshold. We modified statements on section 3.3.1 to: "*Distinct sub-grain boundaries can be observed in all the samples (Fig. 4 (c), 5 (c) and 6 (c)). Many of the subgrain boundaries appear to be straight, some with slight curvature. A small number have strong curvature. Interconnected subgrain boundaries can be observed in some of the grains. Subgrain boundaries subdivide grains into subgrains.*"

Figure R1.1 shows the structures of subgrain boundary in deformed ice samples as well as other experimentally or naturally deformed minerals and metals, e.g. quartz (Cross et al., 2017; Killian and Heilbronner, 2017), Olivine (Hansen et al., 2012), Magnox alloy (Wheeler, 2009) and Zircon (MacDonald et al., 2013). We added a new statement section 4.1.2: "*The subgrain boundary geometry is comparable with other experimentally or naturally deformed rock and metal samples, e.g. quartz (Cross et al., 2017a; Killian and Heilbronner, 2017), Olivine (Hansen et al., 2012), Magnox alloy (Wheeler, 2009) and Zircon (MacDonald et al., 2013).*"

[Figure]

**Figure R1.1.** Illustrations of subgrain boundaries developed in experimentally deformed materials. **(a)** Experimentally deformed ice samples to ~12% true axial strain at -10, -20 and -30 °C from this study. Sub-grain boundaries where misorientation between neighbouring pixels between 2° and 10° are coloured red. Grain boundaries where misorientation between neighbouring pixels higher than 10° are coloured black **(b)** Experimentally deformed quartz sample W1051 (189±30 MPa, 1000 °C, 41% axial strain, $1.9\text{-}2.9 \times 10^{-5}$ s$^{-1}$) from Cross et al., 2017. Each pixel is coloured by the value of mis2mean (misorientation between each pixel and the mean orientation of their parent grain). **(c)** Experimentally deformed quartzite sample W946 (1.5 GPa, 875 °C, 3.3 shear strain, $3.1 \times 10^{-5}$ s$^{-1}$) from Killian and Heilbronner, 2017. Each pixel is coloured by Kernel average misorientation (KAM) of a 24-pixel neighbourhood. **(d)** Experimentally deformed olivine sample PT0552 (136 MPa, 8.8 shear strain, $0.551 \times 10^{-3}$ s$^{-1}$) from Hansen et al., 2012. Each pixel is coloured by local misorientation calculated with 5 by 5 pixel averaging filter. **(e)** Experimentally deformed Magnox alloy containing 0.9% Al and 0.005% Be (30% strain, 200 °C, $1.9 \times 10^{-4}$ s$^{-1}$) from Wheeler et al., 2009. Each pixel is coloured by the weighted Burgers vector (WBV) magnitude.

The 3D WBV is projected onto the map plane and marked as arrow. **(f)** Naturally deformed Zircon grain BP06/3 from MacDonald et al., 2013. Each pixel is coloured by a misorientation angle calculated from its orientation relative to a given point.

**4-3** I would be curious to see, for instance, how was measured this subgrain size in the sample deformed at 3% at -10°C, or at 12% at -20°C. I think that the author should clarify this technical aspect as they make a lot of explanation rely on such "average" parameters. Furthermore, provided it is calculated properly, I doubt the distribution is normal, and I think that a metric other than the average would fit best.

The reviewer is correct that the subgrain size distributions are not normal they are skewed in a manner similar to the grain size distributions. In our modification we have tried to focus on elements of the data that are robust and helpful in interpretation. These are basically that subgrains exist and they are smaller than grains. We have removed the data and discussions related to subgrain sizes calculated using boundary misorientation angles of 4°, 6° and 8° (boundary hierarchies) and we have both mean and median subgrain sizes for comparison with grain sizes. The new presentation of data leads to a statement in section 3.3.3: "*Subgrain size distributions (Fig. 4(e), 5(e) and 6(e)) are similar to the grain size distributions (Fig. 4(d), 5(d) and 6(d)), but the median and mean subgrain sizes are smaller than median and mean grain sizes (Table 3).*"

**4-4** In particular, the following assertion is questionnable: "because subgrain rotation recrystallization should produce grains that have similar sizes with subgrains, while bulging nucleation should produce grains that have smaller sizes than subgrains" that rely on a parameter (subgrain size) that is ill measured here, since subgrain structure does not resemble at all the one observe in quartz (Halfpenny et al 2012).

We agree with the reviewer that in this case this approach lacks robustness. We removed the extended discussion that including hypotheses of bulging nucleation in the revised manuscript.

**4-5** On top of that, this expected hierarchy of grain size depending from the nucleation mechanism comes from one study on quartz and should not be taken as granted, see for instance Humphreys 2004 (Materials Science Forum) that shows clearly a bulged grain much larger that the surrounding subgrains. It will therefore depend on the material and its anisotropy, and on the resolution of the observations (ability to distinguish between a grain resulting from a bulge and one resulting from subgrain rotation...)

We agree that in this case the boundary hierarchy analyses do not give useful insight and we have removed the hierarchy data and related description and discussion.

**5-1** p8 l. 8-9: To consider <D_small> as a good representative of the mean recrystallized grain size is also a strong hypothesis that should be justified (either by some specific observations or by references from previous work). It will, in particular, depend on the relative effect of grain boundary migration compare to nucleation during recrystallization (and therefore on the temperature of the test) since an apparent small grain size at high GBM rate could well be a 2D cut of a strongly lobated grain, while, at lower temperature (lower GBM rate), small grains indeed are newly recrystallized grains (see for instance the sample deformed at 8% at -10°C, could one certify that small grains observed on the 2D sections are indeed small grains?). Here again appears the necessity of statistic metrics adapted to the observed distributions.

Small grains observed from the EBSD data can be a 2D cross section of a larger 3D grain. We thank the reviewer for these comments as they have pushed us to complete new analyses aiming at quantifying the effects of two distinct (albeit related) stereological issues that add value to

this paper and maybe will be useful to others as analytical approaches. The first issue relates to the misidentification of "small" grains, as these could appear from slices cut close to the perimeter of a large grain in 3-D (Underwood, 1973). The second issue relates to the oversampling of grains that have highly irregular, branching shapes in 3-D and appear more than once on a 2-D surface (Hooke and Hudleston, 1980; Monz et al, 2020). The new analyses are presented in section 3.3.2 (for observation) and 4.1.2 (for discussion). Details of the stereological analyses are presented in section S3 and S4 of the supplementary material. Here we present key findings:

To assess the first issue (misidentification of "small" grains) we extracted one-dimensional grain size measurements (by linear intercept) from two-dimensional maps. From this analysis, we can state whether a "small" 1-D grain is indeed a "small" grain in 2-D. At ~20% strain the percentage of "small" grains on a 1-D line that correspond to "small" grains in the 2-D EBSD map is 64%, 76% and 43% at -30, -20 and -10 °C, respectively. These data suggest that at 20% strain the presence of "small" grains in 3D is likely, with the confidence in this statement increasing at reduced temperatures. Another observation supports this. Figure R1.2 shows examples of "big" and "small" grains in deformed to ~20% strain at -10 °C and -30 °C. Many "small" grains have "small" grain neighbours. At -30 °C some "small" grains are entirely surrounded by other "small" grains. At -10 °C there are lines of "small" grains in contact along the boundary between "large" grains.

[Figure]

**Figure R1.2.** Examples of "big grains" and "small grains" in PIL007 deformed at -10 °C to ~20% strain, and PIL268 deformed at -30 °C to ~20%. The "small grains" have black grain boundaries, the "big grains" have white grain boundaries. Each grain is coloured by mean orientation with IPF-Y colour code.

It is very difficult (and at -30 C impossible) to have all of these "small" grains linked to large grains in the third dimension whilst maintaining a microstructure that looks like the microstructures in these maps. This is the case at 20% strain. At lower strains the percentage of "small" 1-D segments that correspond to "small" 2-D grains is lower so the confidence with which we can define "small" grains is reduced.

The 1-dimensional data also provide some insight into the second issue, oversampling of grains . In all samples >90% of 2-D grains along an arbitrary line are unique (that is, they are cut only once). Of course, with lines in multiple directions the total percentage of unique grains will decrease. As EBSD provides full crystal orientation data we can extend this analysis to entire maps. We can assess the likelihood of each 2-D mapped grain being connected in the third dimension to another 2-D mapped grain by comparing each grain's orientation (mean

orientation) to all other grains within a certain distance (1mm is used here). If the misorientation between a grain and a nearby grain is below a certain threshold (10°) then we define them as being connected parts of the same grain in 3D. These thresholds probably overestimate the number of grains connected in 3D. 1mm is more than double the size of the largest grain and 10° is more than twice the median and significantly lower than the upper quartile in mis2mean data (the misorientation angle between all pixels in a grain and the mean orientation of that grain) for all samples. The percentage of "unique" grains (that only appear at the surface once) relative to all grains in a 2D map are higher than 70% at all temperatures and strains (Table S2-S4, Table 3).

The procedure outlined in the last paragraph allows us to estimate the number of "distinct" grains (where all 2-D grains attributed to the same 3-D grain are counted as one grain) in each map and from this, the number density (grain number per unit area) of "distinct" grains. The number density of "distinct" grains increases by more than a factor of 3 relative to the starting material in all samples at all temperatures: reaching values > 6 times initial at -10 °C and >12 times initial at the lower temperatures. The number density of "distinct" grains is generally higher at strains of $\varepsilon \geq \sim 12\%$ than at strains of $\varepsilon \leq \sim 8\%$ at all temperatures, and it is generally higher in samples deformed at -20 and -30 °C than samples deformed at -10 °C.

[Figure]

**Figure R1.3.** Number density of "distinct" grains as ratio to starting material relative to true axial strain.

The analyses above provide some confidence that in all the experiments the number density of grains has increased relative to the starting material and increases with strain. That requires new grains to be generated and any measure of average grain size to reduce. If we couple this to the grain size statistics presented and the analysis of whether we are misidentifying small grains, the weight of evidence suggests that we have a real population of smaller grains. Our confidence in this statement increases with reducing temperature and increasing strain.

Having outlined new analyses that add robustness to our statements about reducing grain size with strain and the existence of a population of "small" grains we come back to the issue of how we distinguish "big" and "small" grains. There will always be a degree of arbitrariness in this and to reflect this we added a statement in section 3.3.2: "*Our scheme for separating "big" and "small" grains is not perfect, but it does provide a fast and repeatable way of looking at the possible differences in microstructures and CPO of smaller and larger grains.*"

**5-2** For the sake of clarity, I would suggest the authors not to mix result presentations and interpretations and keep interpretations for the discussion part. In particular when interpretation requires additional hypotheses on top of direct observations and results.

We have been through the manuscript and ensured the observations and interpretations are not mixed up in section 3 (results). We have kept necessary brief descriptions of process in section 3 only for the clarity of concepts introduced from other published works and not for interpretation of our own data.

**6.** P9 l. 13 "At -10 °C, the CPO intensity of "small grains" is lower than "big grains", and this contrast becomes strengthened as the temperature decreases." This could also be related with the fact that it is less straightforward to distinguish small grains from big grains for these tests, this should be mentioned here.

Yes, this is a very good point. The small grain population is easier to define and is better defined at low temperatures than at high temperature. This clearly relates to the differences in the balance of key processes at different temperatures. We hope that we have emphasised this the revised manuscript.

**7.** P11 l. 26 The authors mention "much of the stress increase prior to peak stress relates to elastic strain", and, as they notice just after, this is not coherent with the known Young modulus of ice of 9 Gpa... There is a broad literature, dating back to the 70's and 80's (Duval et al. 1983, Jacka 1984 for instance, and review by Schulson and Duval 2009) explaining that the transient behavior of ice is not elastic, but anelastic, and is related to the built of an internal stress field related to strain incompatibilities between grains. **I am therefore very astonished to read this sentence here, and I think that this should be corrected before publication**.

The "dissipative deformation" mentioned here is indeed plastic deformation related to intracrystalline dislocation slip, the porosity loss being very likely negligible.

Published literature labelled the stress increase prior to peak stress in constant displacement rate experiments as: "normally elastic" (Cole, 1987) and "quasi-elastic" (Kirby, 1987). The deceleration during primary creep in constant stress experiments was interpreted as effected by a "delayed elasticity", with a recoverable component of time-dependent elastic strain and an irrecoverable viscous strain (Mellor and Cole, 1982), and "anelasticity" (Duval et al., 1983). The reason we chose to describe the behaviour as substantially elastic is that we have other experiments where we can show that this part of the deformation is recoverable. However, these other experiments are higher rate experiments with slopes on the stress strain curve approaching the 9GPa modulus. The reviewers are correct in pointing out that, in the experiments presented in this paper, the slope is substantially below modulus and the behaviour is not substantially elastic. We have modified the statement in section 4.1.1: "*This likely includes anelastic deformation related to intergranular stress redistribution used to explain primary creep in constant load experiments (Duval et al, 1983). The curvature of the stress strain line at the start of each experiment may relate to initial porosity loss as suggested by rapid increases in ultrasonic p-wave velocity in comparable experiments by Vaughan et al., (2017).*"

**8-1.** Part 4.1.1: Discussion about GBS. The experimental results shown here present no evidence of a grain size sensitive mechanisms, since there is no initial grain size variation, no study of the influence of grain size on the stress – strain-rate relation. I therefore don't understand why GBS is mentioned here, since it is not necessary at all to explain the observations performed.

Indeed, all results presented here can be explained by intracrystalline dislocation slip accommodated by dynamic recrystallization mechanisms, as very well illustrated in the high quality EBSD observations performed. Furthermore, there exist a large number of studies showing that GBS occurs significantly only in fine-grained materials (see Boullier and Gueguen 1975, Goldsby and Kohlstedt 1997) where grain boundary diffusion can play a role (Ashby 1973). Diffusion in ice is known to be very slow, that renders the hypothesis of a diffusion-controlled mechanism quite unlikely, especially for large grains, and high strain-rate conditions as encountered here.

The particular set of experiments used in our paper does not include variable initial grain size. However, comparable experiments do show grain size sensitivity. The set of -10 °C experiments published by Qi et al 2017 have two different initial grain sizes. A plot of strain rate against the peak stresses (Fig. 3, copied below as Fig. R1.4) shows two different best fit lines for the two initial grain sizes. At peak stress (~ equivalent to min strain rate) grain size is unlikely to have changed substantially from the starting material (we have some new experiments to peak stress only that show this to be correct). The easiest interpretation of these data is that there is grain size sensitivity. In this case the sensitivity is manifest between grain sizes of ~0.25 (standard ice on Fig. 3 of Qi et al) and ~0.6mm (course grained ice). There is no clear distinction in the mechanical data for different initial grain sizes at flow stress (~tertiary creep). At flow stress, after strains of ~0.2, grain sizes have evolved substantially and mean grain sizes correlate with the stress magnitude following a piezometer type relationship, as reported for ice by Jacka and Li (1994).

[Figure]

**Figure R1.4.** Plot of strain rate versus stress on logarithmic scales using data from Qi and others (2017).

GBS has always been a problem area since there are few clear microstructural indicators to show that it has occurred: in stark contrast to intracrystalline dislocation slip and accompanying recovery and recrystallisation. Older papers that identify GBS tend to be restricted to studies of very fine materials as it is in these materials that grain size sensitive mechanisms can dominate. An important concept in material science is that mechanisms can co-exist: the whole premise of deformation mechanism maps (https://engineering.dartmouth.edu/defmech/) is based on the idea that the total strain rate is the sum of the strain rates related to each

contributing deformation mechanism. Recently, Kuiper and others (2019a, 2019b) applied the Goldsby-Kohlstedt composite flow law (which considers bulk strain rate as an additive contribution of dislocation creep and GBS) to model the deformation in NEEM ice core. The extrapolation of the experimental data to natural conditions suggests that "GBS-limited creep produces almost all deformation in the upper 2207 m of depth in the NEEM ice core (grain size between ~0.3mm and ~9mm)." GBS will contribute a larger proportion of total strain rate at fine grain sizes (e.g. experiments by Goldsby and Kohlstedt (1997)), but can still be significant at coarser sizes.

The advent of EBSD methods has allowed us to analyse microstructures in new ways and to tease out the potential for GBS in a wide range of materials. The change in misorientation axes from rational (along specific crystal directions) to random (w.r.t. crystal directions) with increasing misorientation and the weakening of CPO in recrystallised grains relative to porphyroclasts are two lines of evidence that are commonly used in the rock deformation community (starting with (Bestmann and Prior, 2003; Fliervoet et al., 1999; Jiang et al., 2000) to identify GBS as an operative mechanisms from the analysis of a final microstructure. We realised that we have not presented the basic misorientation analysis (Bestmann and Prior, 2003) and we have now included data on misorientation axes for low angle and high angle boundaries. These data, and the segmentation of CPOs for coarser and finer grains, show the same patterns that are commonly used to infer GBS in deformed rocks. This does not of course prove that GBS has occurred, it merely says that these ice experiments have microstructural characteristics that match other samples where those characteristics have been used to infer GBS.

If GBS does occur, our interpretation is that this is in addition to dislocation glide (Drury et al., 1985; Gifkins, 1976, 1977; Goldsby and Kohlstedt, 1997, 2001; Hirth, 2002; Hirth and Kohlstedt, 2003; Kuiper et al., 2019a; Kuiper et al., 2019b; Langdon, 2006, 2009; Warren and Hirth, 2006). Some authors term this dislocation accommodated GBS or "disGBS". In this mechanism, the total strain rate is the addition of a dislocation process (that changes crystal shapes and causes lattice rotation and internal distortion) and a GBS process that is probably controlled by a "viscous" mechanism within grain boundaries (small path length diffusion and/or asperity plasticity: idea originally from (Gifkins, 1976)). This is not the same as diffusion creep, irrespective of whether that is controlled by lattice diffusion (Nabarro-Herring creep) or grain boundary diffusion (Coble creep). GBS is required as an accompanying mechanism to polycrystalline diffusion creep, but in that case grain shape change is facilitated by the diffusive mass transfer processes. In diffusion creep, grain size sensitivity comes primarily from the increased path length for diffusion meaning that the change of shape of bigger grains takes longer. In "disGBS" the GBS itself is the prime source of grain size sensitivity. If there is a "viscous" grain boundary volume then the rheology will depend on the volume proportion of the sample that comprises grain boundaries: this proportion will increase with decreasing grain size.

**8-2** The authors could try to calculate the strain-rate expected based on a GBS diffusion flow law (Nabarro-Coble for instance) for similar level of stress as the one of their experiences. They would likely see that the stress – strain-rate curves they obtained are not compatible with a GBS influencing mechanism.

Please see our comments related to diffusion creep in the last paragraph of the response to 8-1.

Also see the response to 15. This outlines how we have modified the discussion of weakening.

Here we expand a little to answer this question: this is from a paper we have in progress to model the effect of grain size on the mechanical evolution of deformed ice and is beyond the scope of inclusion in this paper. The ratio of stress drop after peak is ~35% for samples deformed at warm or cold temperatures (as pointed out elsewhere by the reviewer). A simple model uses just the GBS component of the Goldsby-Kohlstedt composite flow law. The strain rate of GBS can be expressed as:

$$\dot{\varepsilon} = A\sigma^n d^{-p} \exp\left(-\frac{Q}{RT}\right), \qquad (R1.1)$$

where $A$ is a material-dependent parameter $(MPa^{-n}m^p s^{-1})$, $\sigma$ is the stress $(MPa)$, $n$ is the stress exponent, $d$ is the grain size $(m)$, $p$ is the grain-size exponent, $Q$ is the activation energy $(kJmol^{-1})$, $R$ is the gas constant $(= 8.314 \times 10^{-3} \, kJmol^{-1}K^{-1})$ and $T$ is the absolute temperature $(K)$. The flow law parameters of $Q$ and $A$ for GBS $(n = 1.8, q = 1.4)$ were taken from Kuiper and others (2019a, 2019b). For each sample, we calculated the stress, $\sigma$, by substituting mean or median grain size and temperature (Table 3) into Eq. (R1.1). $\sigma$ were normalised with respect to the peak stress. Figure R1.5 shows, the normalised stress estimated using the GBS flow law and measured strain rates and grain sizes as a function of strain at both -10 and -30 °C. There is much more work for us to do, but the models give stress strain patterns that have the same general form as the mechanical data, with an underestimate of weakening at -10C and an overestimate at -30 °C.

[Figure]

**Figure R1.5**. Normalised stress vs stress for samples deformed at **(a)** -10 °C and **(b)** -30 °C. For each temperature series, the stresses estimated from GBS mechanism are normalised by the estimated stress at ~3% strain.

**9** Part 4.1.2: In this part, the authors use the subgrain size measurements to estimate the role of subgrain rotation in the recrystallization mechanisms.

Once again, the subgrain structure observed here is very far from the ones in quartz, to enable using the paper mentioned here as a reference (Trimby et al. 1998), and I think the authors should be much clearer about the way they evaluate the subgrain size before getting to strong an interpretation from this parameter.

Ice behavior, and in particular in the experiments presented here, is very different from the one of more isotropic materials in the sense that the dislocation substructures are not characterized by subgrain cells as observed in Al or Quartz for instance. This is due to the fact that subgrain

substructures as observed in Quartz results from equivalent activity of several slip systems. Although one observe some c-dislocations in the microstructure, slip system activity in ice remains dominated by basal slip, and resulting subgrains have mostly the shape of large tilt and kink bands.

Only close to GB and triple junctions will we find more complex substructures. Is it enough to evaluate an "average" subgrain size? Care must therefore be taken before using interpretations coming from these more isotropic materials. And explanation should be given about how is this subgrain size measured here.

We agree with the reviewer about the lack of clarity of the measurement of subgrains. We have reduced significantly the discussions related to subgrain size. We have plotted subgrain boundary map based on misorientation angle of neighbouring pixels, added misorientation angle analyses and calculated median subgrain size to better support subgrain analyses. Please refer to responses to comments 4-1 to 4-5 for more details.

**10** P13 l. 1-2: Indeed, Jacka and Li Jun 1994 evidenced a linear relationship between grain size and stress during dynamic recrystallization of polycrystalline ice (creep experiments, tertiary creep). I think that this should be mentioned here.

We agree with the reviewer. We added a statement in section 4.1.2: "Jacka and Li (1994) show a linear relationship between ice grain size and stress from deformed ice samples that reach tertiary creep."

**11**. P13. l. 5: here again, caution must be taken with making use of the subgrain size as it is still ill defined... and the ice case can not be compared straightforward to Halpenny et al. studies!

Indeed, observation given l. 16-17 goes in the direction of my remark... Subgrain size, if measurable here, can not be used similarly as in the other studies mentioned since there is no clear subgrain substructure. But, still, subgrain rotation could explain part of the recrystallization by, for instance, closing the bulges (see Chauve et al. 2017, Phil Trans), or by separating grains via highly misoriented tilt or kink bands. But, indeed, one can not talk about "continous" recrystallization as observed in Al for instance (see Sakai et al. Progress in Materials Science, 60(0):130–207, 3 2014 for a review).

We have modified the manuscript by presenting subgrain structures in a clearer way. Please refer to responses to comments 4-1 to 4-5 for details.

**12.** Part 4.1.3 p 14 l. 11: "Because grains with hard slip orientations should have greater internal distortions", there is absolutely no proof of that in ice, and some recent work tend to show that there is no systematic relation between orientation and strain localisation (see Grennerat et a. 2012 for instance) or between orientation and subgrains density (see Journaux et al. 2019 for instance). I think it should not be considered as granted, in particular when not shown directly in your experiments. Have you tried, for instance, to measure the density of GNDs as a function of grain orientation?

We modified our statement in 4.1.3 to: "*Cone-shaped c-axes CPOs have been related to strain-induced GBM favouring the growth of grains with easy slip orientations (high Schmid Factors) (Duval and Castelnau., 1995; Little et al., 2015; Vaughan et al., 2017; Qi et al., 2017). Linked to this is the idea that grains with hard slip orientations should have greater internal distortions (Duval and Castelnau., 1995; Bestmann and Prior 2003), and therefore store higher internal strain energy. If this is correct then hard slip grains are likely to be consumed by grains with easy slip orientations through GBM (Duval and Castelnau., 1995; Piazolo et al., 2006; Killian et al., 2011; Qi et al., 2017; Xia et al., 2018). However, we have to re-*

*evaluate the detail of this idea, as recent studies on deformed ice samples show there is no systematic relation between orientation and strain localisation at low strain (Grennerat et al. 2012). Furthermore, studies of high-strain shear samples find no clear difference in the geometrically necessary dislocation density within the two maxima that develop in simple shear (Journaux et al. 2019). An alternative, and as yet incomplete, explanation from Kamb (1959) relates recrystallisation directly to the elastic anisotropy of crystals and through this to the orientation of the stress field. At this stage the observation that ice CPOs developed at relatively high temperature and particularly at low strain correspond to high Schmid factor orientations remains robust. The underlying mechanisms will need continual review as we collect new data.*"

We have done a whole series of quantitative Weighted Burgers vector (WBV) analyses on our EBSD data. However, we decided to pull all the WBV analyses out from this paper because we found a strong stereological effect, i.e. effects of different 2-D surfaces chosen from the same 3-D sample, on the GND statistics. We will present unpublished data only for discussion here. These data are subject for future publication. We conducted pixel-by-pixel WBV analyses on different orthogonal surfaces from the same deformed ice samples.

An analysis of a uniaxially deformed sample with a nearly random overall CPO (PIL165: 3% strain) illustrates the problem (Fig. R1.6). The absolute values of WBV and the relative values of WBV for grains in different orientations change depending upon which surface (normal or parallel to shortening) is being examined by EBSD.

[Figure]

**Figure R1.6**. **(a)** Illustration of three orthogonal surfaces chosen from a sample (PIL165, -30 °C, ~3% strain, $1 \times 10^{-5} s^{-1}$) for WBV analyses. **(b)** Proportion of pixels with the magnitude of WBV ($\|\mathbf{WBV}\|$) higher than 0.0015 μm⁻¹ as a function of *c*-axis angle to compression axis.

In uniaxially compressed and sheared samples, with strong CPOs, the WBV of different texture components depend on the orientation of the sample surface analysed (Figure R1.7). The WBV of the of the distinct c-axis maxima (~ 45 degrees to compression and normal to shear plane respectively) depend on the orientation of the surface examined (Table R1.1).

[Figure]

**Figure R1.7**. Differences between planes parallel or perpendicular to compression or shear in the proportion of basal-component pixels with $\|\mathbf{WBV}\|$ higher than 0.0015 µm$^{-1}$ as a function of *c*-axis angle to compression for **(a)** PIL177, sample deformed with uniaxial compression at -10 °C to ~12% strain with a strain rate of $1 \times 10^{-5} s^{-1}$, and **(b)** PIL267, sample deformed with direct shear at -30 °C to a shear strain of ~1 with a shear strain rate of ~$1.8 \times 10^{-5} s^{-1}$.

**Table R1.1**. WBV statistics of grains at easy slip orientations from orthogonal surfaces

| Sample No. | Surface type | *c*-axis orientations included in analysis | Proportion of pixels with the magnitude of WBV ($\|\mathbf{WBV}\|$) higher than 0.0015 µm$^{-1}$ | Proportion of pixels dominated by <a>-component WBV within the population of pixels with $\|\mathbf{WBV}\|$ higher than 0.0015 µm$^{-1}$ |
|---|---|---|---|---|
| PIL177 | Parallel to compression | 45°±5° to compression axis | 9% | 42% |
| | Perpendicular to compression | | 3% | 31% |
| PIL267 | Shear plane | 0°-30° to compression axis | 10% | 25% |
| | Profile plane | | 26% | 51% |

The statistics of WBV data are different when the same sample is looked at using different imaging surfaces. (e.g. shear plane vs profile plane). A running hypothesis is that the dislocations are mostly arranged in planar subgrain boundaries and the frequency of observation depends on the orientation of the grain relative to the observation surface. The orientation of the subgrain boundary is a function of grain orientation and Burgers vector.

Stereological effects need to be taken special care of when quantifying GNDs from the EBSD data acquired from a single 2-D sample surface. Conclusions derived from GND calculations can be strongly biased by different imaging surfaces. It is not straightforward to compare different texture components where the relative orientation of c-axes and imaging surface are different. Note that the data and conclusions in (Journaux et al., 2019) will probably not be compromised by this effect as, in the profile plane in which the samples were analysed, the M1 and M2 maxima have identical relative orientations of c-axis and analysis surface.

**13**. P14 l. 20: GBM instead of GMB

This mistake has been corrected.

**14-1**. About GBS and apparent texture weakening in small grains: to my point of view, this apparent texture weakening could be related to the nucleation process itself, and the fact that close to GBs, local misorientation can be high, and induce nucleation orientations varying from parent grains orientations (by bulging or subgrain rotation). This process would be enough to justify the small difference in texture concentration in small grains (that could also be due to more spread in data as there are less pixels measured in small grains, since GBs are interfering with the measurement, reducing its quality in small grain areas ?). See for instance the work of Falus et al. 2011 about Olivine for rotation recrystallization or Chauve et al. 2017 for the orientation of nucleus formed by bulging.

If "spontaneous" nucleation, driven by the relaxation of the dislocation-related internal stress field, can produce nuclei with orientations not related to their corresponding parent grains (Duval et al., 2012), we agree that this could lead to a weaker CPO. For this reason, we have included this as an alternative explanation to the GBS idea.

We added new statements in section 4.1.4: " "*Spontaneous" nucleation driven (Duval et al 2012) by the relaxation of the dislocation-related internal stress field may produce nuclei with orientations not related to their corresponding parent grains (Falus et al., 2011; Chauve et al., 2017), and thus lead to a weaker CPO.'... 'Both hypotheses— "spontaneous" nucleation and GBS—explain a weakening of CPO in "small" grains and these two ideas are not mutually exclusive. Further work is needed to test both hypotheses. Most critical are experiments where nuclei can be observed whilst they are very small and subsequent misorientations can be documented, as might be possible with 3-D microscopy methods (Lauridsen et al, 2003; Poulson et al., 2004), and experiments where fiducial markers are used to confirm the physical existence of offsets on grain boundaries (Schmid et al, 1977; Spiers 1979 ; Beeré, 1978; Eleti et al., 2020).'* "

The data in Chauve et al (2017) can be interpreted equally well by GBS as by spontaneous nucleation and bulging, as was pointed out by the reviewer (Prior) of that paper. Falus et al is one of the few papers in the geoscience world that interprets weakening CPO with reduced grain size as related to a spontaneous nucleation process. There are many more papers (excluding our papers) (Cao et al., 2017; Czertowicz et al., 2016; Kaczmarek and Tommasi, 2011; Linckens et al., 2015; Ohuchi et al., 2015; Park and Jung, 2017; Skemer and Karato, 2008; Skemer et al., 2010; Warren and Hirth, 2006; Warren et al., 2008; Zhao et al., 2019) that interpret almost identical data in terms of the operation of GBS. We think the best way forward in our paper is to make sure that the factual observations are clear and to present both ways (GBS and spontaneous nucleation) that have been used in the literature to interpret similar data.

We don't think that the measurement of fewer pixels in the smaller grains makes any contribution to the weaker CPOs identified. The CPOs are weaker irrespective of whether all

pixels are used or one point per grain. They are weaker if we choose a random subset of grains so that the number of "big" and "small" grains are the same.

**14-2** The work of Qi et al. 2017 mentioned several times in this part concluded that "the dominant mechanism of CPO development occurs with increasing stress, from GBM, which consumes grains with low Schmid factors, at low stress, to the rotation of basal slip planes to an orientation normal to the compression axis at high stress, due to dislocation glide." I didn't find any mention of "grain size sensitive mechanism" as certified l. 25...

Such a grain size sensitive mechanism should be verified by varying grain size during the experiments and evaluate its effect on a given parameter, such as peak stress, strain-rate or so. I maintain that there is no proof of such a GSS mechanism in the experiments presented here, and therefore the interpretation should be cleared about that. That GBS is more active in smaller grains is well known since Boullier and Gueguen work! It does not mean that it should occur in the specific case here, unless otherwise proven...

We agree that rotation of slip planes is a key process in CPO evolution and hopefully our revisions make that much clearer. If GBS occurs it is additional to lattice rotations related to dislocation activity. In the work referred to (Qi et al., 2017: that involves three of the co-authors of this paper) we did not segment the data in a way that required us to bring in interpretations such as GBS, nor spontaneous nucleation. We were always of the view that GBS could be important, as that paper does show the grain size sensitivity of peak stress data. However, it was really the work published by (Craw et al., 2018) that highlighted for the first time an extreme (in that case) difference between CPOs at different grain sizes. GBS is an integral part of the interpretation in that paper and was included in (Qi et al., 2019) to explain some of the features of shear CPOs that are not easily explained by basal slip or dynamic recrystallisation.

**14-3** The hypothesis that GBM being less active at low temperature, the impact of grain rotation driven by intracrystalline slip prevails is much clearer, especially since it is very coherent with the observations that the cone angle is reduced, and more orientations are found close to the vertical. This assertion is, indeed, justified by the experimental observations. This is, in fact, the main "novelty" of the presented work and should be emphasised more. Speculation about GBS tends to lessen this message, and also the interest of the good quality observations performed in this work.

We agree with this and hopefully the revised manuscript makes this clear.

**15** During dynamic recrystallization, weakening is classically (see Humphreys and Haterly 2001 or 2004 for instance, Sakai et al. 2014) attributed to the reduction of hardening based on GBM and nucleation of grains, both reducing the stored strain energy associated with dislocation pile-up or dislocation structures. Therefore dynamic recrystallization induced weakening does not require the interplay of CPO or grain-size sensitive mechanism to be explained. Another point for this consideration about weakening: the relative weakening at about 20% strain is similar for every temperature cases, at about 35% (Sigma_p – Sigma_f/Sigma_p). Therefore there is not more weakening with small grains that without... It should rule out the hypothesis of a grainsensitive mechanism to explain weakening. Nucleation and GBM (each one having different relative influence depending on the temperature) are enough to explain the observed weakening, as expected from the dynamic recrystallization literature.

We agree with the reviewer that balance between GBM and nucleation can also explain the mechanical weakening and it is important to add this into the discussion. One key issue we want to be clear about is that CPO development is not necessarily the key process controlling

weakening (or enhancement): an idea that seems prevalent among the ice sheet modelling community. We modified the statements in section 4.2: " '*All experiments show weakening after peak stress. Weakening is classically observed during dynamic recrystallization, and it has been attributed to a balance between GBM and nucleation of new grains (Montagnat and Duval., 2000; Sakai et al., 2014). In this study, mean and median ice grain size reduces with strain at all temperatures (Table 3, Fig. 11(a)). Grain size is commonly reduced during rock deformation in the laboratory (e.g. Pieri et al., 2001; Hansen et al., 2012) and in nature (Trimby et al., 1998; Bestmann and Prior, 2003). At smaller grain sizes the strain rate contribution of grain size sensitive (GSS) mechanisms increases or the stress required to drive a given strain rate contribution of GSS decreases.'... 'Therefore, further studies are required to quantify: (1) the contribution of nucleation and GBM to the total stress drop if the balance of GBM and nucleation is considered as the weakening mechanism; (2) The contribution of grain size insensitive, e.g. dislocation creep, and grain size sensitive processes, e.g. GBS, to the total stress drop if grain size reduction is considered as the weakening mechanism.'* "

**16** Point 2: from figure 2, the steady state is not so obviously reached, unless, maybe at -10°C. Maybe the authors should be more careful about it, especially about mentioning it in the conclusion.

We have modified the statement to: "In all samples stress rises to a peak stress at ~ 1 to 4% strain and then drops to lower stresses at higher strains."

**17** Point 3: regarding my previous comments concerning the evaluation of a subgrain size, I think that either the authors explain very clearly how they evaluate this subgrain size, and show that it is meaningful based on their experimental observations (that they do observe a subgrain network, although it does not appear clearly in the given figures, from which extracting a subgrain size appears relevant), or this parameter, even if used in the discussion with care, should not appear in the conclusion.

We removed WBV analyses. Instead, we added subgrain boundary analyses by highlighting subgrain boundaries at where the misorientations between neighbouring pixels are between 2° and 10° (Fig. 4(a-c), 5(a-c) and 6(a-c) in modified manuscript). Many of the subgrain boundaries appear to be straight, with some variations around the straight shape. The subgrain boundaries close to bulged grain boundaries are more curved. An interconnection of subgrain boundaries can be observed in some of the grains.

We didn't make it clear that the measurement of subgrain sizes were not based on the data of WBV, and they were based on the misorientation between adjacent pixels. Therefore, the new subgrain boundary plots corresponds to the original subgrain calculations.

We modified the point 3 to: "*All deformed samples develop distinct subgrain boundaries and show a peak at 2°-3° in neighbour-pair misorientation angle distribution. Mean/median subgrain size is smaller than mean/median grain size. These observations suggest recovery and subgrain rotation were active in all deformed samples.*"

**18** Point 5: once again, this conclusion makes use of the subgrain size which measurement method is not clear, and therefore should not be used in the conclusion unless clarified.

We have removed this conclusion since description and discussion of subgrain size have been strongly reduced.

**19** Point 6: I think that there is nothing really new in this point... it has been demonstrated for many materials undergoing dynamic recrystallization, and it is a direct evidence from energy

considerations... Should it really come as an important conclusion? At least, the authors should be care to mentioned "as already observed", or "as expected during dynamic recrystallization"...

We have removed boundary lobateness analyses. This is an issue for a different readership.

**20** Point 7: based on my comments concerning part 4.2, the mention of GBS to explain weakening should be removed. It is also surprising that an hypothesis that is only briefly mentioned in a very short paragraph (4.2), could come to an important conclusion point...

See responses to 8-1, 8-2 and 14-2.

**21** Point 8: same as point 7, and please note that weakening should be measured relatively to the peak stress value (for instance), and it therefore leads to very similar weakening for all temperature conditions (about 35%).

We have removed point 8.

21 - In general, there is a lack of references from the work done on recrystallization (on ice and other materials) by others authors than the authors' team.... this is especially true, for instance, in part 4.1.3, and this should be corrected. In particular when other's work do not come to similar conclusions as the authors...

We have included additional references in the modified manuscript.

22. - Maybe related to this lack of references, some assertions are given with too few justifications, that should come either from experimental observations or from previous works. This should be corrected, and the authors could specify that they are making hypotheses when there is no existing justifications.

We have added more references on concepts that are not clarified. We have specified that we are making hypotheses or interpretations wherever that is the case.

23 - This work does not contain any significant novelty, but provides more detailed and accurate observations at the microstructure scale compared to previous (old!) measurements performed by Jacka and co-authors for instance.

Compared to the extensive literature about dynamic recrystallization at hot temperature (see for instance Humphreys and Haterly 2001 or 2004), there is no novelty, and this literature should be mentioned, especially within the discussion, in order to help the interpretation of the results.

This paper include quantitative microstructural analysis of ice deformed at -20 and -30 °C to progressively higher strains. Such data have never been presented before. To our point of view, these new data are novel. See the comments in section one (General statements)

We present the opening angle evolution of the cone-shaped *c*-axis CPO between this study and previous work. This work had not been systematically done before. The summary view of observations of open-angle evolution with strain as a function of temperature (and ultimately also as a function of stress/ strain rate) is crucial as a test of hypothesis for the deformation and recrystallisation mechanisms that control ice microstructure and ice mechanics.

Last but not the least, we added more data, including misorientation analyses and quantification of repeat counted grains in 2-D using line interception method (done before) and full crystallographic data (completely new) to provide a more detailed microstructural analyses, and to support hypotheses.

24 - The high quality observations enable to assert more clearly some mechanisms as important in the case of recrystallization in ice as, for instance, the fact that at low temperature,

intracrystalline rotation will prevail on GBM and therefore induces texture that are closer to the one observed along deep ice cores.

We agree with the reviewer on this point. We hope we have done this in the modified manuscript.

**25** - It is not clear, all over the text,why the authors want or need to mention GBS as an impacting mechanisms since the experiments performed show absolutely no proof of it, neither in macroscopic data (dependance of peak stress on grain size for instance), nor in microscopic observations. The only observation of small grain necklaces (but limited in number) at the lowest temperature, and a weaker texture in this small grain population is not sufficient, to my point of view, to assert the occurrence of GBS. It could be mentioned as one of the hypothesis among others, but not come to the conclusion as the mechanism at play. In particular, the use of GBS is not necessary to explain stress weakening and does not appear coherent with the results.

See responses to 8-1, 8-2 and 14-2.

26 - I raise again the point about the lack of proper explanation concerning the measurement of subgrain size in the specific case the presented experiments, since the figures shown do not reveal any proper subgrain structure that could be characterized by a dimension (as a mean size for instance). Since different conclusion are taken out of this subgrain size evaluation, it should be corrected before any publication.

We agree with the reviewer. We have provided new subgrain boundary maps and new subgrain/grain size data. We also strongly reduced data and discussion related to subgrain size.

27 - the authors make no use of their observations from the WBV method neither in the discussion, nor in the conclusion... Should it remain in the paper?

We have removed WBV analyses. We have done a whole series of quantitative Weighted Burgers vector (WBV) analyses on our EBSD data. However, we decided to pull all the WBV analyses out from this paper because we found a strong stereological effect, i.e. effects of different 2-D surfaces chosen from the same 3-D sample, on the GND statistics. Please refer to response to comment 12 for details.

The key objective of this paper is to report the detailed changes in microstructures and CPOs to progressively higher strains at low and high temperatures, with the very new data being at lower temperatures. The interpretation of GBS is not central to this and we have downplayed that in the revised manuscript. We still wish to explain the weakening of CPOs in finer grain sizes and have presented two alternative interpretations; GBS and spontaneous nucleation.

We agree with reviewer's comment that our data cannot prove the existence of grain boundary sliding. We would love to have fiducial marker evidence to show directly the GBS effect (e.g.(Eleti et al., 2020; Schmid et al., 1977; Spiers, 1979): this is a significant technical challenge for now. The particular set of experiments presented in our paper does not include variable initial grain size. However, comparable experiments do show grain size sensitivity. The set of -10 °C experiments published by Qi et al 2017 have two different initial grain sizes. A plot of strain rate against the peak stresses (Fig. 3, copied below as Fig. R2.1) shows two different best fit lines for the two initial grain sizes. At peak stress (~ equivalent to min strain rate) grain size is unlikely to have changed substantially from the starting material (and we have some new experiments to peak stress only that show this to be correct). The easiest interpretation of the Qi et al (2017) mechanical data is that there is grain size sensitivity, which is consistent with the operation of GBS.

[Figure]

**Figure R2.1.** Plot of strain rate versus stress on logarithmic scales using data from Qi and others (2017).

If GBS does occur, our interpretation is that this is in addition to dislocation glide (Drury et al., 1985; Gifkins, 1976, 1977; Goldsby and Kohlstedt, 1997, 2001; Hirth, 2002; Hirth and Kohlstedt, 2003; Kuiper et al., 2019a; Kuiper et al., 2019b; Langdon, 2006, 2009; Warren and Hirth, 2006). Some authors term this dislocation accommodated GBS or "disGBS". In this mechanism, the total strain rate is the addition of a dislocation process (that changes crystal shapes and causes lattice rotation and internal distortion) and a GBS process that is probably controlled by a "viscous" mechanism within grain boundaries (small path length diffusion and/ or asperity plasticity: idea originally from (Gifkins, 1976)). This is not the same as diffusion creep, irrespective of whether that is controlled by lattice diffusion (Nabarro-Herring creep) or grain boundary diffusion (Coble creep). GBS is required as an accompanying mechanism to polycrystalline diffusion creep, but in that case grain shape change is facilitated by the diffusive mass transfer process. In diffusion creep, grain size sensitivity comes primarily from the increased path length for diffusion meaning that the change of shape of bigger grains takes longer. In "disGBS" the GBS itself is the prime source of grain size sensitivity. If there is a "viscous" grain boundary volume then the rheology will depend on the volume proportion of the sample that comprises grain boundaries: this proportion will increase with decreasing grain size.

CPO models certainly do not match observations fully for shear (see discussion in Qi et al., 2019) and that paper speculates that GBS may bridge the gap between the results of laboratory experiments and numerical models. Indeed, there is currently a major effort (led by Sandra Piazolo and colleagues) among the community that use the ELLE modelling platform to incorporate GBS: a difficult task. Microstructural modelling is beyond the scope of our paper.

2. The mechanical tests (figure 2) essentially show a dominant temperature effect (known since the early years of glaciology – do the associated activation energy, not calculated here, matches literature data ?) and a softening at strain larger than ~0.03.

Yes, the mechanical data match literature data. We have added a calculation of activation energy to the supplementary information and have referred to this in the text. Best fit to all data (-10, -20 and -30 °C) give activation enthalpies of 98 kJ/mol and 103 kJ/mol from peak and final stress data assuming n=3 and 131 kJ/mol and 138 kJ/mol from peak and flow stress data assuming n=4. These values are close to reported Q values of 71-124 kJ/mol (-5 °C- -30 °C) from Budd and Jacka (1989) and ~133 kJ/mol (-1.5 °C- -12.8 °C) from Glen (1955) and 64-250 kJ/mol from Kuiper and others (2019a, 2019b). Note experiments in this study only cover three temperature values. Hence, the calculated Q values are prone to error. More data points are needed for a more accurate Q investigation.

3. Along the same line, the sentence (p12 line 2) "gbs is kinematically required for all grain size sensitive mechanisms" is incorrect. For example, the Hall-Petch mechanism is largely used in metallurgy to explain size effect observed in many nanometric grains metallic alloys. Hall-Petch is based on the mean free path of dislocations, it explain very well many observations, and does not require any other mechanisms than dislocation glide (no gbs!). Could the mean free path of mobile dislocations have an influence of ice rheology at low temperature ?

Our apologies; the reviewer is correct. That statement does not apply to the full breadth of GSS mechanisms including classic Hall-Petch and also mechanical twinning (Rowe and Rutter, 1990) and we have removed the statement.

As an aside there is a very interesting ongoing discussion of the Hall-Petch (Weertman, 1993) relationship (with strength increasing with grain size) and the inverse Hall-Petch relationship (Masumura et al., 1998) in the materials science literature (Pande and Cooper, 2009; Ryou et al., 2018; Sheinerman et al., 2020). Modelling of the inverse Hall-Petch relationship requires coupling of GBS to intragranular dislocation activity (Carlton and Ferreira, 2007; Ehre and Chaim, 2008; Padmanabhan et al., 2007; Padmanabhan et al., 2014; Ryou et al., 2018; Sheinerman et al., 2020) and the relationships are not very different to those described elsewhere as GBS accommodated by dislocation creep (Goldsby and Kohlstedt, 1997; Hansen et al., 2011; Langdon, 2006, 2009). In minerals, the normal Hall-Petch relationship (increasing strength with decreasing grain size) has only been documented at low homologous temperatures (Hansen et al., 2019; Koizumi et al., 2020) whereas weakening with reduced grain size is the norm at higher temperatures and lower stresses (Brodie and Rutter, 2000; De Bresser et al., 2001; Hiraga et al., 2013; Hirth, 2002; Hirth and Kohlstedt, 2003; Schmid et al., 1977; Ter Heege et al., 2005; Walker et al., 1990). Materials science work defines a material-dependent threshold grain size, above which the Hall-Petch relationship holds and with the inverse Hall-Petch relationship at grain sizes below the threshold (Pande and Cooper, 2009; Ryou et al., 2018). Recent work suggests that the threshold moves to larger grain sizes at lower strain-rates or stresses (Somekawa and Mukai, 2015). The rates that are considered very slow in these metallurgical analysis (e.g. 1 x $10^{-4}$ $s^{-1}$) are very fast in the context of geological or glaciological laboratory experiments and this may explain why we only see evidence of the Hall-Petch effect at low homologous T. Some recent work relates GBS associated with the inverse Hall-Petch relationship with amorphization of the grain boundaries (Guo et al., 2018) and a molecular dynamics modelling study of ice (Cao et al., 2018) generates an inverse Hall-Petch relationship that involves a combination of GBS, grain rotation, amorphization and recrystallization, phase transformation, and dislocation nucleation in both bicrystals and polycrystals.

4. Similarly, about the sentence (p15 line 6) "similarly, we suggest that grain size sensitivity of gbs favours a faster strain rate in small grains": I find no fact in the results supporting this assertion. Strain-rate in various sets of grains is not measured nor estimated here. And also, in section 4.2, the authors make a correlation between the softening observed at -30degC and the grain size, and conclude that the observed softening should likely be attributed to gbs. Gbs could be a possibility, but among many others. For example, what do we know about the density of mobile dislocations ?? If it increases, the stress would decreases as observed. Increase of dislocation density is often used to explain the peak stress for materials with low initial dislocation density (eg. Si, . . .).

Please see our answer to point 1.

5. The statistical relevance of the performed mechanical tests and/or microstructural investigations can also be questioned. Figures 9, 10, 11, 13 show pole figures that do not, by far, exhibit the expected transverse isotropy (expected since the initial specimen are thought to exhibit random CPO with equiaxe grain shape, and since uniaxial compression is transverse isotropic). This severe lack of symmetry in the observed microstructure can originate from (i) initial samples that do not exhibit a random microstructure and/or (ii) mechanical tests that deviate from uniaxial compression (there could be many reasons for that) and/or (iii) the microstructure is not analysed on a sufficiently large material volume (volume smaller than the Representative Volume Element-RVE). Consequently, the global picture shown here (ex. texture strength as function of temperature, which is an interesting result) are probably correct, but I don't think that, with the results shown, authors can dig deeper into the interpretation of active deformation mechanisms. If the lack of texture symmetry is present in the specimens,

then the applied axial strain-rate would generate significant shear stress (or shear-rate, depending on the experimental boundary conditions), affecting of course the texture and microstructure evolutions (so-called out-of-axis tests). Is there any connection with the large spread observed on the mechanical responses (figure 2) ? For example the peak stress at -10 C varies by almost a factor 2, which is considerable and should be discussed. One could expect some associated spread in the microstructure.

The reviewer is correct about symmetric incompleteness and we have added the following text to address this: *"Many deformed samples exhibit an incompleteness of c-axes cone (lack of cylindrical symmetry) (Fig. 8-10). The incompleteness of c-axes cone is more severe for 5 μm EBSD maps collected from a much smaller area than 30 μm EBSD maps (Fig. 12). These phenomena are common to all ice CPOs from measurements on a single sample planes (by EBSD or optical methods: see any of the papers cited), but are not so apparent in neutron diffraction data (Piazolo et al., 2013; Wilson et al., 2019), that sample a larger volume, suggesting that a single plane through a deformed sample does not generally contain sufficient grains for a fully representative CPO."*

The fact that neutron diffraction data gives CPOs that have close to the cylindrical symmetry, for samples that have fewer grains (initially) in an average cross section (Piazolo et al 2013 initial grain size 0.5mm whereas ours <0.3mm: samples in both cases 1 inch diameter) suggests that the sample as a whole has enough grains to b considered mechanically isotropic. In this case the incompleteness of CPOs is an analytical sampling issue and should not impact on mechanical data. A good example of where samples contain too few grains to be considered isotropic is the re-deformation of natural ice with a 20mm grain size (Craw et al., 2018): this gives rise to significant inconsistency in stress strain curves, although yield stress data correlate sensibly with strain rates.

The scatter of peak stress values we have is fairly typical of confined medium constant displacement rate experiments (data for comparison can be extracted from (Durham et al., 1983; Golding et al., 2020). Unconfined constant displacement rate experiments (Hammonds and Baker, 2016; Vaughan et al., 2017) have less variability and it is likely that some of the scatter in confined medium experiments relates to how stable the confining pressure is. Unconfined creep experiments (constant load) also show a range of minimum strain rates for a set of experiments at the same stress (Journaux et al., 2019; Montagnat et al., 2015; Treverrow et al., 2012). To compare constant rate vs constant load experiments, we can calculate the "viscosity" at peak stress/ minimum strain rate. Confined constant rate and unconfined creep tests both have "viscosities" that vary by up to about 2x for experiments at the same rate or stress. Unconfined constant rate experiments have peak stress "viscosities" that vary by up to about 1.1x. These statements are made on a relatively small data set as there seem to be few "repeat" experiments (in terms of load or rate) in the published literature. At the moment we don't have a full explanation as to what controls this variability. We have to account for the variability in studies where it becomes important (e.g. for calibrating flow laws). In this paper it is not so important and the aspect that is important to us – the curve shape with a peak stress followed by weakening is common to all experiments.

6. The discussion in this paper relies on a separation of the grain size distribution between "small grains" and "large grains", invoking a "bimodal" (p7 line 4) grain size distribution. In figures 3, 4, 5, I do not see any bimodal grain size distribution, but rather a unimodal one with a long tail. Therefore the size threshold (p7 line 16) used to separate small and large grains is completely ad hoc, and I am not sure about the effect of this particular choice on the provided discussion. I also don't understand why the authors state that "The small grains are likely

include all the recrystallized grains" (p7 line 19, p8 line 8, . . .) as (i) if GBM occurs, recrystallization can also lead to large grains and (ii) the grain size distribution of the initial microstructure is not shown.

(1) We modified the description of grain size distribution in section 3.3.2: *"For samples deformed to ~3% strain, the grain size distributions are strongly skewed or possibly bimodal, with a clear main peak at finer grain sizes and a tail of coarser sizes with a broad, poorly defined secondary peak corresponding to the mean grain size of the starting material (Fig. 4(d), 5(d) and 6(d))."*

(2) We removed the statement of: "The small grains are likely include all the recrystallized grains." This is a very good point from the reviewer. We (who come from the rock deformation world) sometimes forget that at the high homologous temperatures in ice recrystallised grains can grow to a large size. In much lower homologous temperature experiments in quartz, for example, it is reasonable that recrystallised grains are small and remnant grains large (see for example (Cross et al., 2017; Hirth and Tullis, 1992). We still wish to segment the grain size on the basis of "big" and "small" grains and we hope that our presentation of this is now more robust and does not assign arbitrarily the status recrystallised or remnant on certain grain size populations. The precise threshold we use does not influence the difference in CPOs between "big" and "small" grains as shown in Fig. R2.2, extracted from new supplementary information.

[Figure]

**Figure R2.2**. The contoured *c*-axis CPOs of "big" and "small" grains in samples deformed at **(a)** -10, **(b)** -20 and **(c)** -30 °C to ~12% strain. "Big" and "small" grains are separated using the threshold of mean grain size (row 1), SMR (square mean root) grain size (row 2), median grain size (row 3) and peak grain size (row 4). Number of grains and M-index value are marked at the bottom left corner of the corresponding *c*-axis CPO.

7. The discussion also largely relies of the size of subgrains. However, in figures 3, 4, 5, it is really hard to identify those subgrains in most of the grains. For example in figure 5 at 20% strain, one only sees some disconnected segments in the WBV map in the large yellow or pink-orange grains at the bottom of the micrograph. How do the authors identity the subgrains and calculate their size in such a case ?

We agree with the observation from the reviewer that suggests many of the subgrain boundaries are straight tilt bands or kink bands. The subgrain structure was revealed by Weighted Burgers vector (WBV) method, which picks up pixels with the WBV magnitude ($\|\mathbf{WBV}\|$) higher than 0.0026 $\mu m^{-1}$ (equivalent to misorientation angle between neighbouring pixels higher than ~0.7°). Therefore, many of the subgrain boundaries lower than 2° were selected and they might contain non-neglectable errors (Prior, 1999). Moreover, we didn't make it clear that the measurement of subgrain sizes were not based on the data of WBV, and they were based on the misorientation between adjacent pixels. Therefore, the new subgrain boundary plots corresponds to the original subgrain calculations. We kept the WBV analyses based on the thinking that they might contain more information for further comparison. But the WBV analyses have now been removed completely from this paper.

The new maps (Fig. 4(c), 5(c), 6(c)) that show subgrain boundaries correspond to the much simpler misorientation threshold. We modified statements on section 3.3.1 to: "Distinct sub-grain boundaries can be observed in all the samples (Fig. 4 (c), 5 (c) and 6 (c)). Many of the subgrain boundaries appear to be straight, some with slight curvature. A small number have strong curvature. Interconnected subgrain boundaries can be observed in some of the grains. Subgrain boundaries subdivide grains into subgrains."

8. This experimental study cannot be used without very special care to infer deformation mechanisms occuring in "terrestrial and planetary ice flow" (1st abstract line), as (i) the grain size investigated (~200 microns) is one order of magnitude smaller than the natural one, and (ii) the strain-rate used during the mechanical tests (10-5s-1) is 3 to 6 orders of magnitude larger than in cold regions of ice sheets.

We modified the first line in abstract to: "To understand better the ice deformation mechanisms…" The reviewer raises the key problem that we struggle with, when we are working in the laboratory with application to natural ice. The absolute fastest documented natural terrestrial strain rates are in lateral shear margins ~ $10^{-9}$ s$^{-1}$ (Bindschadler et al., 1996; Jackson and Kamb, 1997). Rates in basal ice is harder to estimate; most models would have strain rate maxima also around ~ $10^{-9}$ s$^{-1}$. Most parts of ice sheets and glaciers have strain rates that are up to 2 orders of magnitude slower than this. To run an experiment from to 10% strain (i.e something that may go from isotropic starting material to a "steady state" microstructure) will take three years at ~ $10^{-9}$ s$^{-1}$. (Jacka and LI, 2000) did an amazing job running experiments for long durations at low rates (down to 4 x $10^{-10}$ s$^{-1}$) but these are really the only experiments that achieve substantial strain at "natural" rates. Specific aspects of ice mechanics have been assessed by deforming natural samples to small strains (<1%) in the lab at relatively slow rates ($10^{-10}$ s$^{-1}$ to $10^{-8}$ s$^{-1}$) (Castelnau et al., 1998; DahlJensen et al., 1997; Jackson and Kamb, 1997). In general, it is virtually impossible to work at natural rates and we have to develop scaling relationships that involve strain rate, temperature and grain size.

9. I wonder whether there is no damage occurring at the high strain-rate considered, particularly at the smaller temperatures?

Stress-strain curves of all experimental runs show a smooth and continuous increase of stress as a function of strain before reaching the peak (Fig. 3). The stress-strain curves of experiments with a development of cracking during deformation normally show an initial yield point before reaching the peak stress (Mellor and Cole, 1982). The initial yield point is interpreted as a reflection of cracking on the mechanical data (Mellor and Cole, 1982). Such yield point is not observed in any of the experiments in this study.

The chief purpose of the confining pressure in these experiments is to suppress brittle phenomena including cracking and frictional sliding. Fig. R2.4 shows the experiment with the highest differential stress, plotted on a Mohr diagram for stress. The green circle shows the shear and normal stresses for surfaces of all orientations and the maximum ($\sigma_1$) and minimum ($\sigma_3 = \sigma_2 =$ confining pressure) plot along the line of zero shear stress. Superposed are two failure envelopes. One is a Coulomb (frictional sliding) envelope using the friction coefficient for ice-ice sliding from (McCarthy et al., 2017). Coulomb envelopes usually underestimate brittle strength. The second failure envelope is the composite envelope from (Beeman et al., 1988). Red and blue Mohr circles show the stress states needed for brittle failure at 20MPa pressure with each of these envelopes. Maximum differential stresses applied in our experiments are substantially below those for needed for brittle failure.

[Figure]

**Figure R2.3**. Typical stress-strain curve for deformed sample with cracking (from Mellor and Cole, 1982)

[Figure]

**Figure R2.4**. Mohr diagram showing stress state of sample PIL164 (the largest differential stress) in green. A coulomb failure envelope using a friction coefficient of 0.29 from (McCarthy et al., 2017) is shown with a red dashed line and the Mohr circle for failure at 20MPa confining pressure is shown in red. The blue lines show the (Beeman et al., 1988) failure envelope from and the Mohr circle for failure at 20MPa confining pressure is shown in blue.

10. p1 line 16 : "displacement rate" instead of "displacement"

Corrected.

11. p1 line 26 : invoking creep stages (secondary, tertiary) for the description of constant strain-rate experiments is misleading.

We have deleted these misleading wording.

12. P5 line 26, I don't understand what is meant with "The CPO data were contoured with half-width of 7.5deg" ?

We modified the statement in section 2.5.2: "The CPO data were contoured with a half-width of 7.5° based on the maximum of multiples of a uniform distribution (MUD) of the points, to more clearly show the CPO patterns."

12. p7 line 26 : to the best of my knowledge, recovery, subgrain rotation and gbm are not deformation mechanisms ! If recovery and/or gbm are initiated, the specimen will not deform.

Thank appreciate the reviewer for pointing out this mistake. We have removed this sentence since we removed boundary hierarchy analyses.

13. eq. 3, how is R (grain radius) estimated for non-spherical grains ??

We have removed grain boundary lobateness analyses.

14. p9, line 1 : I think that calling "m" the 10-10 direction is not standard (m-axes pole figures). Should be clarified ?

We corrected "*m*-axes" to "poles to the *m*-planes".

15. p11 line 26, the sentence "Much of the stress increase prior to peak stress relates to elastic strain" is wrong. First of all, there is no known yield stress for the high temperature rheology of ice, i.e. plastic strain starts as soon any stress is applied, as here in the first part of the loading prior to the peak stress. There are old published data (on single and polycrystals) showing that the initial slope depends on the strain-rate. Of course, there is always an elastic strain associated to the applied stress (Hooke's law). On top of that, the measured slope (~1GPa) very probably also accounts for the way strain is measured experimentally: if it is not measured directly on the specimen (eg. with an extensometer or strain-gage), it is well known that very small modulus are obtained, due to machine rigidity and other artefacts.

intracrystalline dislocation slip, the porosity loss being very likely negligible.

Published literature labelled the stress increase prior to peak stress in constant displacement rate experiments as: "normally elastic" (Cole, 1987) and "quasi-elastic" (Kirby, 1987). The deceleration during primary creep in constant stress experiments was interpreted as effected by a "delayed elasticity", with a recoverable component of time-dependent elastic strain and an irrecoverable viscous strain (Mellor and Cole, 1982), and "anelasticity" (Duval et al., 1983). The reason we chose to describe the behaviour as substantially elastic is that we have other experiments where we can show that this part of the deformation is recoverable. However, these other experiments are higher rate experiments with slopes on the stress strain curve approaching the 9GPa modulus. The reviewers are correct in pointing out that in the experiments presented in this paper the slope is substantially below modulus and the behaviour is not substantially elastic. We have modified the statement in section 4.1.1: "*This likely includes anelastic deformation related to intergranular stress redistribution used to explain primary creep in constant load experiments (Duval et al, 1983). The curvature of the stress strain line at the start of each experiment may relate to initial porosity loss as suggested by rapid increases in ultrasonic p-wave velocity in comparable experiments by Vaughan et al., (2017).*"

17. figures 3, 4, 5 : If I understand (this is not clear in the paper), the shown grain size distributions indicate the number of grains at a given size. It would be more instructive to show the volume fraction, not the number of grains, as the rheology is associated with the volume average of grain deformation.

Grain size distribution has been used to show generation of small grains after deformation. These grains are not observed in undeformed grains. We estimated grain volume for each grain size class for modelling the effect of small grains on mechanical weakening. These grain volume data are subject to another paper.

18. figure 14 is interesting, as it shows that the strain-rate seems to have little effect. To my understanding, this is not expected for thermally activated mechanisms such as recrystallization, where time comes in plays. This figure could be more largely discussed, to my point of view.

We plotted data from this study and previous studies in a diagram of $\theta$ as a function of strain with data subdivided with different temperatures and strain rates to increase our understanding of the processes that might control the c-axes cone opening-angle (Table 4 and Fig. 13). The relation to strain rate within the broader data set in this figure is not very clear, because for any given temperature there is not a big range in strain rate. The exception is the data set plotted from (Qi et al., 2017) at -10 °C and ~ 20% strain which does show a rough decrease in $\theta$ as strain rate (or stress) increases (See Qi et al., 2017 fig 9. This fits with the Zener-Hollomon concept (Zener and Hollomon, 1944) that suggests that decreasing strain rate will have an equivalent effect to increasing temperature.

**1.1** "Understanding ice deformation mechanisms is crucial for understanding the dynamic evolution of terrestrial and planetary ice flow." **(P1, L10-15)** revised as:
*"In order to better understand ice deformation mechanisms, we document the microstructural evolution of ice with increasing strain." (**P1, L10-15**)*

**1.2** " 'Mechanical data show peak and steady-state stresses are larger at colder temperatures as expected from the temperature dependency of creep.' … 'At -30 °C, the c-axis CPO transits from a narrow cone to a cluster, parallel to compression, with increasing strain. This closure of the c-axis cone is interpreted as the result of a more active grain rotation together with a less effective GBM. As the temperature decreases, the overall CPO intensity decreases, facilitated by the CPO' " **(P1, L15 – P2, L5)** re-written as:
*"Microstructural data are generated from cryogenic electron backscattered diffraction (cryo-EBSD) analyses. All deformed samples contain sub-grain (low-angle misorientations) structures with misorientation axes that lie dominantly in the basal plane suggesting activity of dislocation creep (glide primarily on the basal plane), recovery and subgrain rotation.' … 'High-angle grain boundaries between small grains have misorientation axes that have distributed crystallographic orientations. This implies that, in contrast to subgrain boundaries, grain boundary misorientation is not controlled by crystallography. Grain boundary sliding of finer grains or nucleation of those grains in random orientations ("spontaneous" nucleation) could explain the weaker CPO of the fine-grained fraction and the lack of crystallographic control on high-angle grain boundaries.' " (**P1, L15 – P2, L11**)*

**2. Modifications in "Introduction"**

**2.1** Add references of " *Pollard, 2010; Kopp et al., 2017 " (**P2, L14**) and "Budd and Jacka, 1989" (**P2, L22**)*

[revised manuscript text omitted]

**4.2** Change Section 3.1 to *Section 3.2* "At strains larger than ~0.1, stresses reduce only a modest amount, with steady-state reached at a strain of ~0.2. Peak and final stresses are larger at colder temperatures and the peak stresses are better defined at -30 °C than at the warmer temperatures." **(P6, L9-11)** revised as:
*"The rate of stress reduction is at a minimum, for each temperature, at strains larger than ~0.1. Peak and final stresses are larger at colder temperatures. Ratios of peak stress to stresses at higher strain (e.g. final stress of ~20% strain) are approximately the same at all temperatures so that all curves, when normalised to the peak stress look similar." (P7, L16-19)*

**4.3** Change Section 3.2 to *Section 3.3*. "EBSD data are used to generate the illustrative grain orientation maps, grain sub-structure maps, as highlighted by WBV analysis, grain size distributions and subgrain size distributions shown in Fig. 3-5." **(P6, L13-14)** modified to:
*"EBSD data are used to generate the illustrative grain orientation maps, grain sub-structure maps, grain size distributions, subgrain size distributions and misorientation angle distributions shown in Figs. 4-6." (P7, L21-22)*

**4.4** "Note that the quantitative microstructural analyses and CPO data are based on larger areas than those presented in the EBSD maps (Table 2)." **(P6, L16-17)** modified to:

[revised manuscript text omitted]

**5 Modifications of "Discussion"**

**5.1** "The stress-strain curves (Fig. 1) at all temperatures first rise to the peak stresses and then relax to approach near-constant stresses with strains." **(P11, L21-22)** revised to:
*"All stress-strain curves (Fig. 3) show stress rising to the peak stress and then relaxing, with the rate of stress drop decreasing with strain."* **(P14, L13-14)**

**5.2** "…and is comparable to the constant-load experiments (Budd and Jacka, 1989; Jacka and Li, 2000; Treverrow et al, 2012; Wilson and Peternell, 2012) where strain rate first decreases to a minimum and then increases to approach a near-constant strain rate" (**P11, L24-25**) revised to:

*"...and has an approximate inverse relationship (Mellor and Cole, 1982, 1983; Weertman, 1983) to constant-load experiments (Budd and Jacka, 1989; Jacka and Li, 2000; Treverrow et al, 2012; Wilson and Peternell, 2012)…"* (**P14, L15-17**)

**5.3** Add sentence *"Stress-strain curves of all experimental runs show a smooth and continuous increase of stress as a function of strain before reaching the peak. Approximately linear portions of the stress-strain data prior to peak have been termed quasi-elastic (Kirby et al., 1987)."* (**P14, L19-21**)

**5.4** "This and the curvature of the stress strain line at the start of each experiment suggests that there is also some dissipative deformation here. This can include porosity loss (Vaughan et al., 2017) and the intergranular stress redistribution used to explain primary creep in constant load experiments (Duval et al, 1983)." (**P11, L27-30**) revised to:

*"This likely includes anelastic deformation related to intergranular stress redistribution used to explain primary creep in constant load experiments (Duval et al, 1983; Castelnau et al., 2008). The curvature of the stress strain line at the start of each experiment may relate to initial porosity loss as suggested by rapid increases in ultrasonic p-wave velocity in comparable experiments by Vaughan and others (2017)."* (**P14, L22-25**)

**5.5** Add sentences *"As our experiments are all at the same approximate strain rate, we cannot calculate the stress dependency of strain rate (the stress exponent, n). Qi and others (2017) calculate a peak stress n value of 3 and flow stress n value of 3.9 for comparable experiments (including PIL007 used here) at -10 °C. Craw and others (2018) calculate a peak stress n value of 4.1 for comparable experiments at -30 °C."* (**P14, L26-29**)

**5.6** Add sentences *"Our peak and final stress data can be used to calculate the activation energy by assumption of a value of stress exponent, n (see section S6 of the supplementary material for the calculation). Best fit to all data (-10, -20 and -30 °C) give activation energy of 98 kJ/mol and 103 kJ/mol from peak and final stress data assuming n=3 and 131 kJ/mol and 138 kJ/mol from peak and flow stress data assuming n=4. These numbers are consistent with published values (64-250 kJ/mol) at relatively high temperature (Glen, 1955; Goldsby, 2001; Budd and Jacka, 1989; Cuffey and Paterson, 2010; Durham et al., 2010; Kuiper et al., 2019a, 2019b)."* (**P15, L1-6**)

**5.7** Remove "The drop of stress after peak correlates with dynamic recrystallization driven grain size reduction and CPO development (Jacka and Maccagnan, 1984; Vaughan et al., 2017; Qi et al., 2019). Experiments with initial grain size as a variable, under comparable conditions to our experiments, suggest that grain size sensitive mechanisms are important (Qi et al., 2017). Grain boundary sliding (GBS) is kinematically required for all grain size sensitive mechanisms (Stevens,1971; Gates and Stevens, 1974), including diffusion creep (Boullier and Gueguen, 1975; Behrmann and Mainprice, 1987) and dislocation slide accompanied by GBS (disGBS) (Warren and Hirth, 2006). Goldsby and Kohlstedt (1997, 2001, 2002) suggest a general importance of GBS on the basis of the constitutive law parameters required to fit the mechanical data from experimentally deformed fine-grained ice. Recent studies suggest GBS in fine-grained ice layers has a key role in controlling the Greenland ice flow (Kuiper et al., 2019a, 2019b) by applying the Goldsby-Kohlstedt flow law (Goldsby and Kohlstedt, 1997, 2001) to modelling the deformation in the NEEM (North Greenland Eemian Ice Drilling) deep ice core. The grain size reduction resulting from dynamic recrystallization is thought to cause mechanical weakening by increasing the strain rate contribution of grain size sensitive deformation mechanisms (De Bresser et al., 2001). A development of strong CPO can also lead to mechanical weakening in viscously anisotropic materials (Durham and Goetze, 1977; Hansen et al., 2012) such as ice." (**P12, L1-11**)

**5.8** Sub-divide section 4.1.2, remove discussions related with boundary hierarchy analyses and boundary lobateness (sphericity parameters). Rewrite section. 4.1.2:

*"**4.1.2.1. Nucleation**

'The number density (number of grains per unit area) of "distinct" grains (counting 2-D grains attributed to the same 3-D grain as one: section S4 of supplementary material) increases by more than a factor of 3 times that of the starting material in all deformed samples at all temperatures (Table 3). We can be reasonably confident that the number of grains in the samples has increased as a function of deformation. This requires a process of nucleation to create new grains. For all the deformed samples, the grain size distributions are characterised by peaks at finer grain sizes, and a smaller mean/median grain size compared with the undeformed sample (Fig. 2,*

*4-6, Table 3). The smallest grains in the deformed samples were not present in the starting material. These observations suggest that nucleation generates the grains with smaller sizes. Grain number density generally increases and all measures of 2-D grain size decrease with strain (Table 3), at all temperatures, suggesting that nucleation operates continuously as part of the recrystallisation process throughout the deformation.'*

**4.1.2.2. Dislocation activity, recovery, subgrain rotation and subgrain rotation recrystallisation**

*'Microstructure maps show subgrain boundaries in all deformed ice samples (Fig. 4(a-c), 5(a-c) and 6(a-c)). The subgrain boundary geometry is comparable with other experimentally or naturally deformed rock and metal samples, e.g. quartz (Cross et al., 2017a; Killian and Heilbronner, 2017), Olivine (Hansen et al., 2012), Magnox alloy (Wheeler, 2009) and Zircon (MacDonald et al., 2013). The misorientation axes for subgrain boundaries are generally rotations around rational crystallographic axes, particularly directions in the basal plane, suggesting that the boundaries may represent arrays of dislocations (Humphreys and Hatherley, 2004; Shigematsu et al., 2006). There is much higher frequency of low angle (Particularly < 10°) neighbour-pair misorientations than are expected from the CPO (as shown by the random-pair misorientation angles).'... 'At all temperatures and strains the mean/median subgrain size is smaller than the mean/median grain size. This indicates that the subgrain rotation recrystallization could be the nucleation mechanism that generates the "small" grain population. Previous studies on deformed metals and quartzites describe the structure of smaller grains encircling larger grains as "core-and-mantle" structure (Gifkins, 1976; White, 1976).' ... 'The network of smaller grains that encircle bigger grains at strains higher than 12% at -20 and -30 °C is consistent with the operation of a subgrain rotation recrystallization mechanism. The network of finer grains encircling larger grains has been observed in deformed metals, and it is named as the "necklace structure" in the material science literature (e.g. Ponge and Gottstein, 1998; Jafari and Najafizadeh, 2009; Eleti et al., 2020). Lately, Eleti and others (2020) used a fiducial marker grid to show that the deformation of finer grains in the necklace structure includes a significant component of GBS.' ... 'Jacka and Li (1994) show an inverse relationship between ice grain size and stress from deformed ice samples that reach tertiary creep.'*

**4.1.2.3. Grain boundary migration**

*'Lobate grain boundaries are commonly interpreted as the result of strain-induced grain boundary migration (GBM) (Urai et al., 1986; Jessell, 1986; Duval and Castelnau., 1995). Samples deformed at -10 and -20 °C show more grains with lobate boundaries at higher strains (>~3%), suggesting more widespread strain-induced GBM with an increasing strain. The proportion of repeated (i.e. interconnected and highly lobate) grains is generally higher in the higher-temperature experiments (Table 2, section S4 of the supplementary material). This observation suggests that GBM is also more widespread at higher temperatures.' "* **(P15, L8 – P16, L29)**

**5.9** Separate section 4.1.3 (Inferences from CPO development) **(P14, L1 – P15, L8)** into two sections of *4.1.3 (CPO development)* and *4.1.4 (CPO development: differences related to grain size). Sections of 4.1.3 and 4.1.4 are mostly newly written.*

**"4.1.3 CPO development**

[revised manuscript text omitted]

**8 Modifications to "Tables"**

**8.1** Add new data of repeat-counted grains and number density of "distinct" grains in Table 3.

**8.2** Add new cone opening-angle data from Hooke and Hudleston (1981) in Table 4.

**9 Modifications to "Figures"**

**9.1** Remove Fig. 6, 7, 8.

**9.2** Add *Fig. 2* -microstructure of undeformed ice sample

**9.3** Add misorientation data to *Fig. 4-6*.

**9.4** Add detailed misorientation analyse of "core-and-mantle" structure in *Fig. 7*.

**9.5** Add distribution of grain size, subgrain size and number density of "distinct" grains as a function of strain in *Fig. 11*.

**9.6** Add CPO data of randomly selected small grains with the same number of big grains in *Fig. 12*.

**9.7** Add natural ice data from Hooke and Hudleston (1981) to *Fig. 13*.

**9.8** Remove bulging in *Fig. 14*.

[revised manuscript text omitted]
}$ (µm) $^7\varphi \geq 2°$ | $^8\bar{D}$ (µm) | $^9D_{median}$ (µm) | $^{10}D_{q,25\%}$ (µm) | $^{11}D_{q,75\%}$ (µm) | $^{12}D_{SMR}$ (µm) | $^{13}\overline{D_{big}}$ (µm) | $^{14}\overline{D_{small}}$ (µm) | $^{15}D_{peak}$ (µm) | $^{16}d_{peak,2°}$ (µm) |
|---|---|---|---|---|---|---|---|---|---|---|---|---|---|---|---|
| undeformed | - | - | 1.90 | 9.97E-06 | 1.00 | 291/280 | 297 | 291 | 165 | 413 | 274 | - | - | 300 | - |
| PIL176 | -10 | 0.03 | 9.45 | 3.24E-05 | 3.25 | 134/79 | 156 | 117 | 48 | 250 | 132 | 250 | 51 | 30 | 20 |
| PIL163 | -10 | 0.05 | 11.71 | 4.75E-05 | 4.76 | 104/77 | 125 | 98 | 54 | 171 | 110 | 197 | 58 | 35 | 25 |
| PIL178 | -10 | 0.08 | 13.47 | 3.82E-05 | 3.83 | 127/108 | 140 | 119 | 72 | 188 | 127 | 194 | 63 | 55 | 50 |
| PIL177 | -10 | 0.12 | 14.19 | 5.14E-05 | 5.15 | 96/77 | 114 | 90 | 54 | 155 | 101 | 184 | 59 | 40 | 30 |
| $^1$PIL007 | -10 | 0.19 | 13.07 | 6.25E-05 | 6.27 | 96/78 | 106 | 88 | 51 | 143 | 96 | 174 | 58 | 50 | 45 |
| PIL254 | -20 | 0.03 | 7.40 | 5.75E-05 | 5.77 | 91/46 | 114 | 62 | 36 | 174 | 93 | 197 | 38 | 25 | 20 |
| PIL182 | -20 | 0.04 | 5.30 | 3.97E-05 | 3.98 | 103/67 | 148 | 122 | 62 | 220 | 131 | 188 | 42 | 30 | 25 |
| PIL184 | -20 | 0.08 | 10.61 | 4.73E-05 | 4.74 | 88/58 | 122 | 89 | 48 | 164 | 105 | 169 | 42 | 45 | 20 |
| PIL185 | -20 | 0.12 | 7.76 | 1.05E-04 | 10.49 | 55/40 | 75 | 53 | 36 | 85 | 66 | 132 | 41 | 30 | 20 |
| PIL255 | -20 | 0.20 | 12.29 | 1.28E-04 | 12.85 | 55/46 | 64 | 53 | 36 | 81 | 59 | 106 | 41 | 30 | 25 |
| PIL165 | -30 | 0.03 | 2.07 | 3.15E-05 | 3.16 | 108/60 | 149 | 108 | 48 | 241 | 126 | 203 | 38 | 40 | 20 |
| PIL162 | -30 | 0.05 | 4.87 | 7.27E-05 | 7.29 | 70/49 | 103 | 76 | 45 | 135 | 91 | 144 | 40 | 35 | 20 |
| PIL164 | -30 | 0.07 | 5.58 | 6.67E-05 | 6.69 | 59/38 | 98 | 61 | 39 | 113 | 82 | 158 | 39 | 30 | 20 |
| PIL166 | -30 | 0.12 | 6.01 | 1.34E-04 | 13.45 | 57/47 | 67 | 54 | 37 | 79 | 61 | 104 | 70 | 35 | 25 |
| PIL268 | -30 | 0.21 | 5.66 | 1.18E-04 | 11.88 | 42/30 | 60 | 37 | 29 | 53 | 50 | 158 | 35 | 30 | 20 |

[revised manuscript text omitted]

| Study | Sample | Medium | Temp | Col5 | Col6 | Test type | Conditions | Value | Final |
|---|---|---|---|---|---|---|---|---|---|
| Qi et al (2017) | PIL7 | $H_2O$ | -10 | N/A | 37 | | $\dot{\varepsilon} = \sim 1 \times 10^{-5} s^{-1}, \varepsilon = 18\%$ | $1.10 \times 10^{-5}$ | 18.0 |
| | PIL32 | $H_2O$ | -10 | N/A | 34 | | $\dot{\varepsilon} = \sim 2 \times 10^{-6} s^{-1}, \varepsilon = 21\%$ | $2.31 \times 10^{-6}$ | 21.0 |
| | PIL33 | $H_2O$ | -10 | N/A | 26 | Constant displacement rate | $\dot{\varepsilon} = \sim 2 \times 10^{-4} s^{-1}, \varepsilon = 22\%$ | $2.42 \times 10^{-4}$ | 22.0 |
| | PIL35 | $H_2O$ | -10 | N/A | 35 | | $\dot{\varepsilon} = \sim 1 \times 10^{-5} s^{-1}, \varepsilon = 13\%$ | $1.35 \times 10^{-5}$ | 13.0 |
| | PIL36 | $H_2O$ | -10 | N/A | 34 | | $\dot{\varepsilon} = \sim 5 \times 10^{-5} s^{-1}, \varepsilon = 19\%$ | $5.02 \times 10^{-5}$ | 19.0 |
| Vaughan et al (2017) | def013 | $H_2O$ | -5 | 206641 | 42 | | $e = 3\%$ | $1.03 \times 10^{-6}$ | 3.0 |
| | def012 | $H_2O$ | -5 | 309428 | 36 | Constant displacement rate | $e = 5\%$ | $1.05 \times 10^{-6}$ | 5.1 |
| | def011 | $H_2O$ | -5 | 218653 | 38 | $\dot{e} = \sim 1 \times 10^{-6} s^{-1}$ | $
[revised manuscript text omitted]

---

## Referee Report (RR1)

Second review, paper by Fan et al. 2020, The Cryosphere

Responses to the authors' responses to my comments.

Thanks a lot to the authors for the detailed and complete responses to my comments and those of Olivier Castelnau, the other reviewer.
The responses are very detailed, what convinced me to review the paper a second time.

I will not go too much into details again, and I will focus on the main problems that I still see concerning some interpretations. I will provide more details below.

- First, I acknowledge the simplification made by removing the part concerning the WBV study that was, indeed, not utilised enough. This lightens the paper, and makes it clearer.
Nevertheless, my remarks concerning the calculation of a subgrain size remains the same since, whether one uses the WBV analysis results, or misorientation measurements (here with a threshold of 10°) does not modify the fact that the results presented here do not show any subgrain substructure that would enable to identify an "average" subgrain size.
In the situation of figure R1.1 a, when considering the large grains separated by a unique and nearly straight subgrain, what is the subgrain size deduced from your measurement? The same question arrises when considering the grains that are surrounded, at their boundaries by a few small subgrains. What is the subgrain size in this condition? Does it include the part of the inner part of the large grain (that is, indeed, a subgrain)? See figure below, within which we can really question the representativity of a subgrain size...
To bring this to light, maybe you could have plot not only the median, but the quartile, that would very likely have shown a large spread of data.

From figure 4c

[Figure]

Is it SG1?
SG2?

And in this grain, one very small subgrain, and another one nearly the size of the grain? Or is only the small one taken into account?

Since the authors strongly reduced the focus on the subgrain size data, this aspect concerning the metric is less critical, but the problem has not been solved to my point of view.

- Second, concerning the willingness of the authors to use GBS in order to explain the weakening associated to their observations.
Once again, there is absolutely no observation made here, neither the necessity, that render this GBS explanation robust.
Weakening can be explained by GBM, nucleation, subgrain rotation that are very clearly observed.
Texture formation can be very well explained by the same processes, together with dislocation creep, and the results at -30°C are new and very interesting in the sense that they make a clearer link between nucleation of small grains and strong clustered texture, so here again, GBS is not necessary
… so why?

Why is it so necessary to evoke GBS?

Let me mention that most of the articles evoked to justify GBS are coming from the same team as the authors', and also mention GBS as an interpretation, and are not providing any evidence of GBS. These interpretations are basically all built on the only observation of area with small grains.

On top of that, I have a strong concern about the explanations given after my comment 8-2, and based on figure R1.5.

The flow laws that are commonly used to characterized the mechanical response of a material are based on STATIONNARY behaviors, even if very short such as the peak stress for ice (or minimum creep rate in constant load conditions). Therefore a law of the type given by equation R1.1 (or the Glen's law for the minimum creep rate of ice), does hold only during this (pseudo) stationnary state. In the case provided by the curves of figure R1.5, the only place where it should be tested is the peak stress, since no other minimum, maximum or stationary behavior could be reached ( such a stationary state is sometime reached during tertiary creep after 10% strain, as in Treverrow et al. 2012 for instance, but this quasi-stationary behavior results from a balance between a recovery process and a hardening one).

Then, the only way to verify a grain size dependance as written in equation R1.1 is to hold tests at different initial grain sizes, with all other parameters remaining similar, and over a large enough range of grain sizes. During the transient part (here before and after the peak stress), the law to be verified should include a time dependent parameter (such as a Andrade type law for instance). Ignoring this time dependant parameter could lead to a false grain size sensitivity (or any other type of sensitivity).

- One comment about figure R1.6: Wheeler et al. 1999 make it very clear, in the way they evaluate the WBV, that it is not an absolute measurement, and that it only provides an "lower bound", and therefore can only be used as a comparative tool with a lot of care (same type of surfaces observed, very similar deformation history, enough grains for enough statistics, etc...). We have tried to be very careful about that in our papers (Chauve et al. 2017, and Journaux et al. 2019) although our quantitative comparison must still be taken with care, and we made it "relative" (WBV_c / WBV_a). I thank the authors for this clear demonstration of such a necessity to treat this type of EBSD analysis with care.

- About comment 14-1:

I am sorry to read that a link was done between "spontaneous" nucleation and the study by Falus et al. 2011. Spontaneous nucleation, like modelled by Duval et al. 2012 can not give any clue about the nucleus orientation... and Chauve et al. 2017 only evoked it as a way to explain an orientation (of one single nucleus!) that had, apparently, no relation with the parents' orientations. Falus et al. 2011, and applicable also for most nuclei observed by Chauve et al. 2017, mentioned subgrain rotation (rotation or continuous recrystallization) as the main explanation for a weakening of the texture, and orientation spread away from the parent grain orientations, but not totally disconnected from these initial orientations. In section 4.1.4, mentioning nucleation by subgrain rotation (including bulging resulting from strain induced grain boundary migration, as in Chauve et al. 2017b) would be enough to explain the texture weakening, and more in phase with your observations in the small grain networks where small grains keep a strong relation of orientation with parent grains, cf fig 7b and 12 (such as explained by Humphreys and Haterly 2004 for metals, but already mentioned to impact recrystallization texture by Guillopé and Poirier 1979, suggested also for recrystallization along ice core by De La Chapelle et al. 1998, and clearly shown by Falus et al. 2011). Spontaneous nucleation is not expected to produce nucleus with any specific orientation, and GBS could also lead to very different orientations that are not observed here.

By the way, the increasing role of subgrain rotation with decreasing temperature (clearly stated in your conclusion, points 2 and 3), together with the fact that there is more difference between small

grain orientations and large grain orientations at lower temperature is coherent with the dominant role of nucleation by subgrain rotation at lower temperature.

- Another question, that I am not sure to have asked in the previous comments: at strain rates close to 10-5 s-1, with no hydrostatic pressure, ice is weakening by the formation of decohesion or fracture at or close to grain boundaries, in order to accommodate the imposed strain and the strain incompatibilities between grains. The hydrostatic pressure prevent microcracks to open, but then, what could be its impact on recrystallization mechanisms? And could it be that, as in Bourcier et al. 2013, a regime so close to the brittle behavior would, indeed, enhance GBS by the help of microcracks and decohesion?

Comments about the new version of the manuscript:

- Abstract: lines 9-11. Please correct, see previous comment, by replacing the mention to spontaneous nucleation by the mention of nucleation by rotation of subgrains (rotation recrystallization) instead. Spontaneous nucleation model, as existing so far, does not allow to predict any type of orientation relation between nuclei and parent grains.

- Part 3.3.2: I don't understand what is this square mean root diameter... I know Root Mean Square parameter (RMS), that is $square(mean(x^2))$, and that has a statistical meaning, but I don't know the meaning of $mean((square(x))^2)$, or $(mean(square(x)))^2$, it is not clear... Please verify

- Part 3.4.4, line 14: Fig 8-10 instead of 9-11?

- p 12 line 27, "strai,n" → "strain"

- p 15 line 9, only here is the definition of "number density" given while it is used before, please give the definition when first using it.

- p 16 line 2: reference by Placidi et al. 2004 has no reason to be here since it is modeling.

- p16 line 16-17: I don't see why is this study by Eleti et al. mentioned here? I would suggest to keep it for the discussion part.

- p 16 line 20: please also mention the reduced role of GBM when decreasing temperature in impacting the grain size! This is likely the most important one, since grain boundary mobility is strongly reduced...

- Part 4.1.4: please see my comment about the various hypotheses for nucleation mechanisms and impact on texture.
Again here, the two strong hypotheses to mention concerning the weakening of texture in the small grains would be (1) nucleation by subgrain rotation (strengthened by the observation of subgrains whatever the level of strain and the temperature conditions), and (2) GBS in the fine grain necklace (I am not convinced, but let's assume it as a likely mechanism, and here you could mention the ref of Eleti et al.). The first hypothesis has been documented directly and indirectly by several authors (see reference in my previous comment on this subject, but there might exist others). Spontaneous nucleation could be mentioned, but can not be used at the same level since there exist no study showing it systematically, and showing the clear effect on nucleus orientation. Furthermore, if this spontaneous nucleation dominates, then there should be no orientation relation between nucleus and parent grains, while you observe one here.

And please correct the fact that Falus et al. 2011 only mention rotation recrystallization and not at all spontaneous nucleation.

- p 18 line 32: please correct "observations" into "interpretations" since Craw et al 2018 do not show more proof than you of GBS, but use, as you do, GBS to interpret their observations...

- p 19 line 9-11: please be careful with the interpretation of a "hidden grain size sensitivity". I suspect that this is mainly because data from non stationary flow are used to extract parameters from a law including a grain size sensitivity devoted to a stationary state, see my comment about your figure R1.5. Therefore this is not "hidden", this is just not applicable...

- p 19 line 12: replace "spontaneous nucleation" by "nucleation by subgrain rotation".

- p19 line 21: the assertion (1) is incorrect, or not clearly stated... You might mean "softening owing to the reduction of stored strain energy by nucleation and grain boundary migration (or recrystallization processes)" (not "defects", because we don't know which defect you are talking about). This softening has been documented for ages by people studying recrystallization... So please cite Humphreys and Haterly 1996 for a review (or maybe Derby and Ashby 1987), and for ice, maybe Duval 1979, or maybe also Weertman 1983 ?

- p 19 line 27-28: again, the statement is wrong. The balance is between accumulated stored strain energy through dislocation (hardening) and recrystallization mechanisms that reduce this stored energy (both nucleation and GBM, recovery processes)... And again, Montagnat et al. or Sakai et al. are not the one showing that, it had been demonstrated by the whole recrystallization community for a long time!

- p 20 line 9: same mistake again! Not at all the balance between "GBM and nucleation"! Both processes are the softening processes associated with recrystallization... So it is about a balance between softening and hardening processes (that is indeed not balanced here since your experiments still are in the softening part).

- p 21 line 6 "we interpret that the of"... something is missing

- conclusion point 5: reference to "spontaneous" nucleation should be replaced, here, by the mention of nucleation by rotation recrystallization, as detailed in my comments before. Or you could keep it, but as a 3$^{rd}$ and very hypothetic mechanism.

- M. Bourcier, M. Bornert, A. Dimanov, E. Héripré, and J. L. Raphanel. Multiscale experimental investi- gation of crystal plasticity and grain boundary sliding in synthetic halite using digital image correlation. Journal of Geophysical Research: Solid Earth, 118(2):511–526, 2013.

- T. Chauve, M. Montagnat, F. Barou, K. Hidas, A. Tommasi, and D. Mainprice. Investigation of nucle- ation processes during dynamic recrystallization of ice using cryo-ebsd. Philosophical Transactions of the Royal Society A: Mathematical, Physical and Engineering Sciences, 375(2086), 12 2017 b.

- B. Derby and M.F. Ashby. On dynamic recrystallization.Scr. Metall., 21(6):879–884, 1987.

- P. Duval. Creep and recrystallization of polycrystalline ice. Bull. Mineral, 102:80–85, 1979.

- F. J. Humphreys and M. Haterly. Recrystallizationand related annealing phenomena. Pergamon, 1996.

- J. Weertman. Creep deformation of ice. Ann. Rev. Earth Planet. Sci., 11:215–240, 1983.

---

## Author Response (AR2)

**This document is comprised by:**

**Response to Reviewer 1**

We thank Reviewer 1 for her thoughtful and helpful second review of our paper. These insights motivate us to explore more possibilities to explain key observations and to particularly to explain our data and interpretations more clearly. We think that the Reviewer's comments have significantly improved the reasoning and clarity in our paper.

Reviewers comments are in blue type. Highlight: new text in the manuscript highlighted in the same colour.

*Grain Boundary Sliding (GBS)*

The main concern of the reviewer remains - why do we include GBS as an interpretation for the weaker CPOs of small grains? As we explain below, it is our misorientation axis data that push us to maintain inclusion of GBS as a viable, even likely process whereby nucleated grains can change orientation. We added these misorientation data after the first review to address this issue, but clearly had not used or explained these data effectively. We believe we have rectified these issues in the revised manuscript. Below is a detailed explanation of why the misorientation data are key to our interpretation.

We agree with the reviewer that subgrain rotation recrystallization is the most likely nucleation mechanism. We agree that, if the small grains represent nuclei developed through subgrain rotation recrystallization, they would likely have a wider dispersion of orientations than larger ("parent"?) grains. However, the accepted understanding of subgrain rotation recrystallisation is that immediately upon nucleation, a new grain should have an orientation relative to the parent that reflects the dislocation structure of the final subgrain boundary segment at the moment it becomes a grain boundary. This is implicit in the first descriptions of the subgrain rotation recrystallisation process (Guillope and Poirier, 1979; Poirier and Nicolas, 1975), is built into numerical simulations of the process (Gomez-Rivas et al., 2017; Signorelli and Tommasi, 2015) and is stated in all editions of John Humphrey's textbook (Humphreys and Hatherley, 1996; Rollett et al., 2017): "The orientation of the nucleus is present in the deformed structure. There is no evidence that new orientations are formed during or after nucleation, except by twinning" (from section 6.63, 7.63 or 7.62 depending on edition).

The rotation recrystallisation model, at least in part, was erected on the basis of misorientation data, although such data were very hard to collect at the time (Poirier and Guillope, 1979; Poirier and Nicolas, 1975; White, 1973). EBSD has made such data much easier to collect. One of the authors (Prior) was involved in one of the first more extensive EBSD investigations (Bestmann and Prior, 2003) of microstructures that would conventionally be explained recovery, subgrain rotation and subgrain rotation recrystallisation. In this study subgrains and recrystallised grains of approximately the same size are developed in a shear zone within a large crystal (the parent). Low-angle boundaries around subgrains have misorientation axes in rational crystallographic orientations, as would be expected for subgrain rotation. High-angle grain boundaries surrounding recrystallised grains have misorientation axes that are randomly oriented within the crystal, inconsistent with the operation of subgrain rotation recrystallisation alone. Bestmann and Prior (2003) explained these observations by allowing the newly formed nuclei to change orientation by sliding on grain boundaries. Comparable observations have been made in many dynamically recrystallised rocks (see citations of Bestmann and Prior (2003) and the reference list below) and many have been interpreted in the same way.

In our paper, the ice misorientation axes for low- and high-angle boundaries have the same general pattern as that identified by Bestmann and Prior (2003). Low-angle boundaries have

misorientation axes that are oriented tightly in the basal plane, consistent with these boundaries comprising arrays of dislocations that have developed by recovery and subgrain rotation. High-angle boundaries, on the other hand, do not have misorientation axes oriented in the basal plane, nor in any other preferred crystallographic orientation. This relationship still holds when the data are restricted to the high-angle boundaries between central large grains (parents?) and surrounding small grains in the "core and mantle" structures of the lower-temperature experiments. Explaining the misorientation data requires either a different nucleation mechanism (i.e. not rotation recrystallisation) or an additional process to change the orientation of grains during or after nucleation. The common interpretation of comparable data in the rock deformation literature (not just our group) is that GBS changes the orientation of grains after nucleation. For these reasons, we chose to retain GBS as our preferred explanation of the CPO data for nucleated grains.

We continue to emphasise that GBS is just one interpretation of the data, and we present the other possibilities that can explain the same observations. We think it is important to include GBS as a viable explanation of the data so that future generations of scientists can test its validity and importance.

The discussion of the role of grain size-sensitive processes (including GBS) in the evolution of the mechanical data has also been substantially reduced.

**point-by-point reply to comments**

**R1.1:** First, I acknowledge the simplification made by removing the part concerning the WBV study that was, indeed, not utilised enough. This lightens the paper, and makes it clearer.

Nevertheless, my remarks concerning the calculation of a subgrain size remains the same since, whether one uses the WBV analysis results, or misorientation measurements (here with a threshold of 10°) does not modify the fact that the results presented here do not show any subgrain substructure that would enable to identify an "average" subgrain size.

In the situation of figure R1.1 a, when considering the large grains separated by a unique and nearly straight subgrain, what is the subgrain size deduced from your measurement? The same question arises when considering the grains that are surrounded, at their boundaries by a few small subgrains. What is the subgrain size in this condition? Does it include the part of the inner part of the large grain (that is, indeed, a subgrain)? See figure below, within which we can really question the representativity of a subgrain size...

To bring this to light, maybe you could have plot not only the median, but the quartile, that would very likely have shown a large spread of data.

Since the authors strongly reduced the focus on the subgrain size data, this aspect concerning the metric is less critical, but the problem has not been solved to my point of view.

The stereological issues associated with WBV analyses will be presented in a separate paper, which is under preparation for future publication.

We thank the reviewer for pointing out some lack of clarity with respect to the metrics we use in subgrain size statistics. In our study, the subgrain size metrics consider all the inner parts of grains segmented by subgrain boundaries, including both large and small subgrains. Grains entirely surrounded by high-angle boundaries and without an internal subgrain boundary are also included, as the high-angle boundaries are above the chosen threshold for subgrain boundary misorientation. A figure from Trimby et al (1998) is useful here (Fig. R1.1). The upper picture shows boundaries with ≥4° misorientation, and the lower shows boundaries with ≥10° misorientation for the same field of view. If the subgrain boundary misorientation threshold is set at ≥4° and the grain boundary misorientation threshold is set at ≥10° then all the enclosed areas in the upper figure are counted as subgrains and all the enclosed areas in the

lower figure as grains. We have added a statement to section 2.5.1 to clarify this. This approach is a standard approach in geology and in metallurgy (e.g. (Adams et al., 1993; Coutinho et al., 2017; Humphreys, 2001, 2004; Sintay et al., 2009) although the values of the misorientation thresholds vary.

[Figure]

**Fig. R1.1** Boundary misorientation maps for the low strain zone from quartz mylonite shear zone from Torridon, NW Scotland (after Fig. 5 from Trimby et al., 1998). (a) Boundaries with ≥ 4° misorientation. (b) Boundaries with ≥ 10° misorientation.

We also thank the reviewer for the suggestion of using interquartile measures to better describe the skewed subgrain size distributions. In the modified manuscript, we have added a lower quartile and higher quartile of subgrain size in Table 3 and the interquartile range (IQR), with upper and lower bounds constrained by higher and lower quartile subgrain size, respectively, that have been plotted along with mean and median subgrain size in Fig. 11(b). The related description of our approach is given in section 3.3.3.

**R1.2** - Second, concerning the willingness of the authors to use GBS in order to explain the weakening associated to their observations.

Once again, there is absolutely no observation made here, neither the necessity, that render this GBS explanation robust.

Weakening can be explained by GBM, nucleation, subgrain rotation that are very clearly observed. Texture formation can be very well explained by the same processes, together with dislocation creep, and the results at -30°C are new and very interesting in the sense that they make a clearer link between nucleation of small grains and strong clustered texture, so here again, GBS is not necessary

… so why?

Why is it so necessary to evoke GBS?

Our reasoning for including GBS is explained at the beginning of this response to the reviewer (see above). We will add here that there has always been an issue with the microstructural identification of GBS. The microstructures that can be used to interpret the operation of the other processes, such as GBM (irregular-lobate boundaries), nucleation (in an experiment or a strain series in nature, determined by the increase in the number of grains), subgrain rotation

(low-angle boundaries, increasing misorientation angles with strain, rational misorientation axes) are clear and generally easy to identify. The signature of GBS is more subtle: if a grain rotates by sliding on its boundary, how does one recognize that in a final microstructure? To understand the difficulty of identifying GBS in experiments, it is worth looking at the beautiful syn-microscopic experiments on octachloropropylene published by Jin Han Ree (Ree, 1994), described below.

Fig R1.2 shows part of Ree's Figure 10; a time series of drawings of the microstructure. Ree was able to use marker particle lines to demonstrate translational movement on grain boundaries (GBS). In the absence of the (fiducial) marker lines, or some other strain marker, it would be very difficult to identify GBS, even in the case of a time-series experiment. The single time step microstructures show good evidence of GBM (lobate boundaries) and subgrain rotation (low angle boundaries) but finding evidence for GBS in a single microstructural image is difficult.

[Figure]

**Figure R1.2** Part of Figure 10 from Ree (1994), showing a time series of drawings of the microstructure of deforming octachloropropane.

This is why the misorientation axis criteria outlined earlier in this response are so important. The change from crystallographically controlled low-angle boundaries to high-angle boundaries with little crystallographic control requires explanation and could be a GBS signal. In the case of our experiments, this interpretation works bests for the low-temperature experiments, but in the -10 °C experiments it is less clear, as the "core and mantle" structures are absent. It is likely that GBM will modify the grain boundary topology, particularly in those warmer experiments. Ree's experiments shown above and other time series experiments (Drury and Humphreys, 1988) show that GBM and GBS can (and indeed must, Ashby, 1972; Sundberg and Cooper, 2008) operate in parallel and that GBS would be cryptic in a final microstructure.

We have restructured the discussion to address the reviewer's concerns. The misorientation data are now discussed more thoroughly in section 4.1.2.2. (Dislocation activity, recovery, subgrain rotation and subgrain rotation recrystallisation). In this section we highlight the difference of misorientation axes of low- and high-angle boundaries and the implication that subgrain rotation recrystallisation alone cannot explain these data. GBS is not mentioned in this section. In section 4.1.4 (CPO development: differences related to grain size) we discuss the greater orientation spread of the small grains. We note that subgrain rotation recrystallisation can give a bigger orientation spread, but the lack of crystallographic control across high-angle boundaries, particularly the small-large grains in core and mantle structures (in the cold experiments), means that it is difficult to explain the weakening of the CPO by subgrain rotation recrystallization alone. We then discuss briefly two ways to explain these

data: a different nucleation process (nucleation with random orientations) and GBS following nucleation. We state (in section 4.1.4) that data to infer GBS in the higher-temperature experiments are less clear but we still allow that it is possible, given the cryptic nature of microstructures under conditions where enhanced GBM and GBS coexist, as discussed above.

We emphasize that the interaction of GBS with dislocation processes and GBM is well-recognized in the ice literature. Duval (1985) states "*Grain boundary migration associated with grain growth appears to be an efficient accommodation process for grain boundary sliding and dislocation glide.*" The literature review by Faria and others (2014) states: "*Hondoh and Higashi (1983) and Liu et al. (1993, 1995) used X-ray topography to study the interactions between dislocations and grain boundaries in ice bicrystals and polycrystalline ice, respectively. They could demonstrate that the regions surrounding grain boundaries (viz. the "mantle" of the grain, after Gifkins, 1976) generally deform before the grain interiors (viz. the "core" of the grain). Dislocations are emitted from stress concentrations at grain boundaries, caused by strain misfits and/or grain boundary sliding, and this process completely overwhelms any lattice dislocation generation mechanism.*" and "*Above -10 °C, the increase of the minimum strain rate with temperature is enhanced and the Arrhenius law breaks down (Glen, 1955, 1975; Hooke, 1981; Budd and Jacka, 1989). It is believed that grain boundary sliding and the presence of water within the grain boundaries may be the main causes of this creep enhancement (Barnes et al., 1971)*".

**R1.3:** Let me mention that most of the articles evoked to justify GBS are coming from the same team as the authors', and also mention GBS as an interpretation, and are not providing any evidence of GBS. These interpretations are basically all built on the only observation of area with small grains.

To address the reviewer's comment that most of the interpretations of GBS are from the authors' group, we provide two lists of published papers on ice and rock deformation in which GBS is a key interpretation of mechanical data and/or microstructural observations or is integral to the description of deformation. The lists are appended to the end of this document and are separate from the main reference list for this document.

GBS is accepted as a process that can occur in rock deformation. The data we use to infer GBS from ice microstructures is very similar to data used to infer GBS from rock microstructures, by us and by many other authors. *List 1* (page 11-13) is an incomplete list of papers that have inferred GBS in rocks in the last 5 years: generated from a quick Web of Science search, with a manual check that the paper really does infer GBS. We have not included papers from our broad group although authors who Prior and Goldsby have published with are highlighted (neither Prior or Goldsby had any role in these publications).

GBS in the ice microstructure literature extends to authors well beyond our broad research group. We are responsible for approximately a quarter of the papers that infer GBS in ice (see *List 2* (page 13-14)).

**R1.4:** On top of that, I have a strong concern about the explanations given after my comment 8-2, and based on figure R1.5.

The flow laws that are commonly used to characterized the mechanical response of a material are based on STATIONNARY behaviors, even if very short such as the peak stress for ice (or minimum creep rate in constant load conditions). Therefore a law of the type given by equation R1.1 (or the Glen's law for the minimum creep rate of ice), does hold only during this (pseudo) stationnary state. In the case provided by the curves of figure R1.5, the only place where it should be tested is the peak stress, since no other minimum, maximum or stationary behavior could be reached ( such a stationary state is sometime reached during tertiary creep after 10% strain, as in Treverrow et al. 2012 for instance, but this quasi-stationary behavior results from a balance between a recovery process and a hardening one).

Then, the only way to verify a grain size dependance as written in equation R1.1 is to hold tests at different initial grain sizes, with all other parameters remaining similar, and over a large enough range of grain sizes. During the transient part (here before and after the peak stress), the law to be verified should include a time dependent parameter (such as a Andrade type law for instance). Ignoring this time dependant parameter could lead to a false grain size sensitivity (or any other type of sensitivity).

We thank the reviewer for their comment. A time-dependent parameter should be considered during the modelling of ice mechanical behaviour. This is a very useful suggestion, as we are modelling ice mechanical behaviour based on grain size and CPO data in another paper. We realise there are balances of mechanical hardening and softening mechanisms as the strain increases during ice deformation. Modelling the ice mechanical behaviour is beyond the scope of this paper, and we have reduced the discussion of it significantly in this paper. We plan to present a much more complete analysis of our data and literature data in a separate paper.

**R1.5:** One comment about figure R1.6: Wheeler et al. 1999 make it very clear, in the way they evaluate the WBV, that it is not an absolute measurement, and that it only provides an "lower bound", and therefore can only be used as a comparative tool with a lot of care (same type of surfaces observed, very similar deformation history, enough grains for enough statistics, etc...). We have tried to be very careful about that in our papers (Chauve et al. 2017, and Journaux et al. 2019) although our quantitative comparison must still be taken with care, and we made it "relative" (WBV_c / WBV_a). I thank the authors for this clear demonstration of such a necessity to treat this type of EBSD analysis with care.

We thank the reviewer very much for appreciating our work related with stereological issues of WBV. The stereological issues associated with WBV analyses will be presented in a separate paper, which is under preparation for future publication.

**R1.6:** About comment 14-1: I am sorry to read that a link was done between "spontaneous" nucleation and the study by Falus et al. 2011. Spontaneous nucleation, like modelled by Duval et al. 2012 can not give any clue about the nucleus orientation... and Chauve et al. 2017 only evoked it as a way to explain an orientation (of one single nucleus!) that had, apparently, no relation with the parents' orientations. Falus et al. 2011, and applicable also for most nuclei observed by Chauve et al. 2017, mentioned subgrain rotation (rotation or continuous recrystallization) as the main explanation for a weakening of the texture, and orientation spread away from the parent grain orientations, but not totally disconnected from these initial orientations. In section 4.1.4, mentioning nucleation by subgrain rotation (including bulging resulting from strain induced grain boundary migration, as in Chauve et al. 2017b) would be enough to explain the texture weakening, and more in phase with your observations in the small grain networks where small grains keep a strong relation of orientation with parent grains, cf fig 7b and 12 (such as explained by Humphreys and Haterly 2004 for metals, but already mentioned to impact recrystallization texture by Guillop. and Poirier 1979, suggested also for recrystallization along ice core by De La Chapelle et al. 1998, and clearly shown by Falus et al. 2011). Spontaneous nucleation is not expected to produce nucleus with any specific orientation, and GBS could also lead to very different orientations that are not observed here. By the way, the increasing role of subgrain rotation with decreasing temperature (clearly stated in your conclusion, points 2 and 3), together with the fact that there is more difference between small grain orientations and large grain orientations at lower temperature is coherent with the dominant role of nucleation by subgrain rotation at lower temperature.

We apologize for making an improper connection between "spontaneous" nucleation and the study by Falus and others (2011). Falus and others (2011) suggest that subgrain rotation recrystallization has a contribution to CPO weakening but cannot explain the most extreme dispersions in the recrystallised grain orientations as these exceed what is even possible by subgrain rotation (p1532). Their interpretation is that "*The higher dispersion of the olivine CPO in the mylonites may be attributed to a more important contribution of bulging processes to nucleation, which allows for higher misorientations between parent and recrystallized grains.*"(p1539). We cannot compare directly with the Falus et al. data set as they only show misorientation axis data for low-angle boundaries; instead their interpretations are informed by dispersion data for individual crystal directions. They do not rule out GBS: "*Finally, although the studied samples present no clear evidence for grain boundary sliding, this process might also contribute to a higher dispersion of the CPO in the finest-grained samples.*" (p1539).

Extracted from the longer comment above: nucleation by subgrain rotation (refs) would be enough to explain the texture weakening, more in phase with your observations in the small grain networks where small grains keep a strong relation of orientation with parent grains, cf fig 7b and 12. This response, in part, repeats what we have already said, but we would like to address this sub-comment directly. The coloured dispersed orientations in fig 7b represent the larger grains (ref 1-5) in 7a and their dispersion pattern (~

great circles) and the misorientation axes of low-angle boundaries (7c) fits a recovery and subgrain rotation model as shown schematically by the grey dashed line representing subgrain boundaries in fig 12 (labelled SGR-subgrain rotation, but not recrystallisation). We think we are in agreement with the reviewer in this part of the interpretation. The problem comes in linking neighboring small grains and large grains. The left-most figure of 7d (ref 1), for example, shows all of the misorientation axes between pixels on either side of the grain boundary enclosing grain ref1 (i.e., all misorientations linking pixels along the boundary of grain ref1 to neighbouring pixels in small grains). These are high-angle boundaries so the angular error on the misorientation axis is low. Each cluster in this fig relates to misorientations to a single grain and the variance in each cluster relates to the internal distortion of ref1 and the small grain in question. The misorientations for ref1 (or any of the other ref grains) do not lie in the basal plane and have no preferred nor rational crystallographic orientation. Subgrain rotation recrystallisation can be considered a nucleation mechanism for the small grains, but alone it cannot explain the misorientation axes of the high-angle boundaries. Either a different nucleation mechanism is needed, or an additional process is needed to change the orientation of grains during or after nucleation.

We thank the reviewer for clarifying their view of nucleus orientations associated with spontaneous nucleation. In our discussion we now refer to nucleation of grains with random orientations rather than spontaneous nucleation. There is some precedent for this idea in publications on metals deformed at high homologous temperatures that interpret small recrystallised grains as nuclei with random orientations (Hasegawa and Fukutomi, 2002; Hasegawa et al., 2003). The proposed mechanism for the randomly oriented nuclei is that they are produced at the tips of irregular boundaries of "parent" grains, without introducing GBS (Hasegawa and Fukutomi, 2002; Hasegawa et al., 2003).

**R1.7:** Another question, that I am not sure to have asked in the previous comments: at strain rates close to 10-5 s-1, with no hydrostatic pressure, ice is weakening by the formation of decohesion or fracture at or close to grain boundaries, in order to accommodate the imposed strain and the strain incompatibilities between grains. The hydrostatic pressure prevent microcracks to open, but then, what could be its impact on recrystallization mechanisms? And could it be that, as in Bourcier et al. 2013, a regime so close to the brittle behavior would, indeed, enhance GBS by the help of microcracks and decohesion?

The chief purpose of the confining pressure in these experiments is to suppress brittle phenomena including cracking and frictional sliding. Figure R1.3 shows the experiment with the highest differential stress, plotted on a Mohr diagram for stress. The green circle shows the shear and normal stresses for surfaces of all orientations and the maximum ($\sigma_1$) and minimum ($\sigma_3 = \sigma_2$ = confining pressure) plot along the line of zero shear stress. Superposed are two failure envelopes. One is a Coulomb (frictional sliding) envelope using the friction coefficient for ice-ice sliding from McCarthy et al. (2017). Coulomb envelopes usually underestimate brittle strength. The second failure envelope is the composite envelope from Beeman et al. (1988). Red and blue Mohr circles show the stress states needed for brittle failure at a confining pressure of 20 MPa with each of these envelopes. Maximum differential stresses applied in our experiments are substantially below those for needed for frictional sliding or brittle failure. Ice deformed in the study were far away from the brittle or frictional sliding regime.

[Figure]

**Figure R1.3**. Mohr diagram showing stress state of sample PIL164 (the largest differential stress) in green. A coulomb failure envelope using a friction coefficient of 0.29 from (McCarthy et al., 2017) is shown with a red dashed line and the Mohr circle for failure at 20MPa confining pressure is shown in red. The blue lines show the (Beeman et al., 1988) failure envelope from and the Mohr circle for failure at 20MPa confining pressure is shown in blue.

The effect of pressure in affecting recrystallisation processes of ice is not fully investigated. We have very similar results from microstructural and mechanical from -10 °C experiments at 10MPa and 20MPa and warm temperature experiments at -5 °C have microstructures very similar to the many published unconfined experiments at this temperature, so at present we do not think the pressure has a significant effect on recrystallisation. In the Bourcier and others (2013) paper the bulk tests image unconfined free surfaces and the SEM tests image free surfaces under high vacuum. These are great experiments and clearly cracking and GBS are closely linked processes, but in the unconfined state at the strain rates they employ cracking is expected. The confining pressure in our experiments puts our experiments further away from the brittle field than a typical unconfined ice experiment. The Mohr analysis outlined above estimates that the shear stresses developed are at least ~3 MPa below what would be needed for shear failure and no tensile stresses will be developed in the sample.

**R1.8:** Abstract: lines 9-11. Please correct, see previous comment, by replacing the mention to spontaneous nucleation by the mention of nucleation by rotation of subgrains (rotation recrystallization) instead. Spontaneous nucleation model, as existing so far, does not allow to predict any type of orientation relation between nuclei and parent grains.

We have replaced references to "spontaneous nucleation" with "nucleation of grains in random orientation". As discussed in previous responses nucleation by subgrain rotation recrystallisation alone is not consistent with high-angle boundary misorientation axis data.

**R1.9:** Part 3.3.2: I don't understand what is this square mean root diameter... I know Root Mean Square parameter (RMS), that is square(mean(x^2)), and that has a statistical meaning, but I don't know the meaning of mean((square(x))^2), or (mean(square(x)))^2, it is not clear... Please verify

Square mean root diameter ($D_{SMR} = (\overline{\sqrt{D}})^2$), with the equation shown in section 3.3.2, is different from root mean square. Root mean square will exaggerate the difference between finer and coarser grain sizes, because each grain size will be squared before averaging. On the contrary, square mean root will decrease the difference between finer and coarser grain sizes, because the root of each grain size will be before averaging. The square mean root diameter is often useful for dynamically recrystallised grain sizes, as a frequency (probability) plot of the square root of grain size (or the log of grain size) often approximates a normal distribution.

**R1.10:** Part 3.4.4, line 14: Fig 8-10 instead of 9-11?

Corrected.

**R1.11:** p 12 line 27, "strai,n" → "strain"

Corrected.

**R1.12:** p 15 line 9, only here is the definition of "number density" given while it is used before, please give the definition when first using it.

We thank the reviewer for catching this unclear statement. We have clarified terminology in section 4.1.2.1 and elsewhere refer to the number of grains per unit area.

**R1.13:** p 16 line 2: reference by Placidi et al. 2004 has no reason to be here since it is modeling.

We have removed the reference of Placidi et al., 2004.

**R1.14:** p16 line 16-17: I don't see why is this study by Eleti et al. mentioned here? I would suggest to keep it for the discussion part.

We have removed the reference of Eleti et al., 2020 from section 4.1.2.2. This reference is kept in the discussion part.

**R1.15:** p 16 line 20: please also mention the reduced role of GBM when decreasing temperature in impacting the grain size! This is likely the most important one, since grain boundary mobility is strongly reduced...

The rate of GBM depends on two parameters - the boundary mobility, which is a function of temperature, and the driving force, conventionally defined by the dislocation density difference (Humphreys and Hatherley, 1996). The dislocation density difference is likely to be controlled by stress (Bailey and Hirsch, 1960; Ajaja, 1991). At a lower temperature, the grain boundary mobility is likely to reduce as suggested by the reviewer. However, the differential stress for a given strain rate increases (Fig. 3). Consequently, the dislocation density difference is likely to be higher at a lower temperature. We have adjusted the text (in section 4.1.2.3) to incorporate this line of discussion.

**R1.16:** Part 4.1.4: please see my comment about the various hypotheses for nucleation mechanisms and impact on texture.

Again here, the two strong hypotheses to mention concerning the weakening of texture in the small grains would be (1) nucleation by subgrain rotation (strengthened by the observation of subgrains whatever the level of strain and the temperature conditions), and (2) GBS in the fine grain necklace (I am not convinced, but let's assume it as a likely mechanism, and here you could mention the ref of Eleti et al.). The first hypothesis has been documented directly and indirectly by several authors (see reference in my previous comment on this subject, but there might exist others). Spontaneous nucleation could be mentioned, but can not be used at the same level since there exist no study showing it systematically, and showing the clear effect on nucleus orientation. Furthermore, if this spontaneous nucleation dominates, then there should be no orientation relation between nucleus and parent grains, while you observe one here. And please correct the fact that Falus et al. 2011 only mention rotation recrystallization and not at all spontaneous nucleation.

Please refer to previous replies

**R1.17:** p 18 line 32: please correct "observations" into "interpretations" since Craw et al 2018 do not show more proof than you of GBS, but use, as you do, GBS to interpret their observations...

Corrected.

**R1.18:** p 19 line 9-11: please be careful with the interpretation of a "hidden grain size sensitivity". I suspect that this is mainly because data from non stationary flow are used to extract parameters from a law including a grain size sensitivity devoted to a stationary state, see my comment about your figure R1.5. Therefore this is not "hidden", this is just not applicable...

We have removed such statement in the modified manuscript. Modelling work on ice mechanical behaviour will be presented in a separate paper.

**R1.19:** p 19 line 12: replace "spontaneous nucleation" by "nucleation by subgrain rotation".

We have replaced references to "spontaneous nucleation" with "nucleation of grains in random orientation". As discussed in previous responses, nucleation by subgrain rotation recrystallisation alone is not consistent with high-angle boundary misorientation axis data.

**R1.20** - p19 line 21: the assertion (1) is incorrect, or not clearly stated... You might mean "softening owing to the reduction of stored strain energy by nucleation and grain boundary migration (or recrystallization processes)" (not "defects", because we don't know which defect you are talking about). This softening has been documented for ages by people studying recrystallization... So please cite Humphreys and Haterly 1996 for a review (or maybe Derby and Ashby 1987), and for ice, maybe Duval 1979, or maybe also Weertman 1983 ?

We have modified the statement (in section 4.2) in accordance with the reviewer's suggestion.

**R1.21:** p 19 line 27-28: again, the statement is wrong. The balance is between accumulated stored strain energy through dislocation (hardening) and recrystallization mechanisms that reduce this stored energy (both nucleation and GBM, recovery processes)... And again, Montagnat et al. or Sakai et al. are not the one showing that, it had been demonstrated by the whole recrystallization community for a long time!

We have modified this discussion (in section 4.2) in accordance with the reviewer's suggestion and have cited some of the older references.

**R1.22:** p 20 line 9: same mistake again! Not at all the balance between "GBM and nucleation"! Both processes are the softening processes associated with recrystallization... So it is about a balance between softening and hardening processes (that is indeed not balanced here since your experiments still are in the softening part).

We have modified the statement in (section 4.2) in accordance with the reviewer's comment

**R1.23:** p 21 line 6 "we interpret that the of"... something is missing

Oops: We interpret that the open c-axis cone develops because strain-induced GBM favours the growth of grains in easy slip orientations. Text corrected in section 5.

**R1.24:** conclusion point 5: reference to "spontaneous" nucleation should be replaced, here, by the mention of nucleation by rotation recrystallization, as detailed in my comments before. Or you could keep it, but as a 3rd and very hypothetic mechanism.

We have replaced references from "spontaneous nucleation" with "nucleation of grains in random orientation". Again, as discussed in previous responses nucleation by subgrain rotation recrystallisation alone is not consistent with high-angle boundary misorientation axis data.

**List 1: Papers on rock deformation that infer GBS *(last 5 years only.)**

References that make interpretations of grain boundary sliding in rocks (natural and experimental) or use GBS as part of microstructural modelling framework not from the Otago/Liverpool or UPenn research groups. People we have worked with (and published with) highlighted in yellow. Last 5 years only.

Bollinger, C., Marquardt, K., and Ferreira, F., 2019, Intragranular plasticity vs. grain boundary sliding (GBS) in forsterite: Microstructural evidence at high pressures (3.5-5.0 GPa): American Mineralogist, v. 104, no. 2, p. 220-231.

Cao, Y., Jung, H., and Song, S. G., 2017, Olivine fabrics and tectonic evolution of fore-arc mantles: A natural perspective from the Songshugou dunite and harzburgite in the Qinling orogenic belt, central China: Geochemistry Geophysics Geosystems, v. 18, no. 3, p. 907-934.

Ceccato, A., Menegon, L., Pennacchioni, G., and Morales, L. F. G., 2018, Myrmekite and strain weakening in granitoid mylonites: Solid Earth, v. 9, no. 6, p. 1399-1419.

Chatzaras, V., Kruckenberg, S. C., Cohen, S. M., Medaris, L. G., Withers, A. C., and Bagley, B., 2016, Axial-type olivine crystallographic preferred orientations: The effect of strain geometry on mantle texture: Journal of Geophysical Research-Solid Earth, v. 121, no. 7, p. 4895-4922.

Cobden, L., Trampert, J., and Fichtner, A., 2018, Insights on Upper Mantle Melting, Rheology, and Anelastic Behavior From Seismic Shear Wave Tomography: Geochemistry Geophysics Geosystems, v. 19, no. 10, p. 3892-3916.

Czaplinska, D., Piazolo, S., and Zibra, I., 2015, The influence of phase and grain size distribution on the dynamics of strain localization in polymineralic rocks: Journal of Structural Geology, v. 72, p. 15-32.

De Paola, N., Holdsworth, R. E., Viti, C., Collettini, C., and Bullock, R., 2015, Can grain size sensitive flow lubricate faults during the initial stages of earthquake propagation?: Earth and Planetary Science Letters, v. 431, p. 48-58.

Faul, U., and Jackson, I., 2015, Transient Creep and Strain Energy Dissipation: An Experimental Perspective, *in* Jeanloz, R., and Freeman, K. H., eds., Annual Review of Earth and Planetary Sciences, Vol 43, Volume 43, p. 541-569.

Finch, M. A., Weinberg, R. F., and Hunter, N. J. R., 2016, Water loss and the origin of thick ultramylonites: Geology, v. 44, no. 8, p. 599-602.

Gardner, R., Piazolo, S., Evans, L., and Daczko, N., 2017, Patterns of strain localization in heterogeneous, polycrystalline rocks-a numerical perspective: Earth and Planetary Science Letters, v. 463, p. 253-265.

Gardner, R. L., Piazolo, S., Daczko, N. R., and Trimby, P., 2020, Microstructures reveal multistage melt present strain localisation in mid-ocean gabbros: Lithos, v. 366.

Gasc, J., Demouchy, S., Barou, F., Koizumi, S., and Cordier, P., 2019, Creep mechanisms in the lithospheric mantle inferred from deformation of iron-free forsterite aggregates at 900-1200 degrees C: Tectonophysics, v. 761, p. 16-30.

Gilgannon, J., Fusseis, F., Menegon, L., Regenauer-Lieb, K., and Buckman, J., 2017, Hierarchical creep cavity formation in an ultramylonite and implications for phase mixing: Solid Earth, v. 8, no. 6, p. 1193-1209.

Giuntoli, F., Brovarone, A. V., and Menegon, L., 2020, Feedback between high-pressure genesis of abiotic methane and strain localization in subducted carbonate rocks: Scientific Reports, v. 10, no. 1.

Goncalves, P., Poilvet, J. C., Oliot, E., Trap, P., and Marquer, D., 2016, How does shear zone nucleate? An example from the Suretta nappe (Swiss Eastern Alps): Journal of Structural Geology, v. 86, p. 166-180.

Graziani, R., Larson, K. P., and Soret, M., 2020, The effect of hydrous mineral content on competitive strain localization mechanisms in felsic granulites: Journal of Structural Geology, v. 134.

Hansen, L. N., Conrad, C. P., Boneh, Y., Skemer, P., Warren, J. M., and Kohlstedt, D. L., 2016a, Viscous anisotropy of textured olivine aggregates: 2. Micromechanical model: Journal of Geophysical Research-Solid Earth, v. 121, no. 10, p. 7137-7160.

Hansen, L. N., Warren, J. M., Zimmerman, M. E., and Kohlstedt, D. L., 2016b, Viscous anisotropy of textured olivine aggregates, Part 1: Measurement of the magnitude and evolution of anisotropy: Earth and Planetary Science Letters, v. 445, p. 92-103.

Karato, S. I., Olugboji, T., and Park, J., 2015, Mechanisms and geologic significance of the mid-lithosphere discontinuity in the continents: Nature Geoscience, v. 8, no. 7, p. 509-514.

Keppler, R., Stipp, M., Behrmann, J. H., Ullemeyer, K., and Heidelbach, F., 2016, Deformation inside a paleosubduction channel - Insights from microstructures and crystallographic preferred orientations of eclogites and metasediments from the Tauern Window, Austria: Journal of Structural Geology, v. 82, p. 60-79.

Linckens, J., Zulauf, G., and Hammer, J., 2016, Experimental deformation of coarse-grained rock salt to high strain: Journal of Geophysical Research-Solid Earth, v. 121, no. 8, p. 6150-6171.

Liu, S. R., Zhang, J. J., Qi, G. W., and Wang, M., 2016, Ductile deformation and its geological implications for retrograded eclogites from the Hongqiyingzi Complex in Chicheng, northern Hebei, China: Science China-Earth Sciences, v. 59, no. 8, p. 1610-1621.

Lopez-Sanchez, M. A., and Llana-Funez, S., 2018, A cavitation-seal mechanism for ultramylonite formation in quartzofeldspathic rocks within the semi-brittle field (Vivero fault, NW Spain): Tectonophysics, v. 745, p. 132-153.

Maierova, P., Lexa, O., Jerabek, P., Schulmann, K., and Franek, J., 2017, Computational study of deformation mechanisms and grain size evolution in granulites - Implications for the rheology of the lower crust: Earth and Planetary Science Letters, v. 466, p. 91-102.

Mansard, N., Raimbourg, H., Augier, R., Precigout, J., and Le Breton, N., 2018, Large-scale strain localization induced by phase nucleation in mid-crustal granitoids of the south Armorican massif: Tectonophysics, v. 745, p. 46-65.

Marti, S., Stunitz, H., Heilbronner, R., Plumper, O., and Drury, M., 2017, Experimental investigation of the brittle-viscous transition in mafic rocks - Interplay between fracturing, reaction, and viscous deformation: Journal of Structural Geology, v. 105, p. 62-79.

Marti, S., Stunitz, H., Heilbronner, R., Plumper, O., and Kilian, R., 2018, Syn-kinematic hydration reactions, grain size reduction, and dissolution-precipitation creep in experimentally deformed plagioclase-pyroxene mixtures: Solid Earth, v. 9, no. 4, p. 985-1009.

Maruyama, G., and Hiraga, T., 2017a, Grain- to multiple-grain-scale deformation processes during diffusion creep of forsterite plus diopside aggregate: 1. Direct observations: Journal of Geophysical Research-Solid Earth, v. 122, no. 8, p. 5890-5915.

Maruyama, G., and Hiraga, T, 2017b, Grain- to multiple-grain-scale deformation processes during diffusion creep of forsterite plus diopside aggregate: 2. Grain boundary sliding-induced grain rotation and its role in crystallographic preferred orientation in rocks: Journal of Geophysical Research-Solid Earth, v. 122, no. 8, p. 5916-5934.

Menegon, L., Fusseis, F., Stunitz, H., and Xiao, X. H., 2015, Creep cavitation bands control porosity and fluid flow in lower crustal shear zones: Geology, v. 43, no. 3, p. 227-230.

Miranda, E. A., Hirth, G., and John, B. E., 2016, Microstructural evidence for the transition from dislocation creep to dislocation-accommodated grain boundary sliding in naturally deformed plagioclase: Journal of Structural Geology, v. 92, p. 30-45.

Ohuchi, T., Kawazoe, T., Higo, Y., Funakoshi, K., Suzuki, A., Kikegawa, T., and Irifune, T., 2015, Dislocation-accommodated grain boundary sliding as the major deformation mechanism of olivine in the Earth's upper mantle: Science Advances, v. 1, no. 9.

Okudaira, T., Jerabek, P., Stunitz, H., and Fusseis, F., 2015, High-temperature fracturing and subsequent grain-size-sensitive creep in lower crustal gabbros: Evidence for coseismic loading followed by creep during decaying stress in the lower crust?: Journal of Geophysical Research-Solid Earth, v. 120, no. 5, p. 3119-3141.

Papa, S., Pennacchioni, G., Menegon, L., and Thielmann, M., 2020, High-stress creep preceding coseismic rupturing in amphibolite-facies ultramylonites: Earth and Planetary Science Letters, v. 541.

Papeschi, S., Musumeci, G., Massonne, H. J., Bartoli, O., and Cesare, B., 2019, Partial melting and strain localization in metapelites at very low-pressure conditions: The northern Apennines magmatic arc on the Island of Elba, Italy: Lithos, v. 350.

Park, M., and Jung, H., 2017, Microstructural evolution of the Yugu peridotites in the Gyeonggi Massif, Korea: Implications for olivine fabric transition in mantle shear zones: Tectonophysics, v. 709, p. 55-68.

Platt, J. P., 2015, Rheology of two-phase systems: A microphysical and observational approach: Journal of Structural Geology, v. 77, p. 213-227.

Precigout, J., and Stunitz, H., 2016, Evidence of phase nucleation during olivine diffusion creep: A new perspective for mantle strain localisation: Earth and Planetary Science Letters, v. 455, p. 94-105.

Quintanilla-Terminel, A., Zimmerman, M. E., Evans, B., and Kohlstedt, D. L., 2017, Microscale and nanoscale strain mapping techniques applied to creep of rocks: Solid Earth, v. 8, no. 4, p. 751-765.

Rogowitz, A., White, J. C., and Grasemann, B., 2016, Strain localization in ultramylonitic marbles by simultaneous activation of dislocation motion and grain boundary sliding (Syros, Greece): Solid Earth, v. 7, no. 2, p. 355-366.

Sorensen, B. E., Grant, T., Ryan, E. J., and Larsen, R. B., 2019, In situ evidence of earthquakes near the crust mantle boundary initiated by mantle CO2 fluxing and reaction-driven strain softening: Earth and Planetary Science Letters, v. 524.

Soret, M., Agard, P., Ildefonse, B., Dubacq, B., Prigent, C., and Rosenberg, C., 2019, Deformation mechanisms in mafic amphibolites and granulites: record from the Semail metamorphic sole during subduction infancy: Solid Earth, v. 10, no. 5, p. 1733-1755.

Spruzeniece, L., and Piazolo, S., 2015, Strain localization in brittle-ductile shear zones: fluid-abundant vs. fluid-limited conditions (an example from Wyangala area, Australia): Solid Earth, v. 6, no. 3, p. 881-901.

Stewart, C. A., and Miranda, E. A., 2017, The Rheological Evolution of Brittle-Ductile Transition Rocks During the Earthquake Cycle: Evidence for a Ductile Precursor to Pseudotachylyte in an Extensional Fault System, South Mountains, Arizona: Journal of Geophysical Research-Solid Earth, v. 122, no. 12, p. 10643-10665.

Sullivan, W. A., and Monz, M. E., 2016, Rheologic evolution of low-grade metasedimentary rocks and granite across a large strike-slip fault zone: A case study of the Kellyland fault zone, Maine, USA: Journal of Structural Geology, v. 86, p. 13-31.

Tasaka, M., Zimmerman, M. E., and Kohlstedt, D. L., 2017, Rheological Weakening of Olivine plus Orthopyroxene Aggregates Due to Phase Mixing: 1. Mechanical Behavior: Journal of Geophysical Research-Solid Earth, v. 122, no. 10, p. 7584-7596.

Thielmann, M., 2018, Grain size assisted thermal runaway as a nucleation mechanism for continental mantle earthquakes: Impact of complex rheologies: Tectonophysics, v. 746, p. 611-623.

Verberne, B. A., Chen, J. Y., Niemeijer, A. R., de Bresser, J. H. P., Pennock, G. M., Drury, M. R., and Spiers, C. J., 2017, Microscale cavitation as a mechanism for nucleating earthquakes at the base of the seismogenic zone: Nature Communications, v. 8.

Wiesman, H. S., Zimmerman, M. E., and Kohlstedt, D. L., 2018, Laboratory investigation of mechanisms for phase mixing in olivine plus ferropericlase aggregates: Philosophical Transactions of the Royal Society a-Mathematical Physical and Engineering Sciences, v. 376, no. 2132.

Yao, Z. S., Qin, K. Z., Wang, Q., and Xue, S. C., 2019, Weak B-Type Olivine Fabric Induced by Fast Compaction of Crystal Mush in a Crustal Magma Reservoir: Journal of Geophysical Research-Solid Earth, v. 124, no. 4, p. 3530-3556.

Yao, Z. S., Qin, K. Z., and Xue, S. C., 2017, Kinetic processes for plastic deformation of olivine in the Poyi ultramafic intrusion, NW China: Insights from the textural analysis of a similar to 1700 m fully cored succession: Lithos, v. 284, p. 462-476.

Zavada, P., Desbois, G., Urai, J. L., Schulmann, K., Rahmati, M., Lexa, O., and Wollenberg, U., 2015, Impact of solid second phases on deformation mechanisms of naturally deformed salt rocks (Kuh-e-Namak, Dashti, Iran) and rheological stratification of the Hormuz Salt Formation: Journal of Structural Geology, v. 74, p. 117-144.

Zhao, N. L., Hirth, G., Cooper, R. F., Kruckenberg, S. C., and Cukjati, J., 2019, Low viscosity of mantle rocks linked to phase boundary sliding: Earth and Planetary Science Letters, v. 517, p. 83-94.

**List 2: Papers on ice deformation that recognise GBS as a potential process in ice deformation**

GBS inferred from mechanical data
GBS observation in v. course individual GBs (e.g. columnar)inc X-ray and TEM studies of individual boundaries
GBS used to explain geophysical phenomena (planetary in particular)
GBS inferred from microstructures
GBS inferred from microstructure and mechanics
GBS inferred from observations in natural systems
GBS used in modelling

**BOLD: not Otago/ UPenn or related research**

**Barr, A. C., and Pappalardo, R. T., 2005, Onset of convection in the icy Galilean satellites: Influence of rheology: Journal of Geophysical Research-Planets, v. 110, no. E12.**

**Barnes, P., Tabor, D., and Walker, J. C. F., 1971, Friction and creep of polycrystalline ice: Proceedings of the Royal Society of London Series a-Mathematical and Physical Sciences, v. 324, no. 1557, p. 127-&.**

Caswell, T. E., Cooper, R. F., and Goldsby, D. L., 2015, The constant-hardness creep compliance of polycrystalline ice: Geophysical Research Letters, v. 42, no. 15, p. 6261-6268.

Craw, L., Qi, C., Prior, D. J., Goldsby, D. L., and Kim, D., 2018, Mechanics and microstructure of deformed natural anisotropic ice: Journal of Structural Geology, v. 115, p. 152-166.

**Cuffey, K. M., Conway, H., Gades, A., Hallet, B., Raymond, C. F., and Whitlow, S., 2000, Deformation properties of subfreezing glacier ice: Role of crystal size, chemical impurities, and rock particles inferred from in situ measurements: Journal of Geophysical Research-Solid Earth, v. 105, no. B12, p. 27895-27915.**

Durham, W. B., Prieto-Ballesteros, O., Goldsby, D. L., and Kargel, J. S., 2010, Rheological and Thermal Properties of Icy Materials: Space Science Reviews, v. 153, no. 1-4, p. 273-298.

**Duval, P.: Grain Growth and Mechanical Behaviour of Polar Ice, Annals of Glaciology, 6, 79–82, doi:10.3189/1985AoG6-1-79-82, 1985.**

**Elvin, A. A., and Sunder, S. S., 1996, Microcracking due to grain boundary sliding in polycrystalline ice under uniaxial compression: Acta Materialia, v. 44, no. 1, p. 43-56.**

**Faria, S. H., Weikusat, I., and Azuma, N., 2014, The microstructure of polar ice. Part II: State of the art: Journal of Structural Geology, v. 61, p. 21-49.**

**Freeman, J., Moresi, L., and May, D. A., 2006, Thermal convection with a water ice I rheology: Implications for icy satellite evolution: Icarus, v. 180, no. 1, p. 251-264.**

**Freitag, J., Kipfstuhl, S., and Faria, S. H., 2008, The connectivity of crystallite agglomerates in low-density firn at Kohnen station, Dronning Maud Land, Antarctica, *in* Schneebeli, M., ed., Annals of Glaciology, Vol 49, 2008, Volume 49, p. 114-+.**

Goldsby, D. L., 2006, Superplastic Flow of Ice Relevant to Glacier and Ice-Sheet Mechanics, *in* Knight, P. G., ed., Glacier Science and Environmental Change: Oxford, Blackwell, p. 308-314.

Goldsby, D. L., and Kohlstedt, D. L., 1997, Grain boundary sliding in fine-grained Ice I: Scripta Materialia, v. 37, no. 9, p. 1399-1406.

Goldsby, D. L., and Kohlstedt, D. L., 2001, Superplastic deformation of ice: Experimental observations: Journal of Geophysical Research-Solid Earth, v. 106, no. B6, p. 11017-11030.

Golding, N., Durham, W. B., Prior, D. J., and Stern, L. A., 2020, Plastic Faulting in Ice: Journal of Geophysical Research: Solid Earth, v. 125, no. 5, p. e2019JB018749.

**Han, L. J., and Showman, A. P., 2011, Coupled convection and tidal dissipation in Europa's ice shell using non-Newtonian grain-size-sensitive (GSS) creep rheology: Icarus, v. 212, no. 1, p. 262-267.**

**Hawley, R. L., and Morris, E. M., 2006, Borehole optical stratigraphy and neutron-scattering density measurements at Summit, Greenland: Journal of Glaciology, v. 52, no. 179, p. 491-496.**

**Hondoh, T., and Higashi, A., 1983, Generation and absorption of dislocations at large-angle grain-boundaries in deformed ice crystals: Journal of Physical Chemistry, v. 87, no. 21, p. 4044-4050.**

**Ignat, M., and Frost, H. J., 1987, Grain-boundary sliding in ice: Journal De Physique, v. 48, no. C-1, p. 189-195.**

**Iliescu, D., Murdza, A., Schulson, E. M., and Renshaw, C. E., 2017, Strengthening ice through cyclic loading: Journal of Glaciology, v. 63, no. 240, p. 663-669.**

**Kuiper, E. J. N., de Bresser, J. H. P., Drury, M. R., Eichler, J., Pennock, G. M., and Weikusat, I., 2020a, Using a composite flow law to model deformation in the NEEM deep ice core, Greenland – Part 2: The role of grain size and premelting on ice deformation at high homologous temperature: The Cryosphere, v. 14, no. 7, p. 2449-2467.**

Kuiper, E. J. N., Weikusat, I., de Bresser, J. H. P., Jansen, D., Pennock, G. M., and Drury, M. R., 2020b, Using a composite flow law to model deformation in the NEEM deep ice core, Greenland – Part 1: The role of grain size and grain size distribution on deformation of the upper 2207 m: The Cryosphere, v. 14, no. 7, p. 2429-2448.

Liu, F. P., Baker, I., and Dudley, M., 1993, Dynamic observations of dislocation generation at grain-boundaries in ice: Philosophical Magazine a-Physics of Condensed Matter Structure Defects and Mechanical Properties, v. 67, no. 5, p. 1261-1276.

Liu, F. P., Baker, I., and Dudley, M., 1995, Dislocation-grain boundary interactions in ice crystals: Philosophical Magazine a-Physics of Condensed Matter Structure Defects and Mechanical Properties, v. 71, no. 1, p. 15-42.

Min, S., Cole, D. M., and Baker, I., 2007, Effect of fine particles on the flow behavior of polycrystalline ice - (II) anelastic behavior: Chinese Journal of Geophysics-Chinese Edition, v. 50, no. 4, p. 1156-1160.

Qi, C., Prior, D. J., Craw, L., Fan, S., Llorens, M. G., Griera, A., Negrini, M., Bons, P. D., and Goldsby, D. L., 2019, Crystallographic preferred orientations of ice deformed in direct-shear experiments at low temperatures: The Cryosphere, v. 13, no. 1, p. 351-371.

Qi, C., Stern, L. A., Pathare, A., Durham, W. B., and Goldsby, D. L., 2018, Inhibition of Grain Boundary Sliding in Fine-Grained Ice by Intergranular Particles: Implications for Planetary Ice Masses: Geophysical Research Letters, v. 45, no. 23, p. 12757-12765.

Ruiz, J., 2010, Equilibrium Convection on a Tidally Heated and Stressed Icy Shell of Europa for a Composite Water Ice Rheology: Earth Moon and Planets, v. 107, no. 2-4, p. 157-167.

Song, M., 2006, Effects of particles on the anelasticity of granular ice: Philosophical Magazine Letters, v. 86, no. 10, p. 669-672.

Song, M., Cole, D. M., and Baker, I., 2004, Initial experiments on the effects of particles at grain boundaries on the anelasticity and creep behavior of granular ice, in Jacka, J., ed., Annals of Glaciology, Vol 39, 2004, Volume 39, p. 397-401.

Song, 2006, An investigation of the effects of particles on creep of polycrystalline ice: Scripta Materialia, v. 55, no. 1, p. 91-94.

Weiss, J., and Schulson, E. M., 2000, Grain-boundary sliding and crack nucleation in ice: Philosophical Magazine a-Physics of Condensed Matter Structure Defects and Mechanical Properties, v. 80, no. 2, p. 279-300.

**References cited in the responses**

Adams, B. L., Wright, S. I., and Kunze, K., 1993, Orientation Imaging - the Emergence of a New Microscopy: Metallurgical Transactions A-Physical Metallurgy and Materials Science, v. 24, no. 4, p. 819-831.

Ajaja, O., 1991, Role of recovery in high temperature constant strain rate deformation: Journal of Materials Science, v. 26, no. 24, p. 6599–6605, doi:10.1007/BF02402651.

Bailey, J. E., and P. B. Hirsch, 1960, The dislocation distribution, flow stress, and stored energy in cold-worked polycrystalline silver: Philosophical Magazine, v. 5, no. 53, p. 485–497.

Beeman, M., W. B. Durham, and S. H. Kirby, 1988, Friction of ice: Journal of Geophysical Research: Solid Earth, v. 93, no. B7, p. 7625–7633, doi:10.1029/JB093iB07p07625.

Bestmann, M., and D. J. Prior, 2003, Intragranular dynamic recrystallization in naturally deformed calcite marble: diffusion accommodated grain boundary sliding as a result of subgrain rotation recrystallization: Journal of Structural Geology, v. 25, no. 10, p. 1597–1613.

Bourcier, M., M. Bornert, A. Dimanov, E. Héripré, and J. L. Raphanel, 2013, Multiscale experimental investigation of crystal plasticity and grain boundary sliding in synthetic halite using digital image correlation: Journal of Geophysical Research: Solid Earth, v. 118, no. 2, p. 511–526.

Coutinho, Y. A., Rooney, S. C. K., and Payton, E. J., 2017, Analysis of EBSD Grain Size Measurements Using Microstructure Simulations and a Customizable Pattern Matching Library for Grain Perimeter Estimation: Metallurgical and Materials Transactions a-Physical Metallurgy and Materials Science, v. 48A, no. 5, p. 2375-2395.

Drury, M., and Humphreys, F. J., 1988, Microstructural shear criteria associated with grain boundary sliding during ductile deformation: Journal of Structural Geology, v. 10, no. 1, p. 83-89.

Duval, P., 1985, Grain-growth and mechanical-behavior of polar ice: Annals of Glaciology, v. 6, p. 79-82.

Faria, S. H., Weikusat, I., and Azuma, N., 2014, The microstructure of polar ice. Part II: State of the art: Journal of Structural Geology, v. 61, p. 21-49.

Falus, G., A. Tommasi, and V. Soustelle, 2011, The effect of dynamic recrystallization on olivine crystal preferred orientations in mantle xenoliths deformed under varied stress conditions: Journal of Structural Geology, v. 33, no. 11, p. 1528–1540.

Gomez-Rivas, E., A. Griera, M. G. Llorens, P. D. Bons, R. A. Lebensohn, and S. Piazolo, 2017, Subgrain rotation recrystallization during shearing: Insights from full-field numerical simulations of halite polycrystals: Journal of Geophysical Research: Solid Earth, v. 122, no. 11, p. 8810–8827.

Guillope, M., and J. P. Poirier, 1979, Dynamic recrystallization during creep of single‐crystalline halite: An experimental study: Journal of Geophysical Research, v. 84, no. B10, p. 5557–5567, doi:10.1029/JB084iB10p05557.

Hasegawa, M., and H. Fukutomi, 2002, Microstructural Study on Dynamic Recrystallization and Texture Formation in Pure Nickel: Materials Transactions, v. 43, no. 5, p. 1183–1190.

Hasegawa, M., M. Yamamoto, and H. Fukutomi, 2003, Formation mechanism of texture during dynamic recrystallization in γ-TiAl, nickel and copper examined by microstructure observation and grain boundary analysis based on local orientation measurements: Acta Materialia, v. 51, no. 13, p. 3939–3950.

Humphreys, F. J., and Hatherley, M., 1996, Recrystallization and Related Annealing Phenomena, Elsevier Science.

Humphreys, F. J., 2001, Review - Grain and subgrain characterisation by electron backscatter diffraction: Journal of Materials Science, v. 36, no. 16, p. 3833-3854.

Humphreys, F. J., 2004, Reconstruction of grains and subgrains from electron backscatter diffraction maps: Journal of Microscopy-Oxford, v. 213, p. 247-256.

McCarthy, C., H. Savage, and M. Nettles, 2017, Temperature dependence of ice-on-rock friction at realistic glacier conditions: Philosophical Transactions of the Royal Society A: Mathematical, Physical and Engineering Sciences, v. 375, no. 2086, p. 20150348–20150348.

Poirier, J. P., and A. Nicolas, 1975, Deformation-Induced Recrystallization Due to Progressive Misorientation of Subgrains, with Special Reference to Mantle Peridotites: The Journal of Geology, v. 83, no. 6, p. 707–720.

Ree, J. H., 1994, Grain-Boundary Sliding and Development of Grain-Boundary Openings in Experimentally Deformed Octachloropropane: Journal of Structural Geology, v. 16, no. 3, p. 403-418.

Rollett, A. D., Rohrer, G. S., and Humphreys, F. J., 2017, Recrystallization and Related Annealing Phenomena, Elsevier Science.

Signorelli, J., and Tommasi, A., 2015, Modeling the effect of subgrain rotation recrystallization on the evolution of olivine crystal preferred orientations in simple shear: Earth and Planetary Science Letters, v. 430, p. 356-366.

Sintay, S., Groeber, M. A., and Rollett, A. D., 2009, 3D Reconstruction of Grains, in Schwartz, A. J., Kumar, M., Adams, B. L., and Field, D. P., eds., Electron Backscatter Diffraction in Materials Science.

Sundberg, M., and R. F. Cooper, 2008, Crystallographic preferred orientation produced by diffusional creep of harzburgite: Effects of chemical interactions among phases during plastic flow: Journal of Geophysical Research, v. 113, no. B12, p. 498–16.

Trimby, P. W., D. J. Prior, and J. Wheeler, 1998, Grain boundary hierarchy development in a quartz mylonite : Journal of Structural Geology, v. 20, no. 7, p. 917–935.

White, S., 1973, Syntectonic Recrystallization and Texture Development in Quartz: Nature, v. 244, no. 5414, p. 276-278.

**Response to Reviewer 2**

We thank Reviewer 2 for his thoughtful and helpful review of our paper for a second time. Reviewers comments are in blue type. Highlight: new text in the manuscript highlighted in the same colour.

*Grain boundary sliding (GBS)*

The main reviewer concern is still why we include GBS as an interpretation. The reviewer encourages us to further explore the effect of complex dislocation processes that might leave similar signals in microstructural or mechanical data as GBS. The development of subgrain boundaries is the manifestation of dislocation activities. Subgrain rotation recrystallization, through which low-angle subgrain boundaries comprised of arrays of dislocations can be transformed into high-angle grain boundaries, is interpreted as a dominant dynamic recrystallization process in this study. Therefore, we tested the role of subgrain rotation recrystallization on the weakening of CPO within a small grain population using misorientation data. These data were added after the first review. The misorientation data were clearly not used or discussed effectively in our last revision of the manuscript. In this newly modified manuscript, these misorientation data are more extensively discussed. It is the misorientation axis data that push us to maintain inclusion of GBS as a possible deformation mechanism. We discuss in detail below why the misorientation data are key to our interpretation.

The accepted understanding of subgrain rotation recrystallisation is that immediately upon nucleation of a new grain, the nucleus should have an orientation relative to the parent that reflects the dislocation structure of the final subgrain boundary segment at the moment it becomes a grain boundary. This is implicit in the first descriptions of the subgrain rotation recrystallisation (Poirier and Guillope, 1979; Poirier and Nicolas, 1975), is built into numerical simulations of the process (Gomez-Rivas et al., 2017; Signorelli and Tommasi, 2015) and is stated in all editions of John Humphrey's textbook (Humphreys and Hatherley, 1996; Rollett et al., 2017): "The orientation of the nucleus is present in the deformed structure. There is no evidence that new orientations are formed during or after nucleation, except by twinning" (from section 6.63, 7.63 or 7.62 depending on the edition).

The rotation recrystallisation model, at least in part, was erected on the basis of misorientation data, although such data were very difficult to collect at the time (Poirier and Guillope, 1979; Poirier and Nicolas, 1975; White, 1973). EBSD has made acquisition of such data much easier. One of the authors (Prior) was involved in one of the first more extensive EBSD investigations (Bestmann and Prior, 2003) of microstructures that would conventionally be explained by recovery, subgrain rotation and subgrain rotation recrystallisation. In this study subgrains and recrystallised grains of approximately the same size are developed in a shear zone within a large crystal (the parent). Low-angle boundaries surrounding subgrains have misorientation axes in rational crystallographic orientations as would be expected for subgrain rotation. High-angle grain boundaries, surrounding recrystallised grains, have misorientation axes that are randomly oriented within the crystal, inconsistent with the operation of subgrain rotation recrystallisation alone. Bestmann and Prior (2003) chose to explain these observations by allowing the newly formed nuclei to change orientation by sliding on grain boundaries. Comparable observations have been made in many dynamically recrystallised rocks (see citations of Bestmann and Prior (2003) and the reference list later on) and many have been interpreted in the same way.

In our paper, the ice misorientation axes for low- and high-angle boundaries have the same general pattern as that identified by Bestmann and Prior (2003). Low-angle boundaries have misorientation axes that are oriented tightly in the basal plane, consistent with these boundaries comprising arrays of dislocations (primarily basal dislocations) that have developed by recovery and subgrain rotation. High-angle boundaries, on the other hand, do not have misorientation axes oriented in the basal plane, nor in any other preferred crystallographic orientation. This relationship still holds when the data are restricted to the high-angle boundaries between central large grains (parents?) and surrounding small grains in the "core and mantle" structures of the lower-temperature experiments. Explaining the misorientation data requires either a different nucleation mechanism (i.e. not rotation recrystallisation) or an additional process to change the orientation of grains during or after nucleation. The common interpretation of comparable data in the rock deformation literature (not just our group) is that GBS changes the orientation of grains after nucleation. Thus, we choose to keep this as our preferred interpretation.

We continue to emphasise that GBS is just one interpretation and we present other possibilities that can explain the same observations. We feel it is important to include the GBS interpretation so that future generations of scientists can test it's validity.

**Section two: point-by-point reply to comments**

**R2.1:** Page 19, lines 5-18. I don't like the discussion on GBS which I find extremely restrictive (less than the first version of the paper but still). Apparently the authors want to prove that if there is a grain size sensitivity of the mechanical behaviour (weakening) then it must come from GBS. This is without remembering that plasticity essentially comes from dislocation slip, and that dislocations interact with other dislocations, with grain boundaries, with the internal stress field, can annihilate or climb or be nucleated, etc. leading to many possible size effects. One could for example rewrite line 9-10 "The grain size sensitivity is hidden in the flow stress data as grain size becomes controlled by the flow stress" by "The dislocation processes are hidden in the flow stress data as dislocation motion is controlled by the flow stress". But there is unfortunately no mention of possible complex dislocation processes in the paper, that are well known to be size dependent (in a complex way, the literature is abundant !). Size effects due to strain-gradient plasticity could also be invoked (e.g. work by Geers, Forest, Fleck, etc). Line 14, the sentence "Without GBS another explanation is needed for the grain size sensitivity" require a more open discussion based on all other possible weakening mechanisms. Concerning that point, the response letter cannot be considered as satisfactory. Page 27, concerning the (inverse) Hall-Petch effect, the authors write: "Some recent work relates GBS associated with the inverse Hall-Petch relationship with amorphization of the grain boundaries (Guo et al., 2018) and a molecular dynamics modelling study of ice (Cao et al., 2018) generates an inverse Hall-Petch relationship that involves a combination of GBS, grain rotation, amorphization and recrystallization, phase transformation, and dislocation nucleation in both bicrystals and polycrystals. ". BUT: Guo's paper is on nanocrystalline superhard materials (hardness similar than diamond), and Cao's paper is a MD computational study on 40nm polycrystalline ice with flow stress around 600MPa ! Of course GBS can occurs in some materials but these two references on nm grains have nothing to do with the hundred-microns ice grains considered here. This is not to say that GBS is not active in ice, but the authors should be convincing by using a correct argumentation. That GBS occurs in some nanometric superhard material does not guarantee that it also occurs in ice.

We thank the reviewer for these thoughtful comments. The discussion of the role of grain size-sensitive processes (including GBS) in mechanical evolution is has been substantially reduced. Because this paper is focused on the microstructural and CPO evolution of deformed ice samples, a full discussion of all issues related to grain size-sensitivity, which are quite complex as pointed out by the reviewer, is beyond the scope of this paper. We have another manuscript in preparation wherein grain size-sensitivity is the central topic and we consider these helpful discussion points in writing that paper.

**R2.2:** Page 16 line 16 : similarly, strange to put reference to Elati et al. here to show that GBS happens in some fine grain structure materials. This reference is on a High Entropy Alloy, which may differ significantly from ice deformation (if not please justify). There are other materials in which GBS is active without having to go to fine grains, such as Salt (see papers by Bornert, Dimanov and Bourcier).

We have removed the Eleti et al reference in the modified manuscript.

**R2.3:** Page 8 line 10 : as already indicated in my first review, the distribution shown in figure 4 can really not be described as a "bimodal" one ! Same line 17 (bimodal grain size). Please modify.

Thanks to the reviewer for pointing out this mistake. We have modified the statement in section 3.3.2 in accordance with the reviewer's comment.

**R2.4:** Page 15 line 4 : I don't know whether this is a standard treatment in the glaciology or mineral physics community, but the identification of an activation energy for the "flow stress" (or tertiary creep in "constant load experiments") is not accurate, as the sample microstructures vary between specimens.

The identification of an activation energy for the "flow stress" is a common treatment in the rock deformation community (e.g. Hirth et al., 2001; Hirth, 2002; Renner and Evans, 2002; Karato, 2008; Hansen et al, 2012) and has been applied to an ice flow law (Durham et al, 1983). It can be considered as empirical. Arrhenius plots with large data sets often produce good straight-line fits to the data (e.g., Durham et al., 1983).

**R2.5:** Page 2 line 25 : "displacement rate" instead of "displacement"

Corrected.

[revised manuscript text omitted]
}$ (μm) | $_6D_{median}$ (μm) | $_7D_{q,25\%}$ (μm) | $_8D_{q,75\%}$ (μm) | $_9D_{SMR}$ (μm) | $_{10}\overline{D_{big}}$ (μm) | $_{11}\overline{D_{small}}$ (μm) | $_{12}D_{peak}$ (μm) | $_{13}\bar{d}/_{14}d_{median}$ (μm) | $_{15}d_{q,25\%}/_{16}d_{q,75\%}$ (μm) | $_{17}d_{peak}$ (μm) |
|---|---|---|---|---|---|---|---|---|---|---|---|---|---|---|---|---|
| | | | | | | | | | | | | | | | $_{18}\varphi \geq 2°$ | |
| undeformed | - | - | 1.90 | 9.97E-06 | 1.00 | 297 | 291 | 165 | 413 | 274 | - | - | 300 | 291/280 | 161/392 | - |
| PIL176 | -10 | 0.03 | 9.45 | 3.24E-05 | 3.25 | 156 | 117 | 48 | 250 | 132 | 250 | 51 | 30 | 134/79 | 39/219 | 20 |
| PIL163 | -10 | 0.05 | 11.71 | 4.75E-05 | 4.76 | 125 | 98 | 54 | 171 | 110 | 197 | 58 | 35 | 104/77 | 51/162 | 25 |
| PIL178 | -10 | 0.08 | 13.47 | 3.82E-05 | 3.83 | 140 | 119 | 72 | 188 | 127 | 194 | 63 | 55 | 127/108 | 62/170 | 50 |
| PIL177 | -10 | 0.12 | 14.19 | 5.14E-05 | 5.15 | 114 | 90 | 54 | 155 | 101 | 184 | 59 | 40 | 96/77 | 45/129 | 30 |
| $_1$PIL007 | -10 | 0.19 | 13.07 | 6.25E-05 | 6.27 | 106 | 88 | 51 | 143 | 96 | 174 | 58 | 50 | 96/78 | 46/129 | 45 |
| PIL254 | -20 | 0.03 | 7.40 | 5.75E-05 | 5.77 | 114 | 62 | 36 | 174 | 93 | 197 | 38 | 25 | 91/46 | 29/106 | 20 |
| PIL182 | -20 | 0.04 | 5.30 | 3.97E-05 | 3.98 | 148 | 122 | 62 | 220 | 131 | 188 | 42 | 30 | 103/67 | 33/146 | 25 |
| PIL184 | -20 | 0.08 | 10.61 | 4.73E-05 | 4.74 | 122 | 89 | 48 | 164 | 105 | 169 | 42 | 45 | 88/58 | 36/109 | 20 |
| PIL185 | -20 | 0.12 | 7.76 | 1.05E-04 | 10.49 | 75 | 53 | 36 | 85 | 66 | 132 | 41 | 30 | 55/40 | 28/63 | 20 |
| PIL255 | -20 | 0.20 | 12.29 | 1.28E-04 | 12.85 | 64 | 53 | 36 | 81 | 59 | 106 | 41 | 30 | 55/46 | 32/68 | 25 |
| PIL165 | -30 | 0.03 | 2.07 | 3.15E-05 | 3.16 | 149 | 108 | 48 | 241 | 126 | 203 | 38 | 40 | 108/60 | 32/152 | 20 |
| PIL162 | -30 | 0.05 | 4.87 | 7.27E-05 | 7.29 | 103 | 76 | 45 | 135 | 91 | 144 | 40 | 35 | 70/49 | 31/86 | 20 |
| PIL164 | -30 | 0.07 | 5.58 | 6.67E-05 | 6.69 | 98 | 61 | 39 | 113 | 82 | 158 | 39 | 30 | 59/38 | 27/65 | 20 |
| PIL166 | -30 | 0.12 | 6.01 | 1.34E-04 | 13.45 | 67 | 54 | 37 | 79 | 61 | 104 | 70 | 35 | 57/47 | 32/69 | 25 |
| PIL268 | -30 | 0.21 | 5.66 | 1.18E-04 | 11.88 | 60 | 37 | 29 | 53 | 50 | 158 | 35 | 30 | 42/30 | 24/41 | 20 |

[revised manuscript text omitted]

---

## Author Response (AR3)

**This document is comprised by:**

**1. Response to the editor (P1)**

**2. Response to reviewer 1 (P1)**

**3. Response to reviewer 2 (P1-P5)**

**3. A list of all relevant changes made in the manuscript (P6)**

**4. Revised manuscript with changes marked-up**

**Response to the editor**

We would like to thank the editor for pointing out (1) we should upload the data analysed in this study, and (2) the format of reference is wrong. We have corrected these mistakes in the modified manuscript.

**Response to Reviewer 1**

We thank Reviewer 1 for her thoughtful and helpful reviews of our paper. No correction is required.

**Response to Reviewer 2**

We thank Reviewer 2 for his thoughtful and helpful review of our paper for a third time. Reviewers comments are in blue type. *New text is in italics.*

**R2.1:** ** item 1 of page 20 about gbs : The role of gbs is still not clear to me as I've already explain in my previous reviews, as well as the (poor) evidence of gbs occurence. I also really wonder how GBS can modify the boundary misorientation. This is not discussed in the paper although it should be central to understand how gbs is believed to modify the microstructures. I check some of the cited literature (Jiang et al 2000, Goldsby and Kohlstedt 1997, 2001, Bestmann and Prior 2003, Cross and others 2017b) in which this is not explained either. By the way, the results obtained in previous published paper seems to be improved in this new paper. For example, my understanding of Cross et al 2017b is that these authors "propose" gbs as an active mechanisms, i.e. as one possible solution (bottom of before-last page). In the manuscript, it is written (page 20 line 18) : "Cross et al 2017b found evidence for specific orientation being randomized by gbs". To me, there is a clear difference between a "proposition" and an "evidence".

We thank the reviewer for pointing out our description of Cross and others (2017b) is misleading. In the modified manuscript, we have modified the sentence in section 4.1.4 to: *Recently, Cross and others (2017b) proposed GBS as a possible explanation to the randomization of specific orientations (i.e., those lying in the plane of maximum vorticity).*"

The key concern from the reviewer is: "how GBS can modify the boundary misorientation". Please see the beautiful work done by Maruyama and Hiraga (2017), which used fiducial markers to directly observe and quantify relative sliding between neighbouring grains. Fig. R2.1(Figure 11(b) from Maruyama and Hiraga., 2017) is the schematic drawing illustrating the model of how will GBS lead to a rotation of grains. During grain boundary sliding, anisotropic shear stress resolved on the assemblage of boundaries can generate torque on the grain (Wheeler, 2009, 2010). The torque and thus lead to the rotation of grain and a change of boundary misorientations (Fig. R2.1, Maruyama and Hiraga, 2017).

In detail this means that the rotation effects will depend on grain shapes and grain boundary orientations. Thus, in materials where grains are strongly shaped, and that shape corresponds to crystallography GBM can lead to a CPO. A good example of this is plagioclase feldspar where experiments that yield a Newtonian (n=1) rheology, interpreted as diffusion creep, also show the development of very strong CPOs (Barreiro et al., 2007). This issue and its implications are explored by Maruyama and Hiraga (2017) (see their Fig. 13).

[Figure]

(b)

**Figure R2.1.** Schematic illustrations of grain rotation models during GBS process (from Maruyama et al., 2017). Red lines correspond to preferential slip planes that are oriented in the direction of $\varphi_0$ prior to the deformation and of $\varphi$ after the deformation. Shear due to resolved shear stress ($\tau$) on the preferential slip planes and its consequence on grain rotation are indicated by arrows. $\theta$ represents rotation angle.

**R2.2:** ** section 4.2, last three lines : as indicated in my last review, there are other candidates than the 3 indicated here. For example, according to Orowan's equation, an increase in the density of mobile dislocations during strain (eg. by the activation or multiplication of dislocations sources) can also lead to significant weakening. The authors could explain why they believe that dislocation processes don't come in play here.

We agree with the reviewer that the interaction of dislocations during the deformation is likely to increase the dislocation density and thus impact the mechanical behaviour. Studies of fine-grained polycrystalline materials suggest the strain energy density, which controls the mechanical behaviour, is a complex function of strain and strain gradient, and the strain gradient is grain size sensitive (Hashin and Shtrikman, 1962a, 1962b). Modelling incorporating grain size effects on strain-gradient plasticity shows finer grains should be stiffer than coarser grains, if other processes are not involved in the deformation (Smyshlyaev and Fleck, 1996). Such modelling result suggests if strain-gradient plasticity dominates the mechanical behaviour of ice deformed in this study, the standard ice samples should have a higher peak stress than the coarse-grained ice samples. Moreover, the well-known Orowan stress equation (Eq. (R2.1), Humphreys et al., 2017) suggests the stress, $\tau$, is a function of shear modulus, $G$, Burgers vector, $b$, and channel width, $\lambda$. Grain size decreases with an increasing strain for all deformed ice samples (section 3.2.2, Fig. 4-6, 11(a), Table 3). The channel width, $\lambda$, is likely to decrease with grain size. Therefore, the stress, $\tau$, should increase with strain as predicted by Eq. (R2.1).

$$\tau = \frac{Gb}{\lambda} \qquad (R2.1)$$

However, our mechanical data show stress decreases with strain after peak (Fig. 3). Therefore, we suggest grain size effect of strain-gradient plasticity is unlikely to dominate the

stress drop after peak observed in this study. This is the reason why we chose not to include the complicated dislocation interactions as a weakening mechanism in section 4.2.

**R2.3:** ** page 6 line 25, "Misorientation angle distributions are illustrated as " is written twice.

Corrected.

**R2.4:** ** page 8 line 15, notation D_SMR=\bar sqrt(D**2) is misleading. It seems to be the square of the square root, i.e. D ??

Thanks for pointing this out. It has been modified to: $D_{SMR} = \left( \overline{\sqrt{D}} \right)^2$

**R2.5:** ** page 8 line 17 : there is still mention to a "bimodal" grains size distribution which is not visible in any of the presented data… Please see my previous review.

Apologise. We have modified such sentence to: "*To better describe the statistics of the skewed grain size distributions….*"

**R2.6:** ** page 8 line 19 : "grain size converge as the strain increases" … converges towards which value ?? please indicate.

We have modified such sentence to: "$\overline{D}$, $D_{SMR}$, $D_{median}$ and $D_{peak}$ have the relation of $\overline{D} > D_{SMR} > D_{median} > D_{peak}$, and converge to $D_{median}$ as the strain increases (Fig. 4 (d), 5 (d), 6 (d) and Table 3)."

**R2.7:** ** page 21 line 24 : "analytical numerical models" is weird… either analytical, or numerical ?

We have modified such sentence to: "*…. and there are numerical models that seek to quantify this relationship ….*"

We corrected the format of references based on the requirement.

[revised manuscript text omitted]